# A Comparative Study of Label-free Representation Quality Metrics in Deep Learning

## Abstract

We present a comparative study of label-free metrics for assessing the quality of representations in deep neural networks to understand their reliability under a wide variety of configurations. We group existing label-free metrics into three families based on their construction and analytically establish connections between metrics within the same family. We then characterise the sensitivity of spectral metrics through controlled synthetic experiments. Finally, all label-free metrics are evaluated against downstream task accuracy across a diverse set of 260 vision models on six datasets spanning generic object classification, fine-grained object classification, scene recognition and geospatial task, stratifying results by architecture class and training objective. We find that intrinsic dimensionality (ID) is a reliable predictor in most setting, while the reliability of other metrics is strongly moderated by architecture class and training objective. Our results provide a clearer understanding of what label-free representation quality metrics measure, when they are reliable, and how to interpret them in practice.

## 1 Introduction

The popularity of deep learning research has largely been driven by supervised learning, where models are trained to perform a sequence of transformations from input data to corresponding labels using large annotated datasets (Krizhevsky et al., 2012; LeCun et al., 1998). The intermediate representations learnt by these models were shown to retain sufficient structure useful for new datasets and tasks (Yosinski et al., 2014; Donahue et al., 2014). The transferability of these models to other datasets inspired research on building general-purpose feature extractors that do not depend on task-specific supervision. Self-supervised learning (SSL) has emerged as the dominant paradigm for this purpose, with methods based on contrastive learning (Chen et al., 2020; He et al., 2020; Bardes et al., 2022), self-distillation (Caron et al., 2021), and masked reconstruction (He et al., 2022), demonstrating that models trained without labels can produce representations that match or exceed those of their supervised counterparts across a wide range of downstream tasks.

With the increasing availability of such models through public repositories such as HuggingFace (Wolf et al., 2020), selecting a model to use as a feature extractor for a given task has become a challenge. In practice, the suitability of a model for a task is evaluated using simple algorithms, including k-nearest neighbours (Wu et al., 2018), linear probing (Kornblith et al., 2019), or full fine-tuning (Kornblith et al., 2019). However, as the number of publicly available models continues to grow, an exhaustive search over all candidates becomes computationally expensive. A parallel line of work has sought to identify the characteristics that a good representation space should exhibit for any downstream task. These studies examine the covariance structure (Agrawal et al., 2022; Garrido et al., 2023) and intrinsic dimensionality (Ansuini et al., 2019) of the representation manifold. Crucially, the proposed metrics that capture these properties are computed solely from the features themselves. This has motivated their use as label-free metrics that require no training, offering a computationally efficient alternative for model selection.

The increasing usage and number of label-free metrics raise important questions about when to use which metric as well as how reliable they are across different architectures, training objectives, and datasets. Existing evaluations (Agrawal et al., 2022; Garrido et al., 2023) tend to validate a single metric against

a narrow set of architectural and training configurations, making it difficult to draw general conclusions about their reliability. Furthermore, several spectral metrics are constructed from a common underlying eigenspectrum, yet it remains unclear how they differ in their sensitivity to spectral shape and representation dimensionality. Since these metrics are computed solely from a geometric perspective, without any knowledge of the downstream task, some degree of unreliability is to be expected a priori. Thus, examining the conditions under which these metrics serve as reliable proxies for downstream performance is of considerable value.

We perform a comparative analysis of representation quality metrics to evaluate their reliability with respect to training configurations as well as their relation to downstream performance. Our main contributions are:

- **A taxonomy for label-free metrics** that groups them into three families based on their construction: (i) Covariance spectrum-based, (ii) Pairwise relation-based, and (iii) Manifold-based. We further bring their formulations into a common framework to highlight similarities between metrics within each group.

- **Sensitivity analysis of spectral metrics** through controlled synthetic experiments that characterise how covariance spectrum-based metrics respond to changes in the shape of the eigenspectrum and in representation dimensionality.

- **A systematic study** demonstrating the correlation between the metrics and test accuracy from an Ordinary Least Squares fit, as well as their consistency when varying architecture class, and training objective.

Broadly, our results suggest that existing label-free metrics should be used with caution, as their reliability is influenced by model architecture and training objective. We hope our insights are useful in developing more robust and generalisable representation quality metrics.

## 2 Related Work

Representation learning aims to extract informative features from raw data that facilitate efficient transfer to downstream tasks. Self-supervised learning (SSL) emerged as the dominant paradigm for this purpose, with methods based on contrastive learning (Chen et al., 2020; He et al., 2020; Bardes et al., 2022), self-distillation (Caron et al., 2021; Oquab et al., 2024), and masked reconstruction (He et al., 2022) producing representations that are competitive with supervised counterparts across a range of downstream tasks, especially under fine-tuning. To measure the quality of extracted representations for downstream tasks, a variety of evaluation methods have been used. Broadly, these methods can be classified into label-dependent metrics (which require labels) and label-free metrics (which do not).

**Label-dependent metrics:** Linear probing, in which a linear classifier is trained on frozen features to assess how much task-relevant information is encoded, has been shown to often be predictive of transfer accuracy (Zhang et al., 2016). A common alternative is to use K-Nearest-Neighbour (KNN) classification or clustering of the same frozen features as proxy for representation quality (Balestriero et al., 2023). Fine-tuning the entire model using the target task can also be used as proxy (He et al., 2022), but is substantially more computationally costly. More efficient transferability metrics, which do not require fine-tuning, have been proposed. These include LEEP, which estimates a joint distribution over source predictions and target labels (Nguyen et al., 2020); LogME, which computes the marginal likelihood of target labels given features (You et al., 2021); and PACTran, which computes Probably Approximately Correct Bayesian bounds on the labelled empirical risk (Ding et al., 2022). However, all of these methods still require access to target labels. This continued dependence on labelled data motivates the development of completely label-free evaluation metrics.

**Label-free metrics:** Renggli et al. (2022) showed that commonly used task-aware metrics, such as fine-tuning, linear probing, and KNN, may produce inconsistent model rankings across tasks and datasets. To avoid dependence on labels, several label-free metrics (which we detail in Section 3) have been proposed, motivated by different theoretical perspectives. These include metrics based on the spectral properties

of the representation covariance matrix (Garrido et al., 2023; Agrawal et al., 2022; He & Ozay, 2022), relational metrics that work with inter-point relations (Tsitsulin et al., 2023; Liao et al., 2024), and the local neighbourhood structure of the representation manifold (Ansuini et al., 2019). However, these metrics have largely been proposed and validated in isolation, often as by-products of new SSL methods, making it difficult to assess their reliability. A further practical limitation is that some metrics are computed on projector outputs rather than on backbone representations. For example, RankMe (Garrido et al., 2023) was originally computed this way, but projector embeddings are discarded after SSL training and are not available in publicly released models. This constraint has not been consistently addressed in prior work, limiting the applicability of such metrics to comparing models from public repositories.

Direct comparisons of label-free representation quality metrics are rare. Tsitsulin et al. (2023) compile prior label-free representation quality measures (Agrawal et al., 2022; Garrido et al., 2023; He & Ozay, 2022) and introduce additional metrics inspired by numerical linear algebra. They report aggregate results without systematically examining the influence of the model's architecture class, its representation dimension, or its training objective. More broadly, different metrics tend to favour different representation geometries. Spectral measures such as RankMe (Garrido et al., 2023) and NE Sum (He & Ozay, 2022) reward representations with high effective rank and more uniform eigenvalue distributions, while $\alpha$-Req (Agrawal et al., 2022) favour representations with a decaying covariance eigenspectrum, and relational and manifold-based metrics (Tsitsulin et al., 2023; Liao et al., 2024; Ansuini et al., 2019) favour locally structured geometries. Understanding which metrics are reliable under which conditions, and whether they generalise across architecture classes, training objectives, and representation dimensions, requires a systematic evaluation.

## 3 A Taxonomy for Label-Free Metrics

In this section, we review a range of representation quality metrics. We highlight the connections between them by reformulating them where applicable. We then propose a taxonomy that groups these metrics according to their underlying theory.

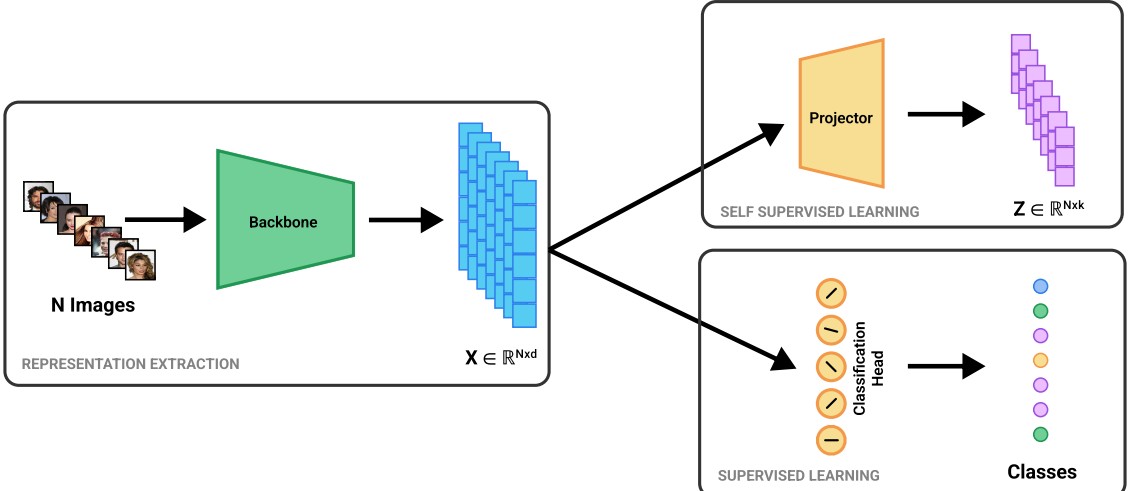

Figure 1: Illustration of the representation extraction pipeline for self-supervised (SSL) and supervised models. In SSL models, the projector MLP is discarded after training, and the backbone output $X \in \mathbb{R}^{N \times d}$ is used as the representation. For supervised models, the penultimate layer activations are used as the representation, also denoted as $X$. In both cases, representation quality metrics are computed directly on X, without access to the projector embeddings Z or classification heads.

**Preliminaries** We consider a model $f_\theta : \mathbb{R}^{n_1 \times n_2 \times \cdots \times n_k} \to \mathbb{R}^d$ that extracts features from inputs, mapping them to a $d$-dimensional representation space. For image inputs, this reduces to $f_\theta : \mathbb{R}^{W \times H \times 3} \to \mathbb{R}^d$, where W and H denote the width and height of an RGB image. We refer to this model as the encoder or backbone

model, and let $X \in \mathbb{R}^{N \times d}$ denote the corresponding extracted representations, where $N$ is the number of samples. In most self-supervised models, a Multilayer Perceptron (MLP) is usually applied to the backbone output to produce embeddings, which we denote by $Z \in \mathbb{R}^{N \times k}$. SSL loss functions are usually applied on these embeddings. Once the SSL model is fully trained, the MLP is discarded. Since our motivation comes from identifying which model to use for feature extraction, we consider the backbone output as the representation and compute the metrics on this output. For the supervised models, we consider the output of the penultimate layer as the representation (see Figure1).

A central underlying question for metrics aiming to estimate representation quality is *which structural properties of a representation correlate with strong downstream performance?* Different metrics provide complementary perspectives on this question by emphasising distinct aspects such as global variance structure, cluster separability, and local neighbourhood consistency.

Table 1 groups and summarises the representation quality metrics that we consider in our analysis. We group them into three broad categories on the basis of how they are constructed: (i) **Spectral metrics** that capture the global structure of representations through the eigenvalues of the representation covariance matrix or on the singular values of the representation matrix $X$ (Section 3.1). (ii) **Relational metrics** consider the pairwise relation between these points in the representation space as a basis for measuring representation quality (Section 3.2). (iii) **Manifold-based metric** which works with the manifold geometry of the representation (Section 3.3).

Table 1: Summary of representation quality metrics considered in this work, grouped by construction. Metrics take the representation matrix, $X \in \mathbb{R}^{N \times d}$, as input, where $N$ is the number of samples and $d$ is the representation dimension (we assume $N \gg d$).

| Metric | Category | Complexity | Description |
|---|---|---|---|
| $\alpha$-ReQ (Agrawal et al., 2022) | Spectral | $\mathcal{O}(Nd^2)$ | Fits power law $\lambda_i \propto i^{-\alpha}$ to the eigenspectrum. Sensitive to fitting range in log-log space (Clauset et al., 2009). |
| RankMe (Garrido et al., 2023) | Spectral | $\mathcal{O}(Nd^2)$ | Entropy of normalised singular values. Originally computed on projector output; we compute on backbone output. |
| NE Sum (He & Ozay, 2022) | Spectral | $\mathcal{O}(Nd^2)$ | Measures degree of feature whitening. Equivalent to Stable Rank (Tsitsulin et al., 2023) for mean-centred representations (Section 3.1). |
| $\kappa$ (Tsitsulin et al., 2023) | Spectral | $\mathcal{O}(Nd^2)$ | Ratio of largest to smallest non-zero singular value. Measures sensitivity of linear system to perturbations. |
| Self-Cluster (Tsitsulin et al., 2023) | Relational | $\mathcal{O}(Nd^2)^{\dagger}$ | Measures deviation of $\ell_2$-normalised representations from a uniform spherical distribution. Reduced from $\mathcal{O}(N^2d)$ via batched reformulation (Section 3.2). |
| DSE (Liao et al., 2024) | Relational | $\mathcal{O}(N'^2 d)$ | Spectral entropy of anisotropic diffusion kernel matrix. Subsampled to $N' = 10{,}000$ for scalability. Analytically related to Self-Cluster (Section 3.2). |
| ID (Ansuini et al., 2019) | Manifold | $\mathcal{O}(N_b^2 d)$ | MLE estimator based on ratio of first and second nearest-neighbour distances. Computed on batches of size $N_b$ and averaged. Moderately underestimates true ID when ID > 20. |

$^{\dagger}$ Reduced from $\mathcal{O}(N^2d)$ via the batched reformulation described in Section 3.2.

## 3.1 Spectral Metrics

We refer to metrics that assess representation quality through the covariance structure of the representation matrix as *spectral metrics*. These metrics operate on the eigenspectrum of the representation covariance matrix, or equivalently on the singular values of the representation matrix $X$. Intuitively, an ideal representation space should avoid *rank collapse*, i.e., variance concentrated in a small number of directions. Spectral

metrics therefore aim to capture properties such as effective rank, spectral decay, and dimensional utilisation. In this section, we restate and, where possible, reformulate them to highlight their similarities.

**$\alpha$-ReQ**   This metric is motivated by the observation that the eigenspectrum of many neural network representations approximately follows a power-law decay $\lambda_i \propto i^{-\alpha}$, often with $\alpha \approx 1$ (Agrawal et al., 2022). The exponent $\alpha$ describes the distribution of variance across principal components. A small $\alpha$ indicates a slowly decaying, near-uniform spectrum (large effective dimensionality), whereas a large $\alpha$ indicates a steep decay (small effective dimensionality). Agrawal et al. (2022) argued that there is a "Goldilocks" regime around $\alpha \approx 1$ where representations are of particularly high quality. This reasoning is supported by a theoretical result from Bartlett et al. (2020), who showed that for linear networks with power-law eigenspectra, generalisation is near-optimal in the infinite-dimensional limit if and only if $\alpha = 1$.

In practice, $\alpha$ is estimated by fitting a straight line to the eigenvalue spectrum plotted in log-log scale and using its slope as the exponent. We note that the slope estimate can be sensitive to the fitting range on the spectrum, which could bias the results (Appendix B.1). In our experiments, we use a consistent heuristic range for all models (eigenvalues indexed $i \in [10, \lfloor 0.9d \rfloor]$) in our experiments.

**RankMe**   By Cover's theorem (Cover, 1965), increasing the number of useful representation directions can make classes more linearly separable, suggesting that representations with higher effective dimensionality may yield better downstream performance. RankMe (Garrido et al., 2023) builds on this idea by measuring effective dimensionality directly, rather than through the rate of spectral decay as in $\alpha$-ReQ. It is defined as the exponential of the Shannon entropy of the normalised singular values of $X$:

$$\text{RankMe} = \exp \left( - \sum_{i=1}^{\min(N,d)} p_i \log p_i \right),$$

where $p_i = \frac{\sigma_i(X)}{\|\sigma(X)\|_1}$ and $\sigma_i$ denotes the singular values in decreasing order. Here, $\|\sigma(X)\|_1$ represents the $\ell_1$ norm of the singular value vector. A higher RankMe indicates that variance is distributed across more directions, signalling that the representation has avoided dimensional collapse. Unlike $\alpha$-ReQ, RankMe makes no power-law assumption on the spectrum.

**Note:** RankMe was originally computed on embeddings (i.e., the output of the projector, $Z \in \mathbb{R}^{N \times k}$ in Figure 1). In this work, we compute it directly on the representations, i.e., the backbone output. While Garrido et al. (2023) assume access to both the pre-trained backbone and the projector, in practice it is not realistic for model selection using existing sources, where popular hubs like Hugging Face typically provide only the backbone. Since effective rank counts every high-variance direction, including those that carry no class-discriminative signal, we hypothesise it reliably flags rank collapse but does not sufficiently capture representation quality.

**NE Sum**   He & Ozay (2022) study the spectrum between collapsed and whitened representations and show that the degree of feature whitening affects generalisation. They introduced NE Sum to measure the degree of whitening of the representation matrix $X$, i.e., how evenly the eigenvalues of the covariance matrix are distributed. It is defined by

$$\text{NE Sum} = \frac{\sum_{i=1}^{d} \lambda_i}{\max_i \lambda_i},$$

where $\lambda_i$ are the eigenvalues of the covariance matrix. NE Sum achieves a maximum when all principal components have equal variance, indicating a fully whitened (sphere-like) distribution of the features. He & Ozay (2022) provide empirical evidence that higher NE Sum values are associated with better downstream performance. We note that the metric **Stable Rank**, used by Tsitsulin et al. (2023) is equivalent to NE Sum under the assumption that the representation is centred (i.e., the columns are mean-subtracted).

Stable rank quantifies the difficulty of estimating a matrix from sub-sampled rows. In our analysis, this translates to the difficulty of estimating $X$. It is defined as follows:

$$\text{StableRank(X)} = \frac{\|X\|_F^2}{\|X\|_2^2} = \frac{\sum_{i=1}^d \sigma_i^2}{\max_i \sigma_i^2}$$

$$= \frac{\sum_{i=1}^d \lambda_i}{\max_i \lambda_i},$$

where $\sigma_i$ denotes the $i$th singular value of $X$, and, since $X$ is centred, $\sigma_i^2 = (N-1)\lambda_i$.

**Condition Number**   To assess the stability of the learnt representations, Tsitsulin et al. (2023) consider the pseudo-condition number. In numerical linear algebra, the condition number of the matrix $X$ is defined by

$$\kappa(X) = \frac{\sigma_1}{\sigma_n},$$

where $\sigma_1$ and $\sigma_n$ are the largest and smallest singular values of $X$, respectively (excluding any zero singular values for stability). Intuitively, a large $\kappa$ indicates that $X$ is close to being rank-deficient, small perturbations in the data or labels can produce large changes in the solution of $WX = Y$. Conversely, a small $\kappa$ indicates that the representation is well-conditioned and the corresponding linear system is less sensitive to perturbations.

**Note:** $\kappa$ directly affects the outcome of certain solvers (especially closed-form OLS solutions) and is therefore only an indirect measure of representation quality. In particular, a high $\kappa$ can degrade an OLS solution even if the representation itself is informative. We find that $\kappa$'s correlation with our linear probe accuracy vanishes when switching to a $k$-NN probe, confirming that its apparent predictive power was largely due to the choice of the OLS solver rather than the representation itself (see Section 4.2).

## 3.2   Relational Metrics

We refer to metrics that assess representation quality through pairwise relationships between samples as *relational metrics*. Rather than using the covariance structure of the representation matrix, these metrics characterise the geometry of the representation space through pairwise interactions among samples. This is typically done by constructing an $N \times N$ relational matrix. These metrics differ in how they quantify pairwise relationships among samples. We restate their construction and, where possible, reformulate them to improve memory efficiency.

**Self-clustering metric**   Tsitsulin et al. (2023) introduce this metric on the basis of the observation that contrastive learning algorithms tend to distribute representations uniformly in the representation space (Assran et al., 2023). The metric estimates how much the embeddings are clustered in the embedding space compared to a random distribution on the sphere. Let $\tilde{X} \in \mathbb{R}^{N \times d}$ denote the $\ell_2$-normalised representations of $X$ (applied to each row of $X$). Assuming that the representations are uniformly distributed on the unit hypersphere, the expected and maximum values of $\|\tilde{X}\tilde{X}^\top\|_F$ are $\mathbb{E}[\|\tilde{X}\tilde{X}^\top\|_F] = N + \frac{N(N-1)}{d}$ and $\max_{\tilde{X}} \|\tilde{X}\tilde{X}^\top\|_F = N^2$ (Tsitsulin et al., 2023), respectively, which gives:

$$\text{Self-Cluster}(\tilde{X}) = \frac{\|\tilde{X}\tilde{X}^\top\|_F - \mathbb{E}[\|\tilde{X}\tilde{X}^\top\|_F]}{\max_{\tilde{X}}(\|\tilde{X}\tilde{X}^\top\|_F) - \mathbb{E}[\|\tilde{X}\tilde{X}^\top\|_F]}$$

$$= \frac{\|\tilde{X}\tilde{X}^\top\|_F - (N + \frac{N(N-1)}{d})}{N^2 - (N + \frac{N(N-1)}{d})}$$

$$= \frac{d\|\tilde{X}\tilde{X}^\top\|_F - N(d + N - 1)}{(d-1)(N-1)N}.$$

Self-Cluster attains the maximum (Self-Cluster$(\tilde{X}) = 1$) when the representations collapse to a single point and is minimum (Self-Cluster$(\tilde{X}) = 0$) when the representations are distributed uniformly on the hypersphere. Alternatively, Wang & Isola (2020) show that the uniform distribution is the maximal entropy configuration on the sphere, a uniformly distributed representation contains maximal information. They further show that their uniformity metric agrees strongly with downstream task performance. Thus, one can expect that a lower Self-Cluster value corresponds to higher downstream accuracy.

Since $N \gg d$, in practice, computing $\tilde{X}\tilde{X}^\top \in \mathbb{R}^{N \times N}$ directly is infeasible for large datasets. Consequently, Tsitsulin et al. (2023) omit this computation for ImageNet-scale datasets. We note that $|\tilde{X}\tilde{X}^\top|_F = |\tilde{X}^\top\tilde{X}|_F$ holds, and instead compute $\tilde{X}^\top\tilde{X} \in \mathbb{R}^{d \times d}$. This reformulation allows $\tilde{X}$ to be partitioned into batches $\{\tilde{X}_{b,i}\}$ and compute $\tilde{X}^\top\tilde{X}$ via aggregation, i.e., $\sum_i \tilde{X}_{b,i}^\top \tilde{X}_{b,i}$, thereby reducing the memory complexity from $\mathcal{O}(N^2)$ to $\mathcal{O}(d^2)$.

**Diffusion Spectral Entropy (DSE)**   Liao et al. (2024) compute the spectral entropy of a diffusion matrix constructed from pairwise similarity measures. The implementation uses an anisotropic Gaussian kernel as follows:

$$G_{ij} = \frac{1}{\sigma\sqrt{2\pi}} \exp\left(-\frac{\|x_i - x_j\|^2}{2\sigma^2}\right),$$

where $x_i$ denotes the representation, i.e., the $i$-th row of $X$. $G$ is then symmetrically normalised to remove the effect of local sampling density and capture only the intrinsic geometry, yielding $K$, which is then used to compute the DSE,

$$K = DGD, \quad D = \text{diag}(1/\sqrt{\sum_k G_{ik}}),$$

$$\text{DSE} = -\sum_i p_i^t \log p_i^t, \quad p_i = \frac{\hat{\lambda}_i}{\sum_j \hat{\lambda}_j},$$

where $\hat{\lambda}_i$ are the eigenvalues of $K$ and $t$ is the diffusion time parameter, which we set to 1 in our experiments. For small $\sigma$, $G_{ij}$ decays rapidly with distance, producing a nearly diagonal $K$ regardless of global geometry; thus, the metric loses sensitivity to global structure. For large $\sigma$, all entries of $G$ saturate towards a constant, again collapsing the eigenspectrum and making the metric uninformative. High DSE indicates a uniform eigenspectrum corresponding to representations with a rich, distributed structure; low DSE indicates a few dominant eigenvalues, corresponding to representations with a simple or highly concentrated structure. A richer, more distributed geometry should therefore expose more usable structure to a linear probe, and Liao et al. (2024) report that diffusion spectral entropy predicts downstream classification accuracy across a large set of models. Since this metric involves pairwise evaluations, we compute the metric for each batch and use the mean over batch as the final metric value. Appendix B reports the relative standard deviation of DSE across batches for all 260 models and 10 datasets; the median is 0.8% with a 95th percentile of 2.2%, confirming that the batched estimator is stable. We exclude the Diffusion Mutual Information (DMI) metric introduced alongside DSE, as it requires label information.

For $\ell_2$-normalised representations with sufficiently large $\sigma$, the Gaussian kernel may be approximated as (see Appendix A):

$$G \approx C\left(\mathbf{1}_{N \times N} + \frac{1}{\sigma^2}\tilde{X}\tilde{X}^\top\right).$$

After anisotropic normalisation, $K$ depends on $\tilde{X}\tilde{X}^\top$, up to scaling by the degree matrix. Since Self-Cluster is derived from $\|\tilde{X}\tilde{X}^\top\|_F$, both DSE and Self-Cluster depend on $\tilde{X}\tilde{X}^\top$. A clustered representation space produces a few dominant eigenvalues in $\tilde{X}\tilde{X}^\top$, which increases $\|\tilde{X}\tilde{X}^\top\|_F$ (i.e., Self-Cluster increases) while reducing spectral entropy (DSE decreases), and vice versa for a uniform eigenspectrum. We empirically confirm this negative correlation in Section 4.2.

### 3.3 Manifold Metrics

We refer to metrics that assess representation quality through the geometry of the representation manifold as *manifold metrics*. These methods are motivated by the observation that learned representations often lie on (or near) a lower-dimensional manifold embedded in the ambient feature space. Manifold-based metrics aim to capture geometric properties such as dimensionality and the local structure of the underlying manifold.

**Intrinsic dimension (ID)** Ansuini et al. (2019) propose a metric for estimating the minimal number of parameters required to describe the representations. It is based on the observation that the ratio of distances to the first and second nearest neighbours follows a known distribution under specific assumptions. Concretely, they employ the TwoNN estimator of Facco et al. (2017). For each point, let $r_1(i)$ and $r_2(i)$ denote the distances to its first and second nearest neighbours, and define $\mu_i = r_2(i)/r_1(i)$. Then, $\mu_i$ follows a Pareto distribution with parameter $d+1$ on $[1, \infty)$, $f(\mu_i \mid d) = d\,\mu_i^{-(d+1)}$. For $N$ points, the likelihood of $\boldsymbol{\mu} = (\mu_1, \mu_2, \ldots, \mu_N)$

$$P(\boldsymbol{\mu} \mid d) = d^N \prod_i \mu_i^{-(d+1)},$$

so the intrinsic dimension $d$ can be recovered as the maximum-likelihood estimate. This estimator captures the dimensionality of a curved, non-uniformly sampled manifold rather than a linear subspace. Lower ID suggests that the learnt representation lies on a lower-dimensional manifold, which the authors show is associated with better generalisation performance. In particular, the ID of the last hidden layer predicts test accuracy. Lu et al. (2023) studied the **Cluster Learnability and Intrinsic Dimension (CLID)**, which can be thought of as an extension of ID. CLID combines the Intrinsic dimension and clusterability (KNN performance, which requires label information). We ignore this in our evaluation as it requires labels. Following Ansuini et al. (2019), we compute ID per batch by drawing 20 independent random subsamples of 90% of the batch size. We apply the TwoNN estimator to each subsample, and take the mean as the per-batch ID estimate. We take the mean of IDs computed for each batch as the final ID. Appendix B reports the distribution of intra-batch and across-batch relative standard deviations across all 260 models and 10 datasets (2,600 pairs). The intra-batch noise has a median of 4.4% and the across-batch noise has a median of 7.6% .

## 4 Comparative Analysis

The metrics in Section 3 have been proposed as proxies for downstream performance while operating only on the feature geometry without requiring label information. However, it remains unclear whether these metrics consistently and reliably predict downstream task performance across a wide range of configurations. In particular, while several spectral metrics are constructed from the same underlying eigenspectrum, it is unclear how they differ from one another when the underlying spectral shape and representation dimensionality remain the same. In this section, (i) we decouple confounders arising from the choice of backbone model and datasets by considering a simulated eigenspectrum and analyse how spectral metrics respond to changes in spectral shape and representation dimensionality (4.1), and (ii) we assess how well all the metrics correlate with OLS fit accuracy across a diverse set of backbone models, datasets, and training objectives (4.2).

### 4.1 Sensitivity of Spectral Metrics

The eigenspectrum of the representation covariance matrix has been shown to exhibit an approximately power-law decay, with a small number of dominant directions that capture most of the variance (Nassar et al., 2020; Xie et al., 2022). A power law of the form of $\lambda_i \propto i^{-\alpha}$ provides a natural parametric family to characterise this decay. In addition, (Agrawal et al., 2022) used this formulation to study representation quality. Thus, we synthetically generate eigenspectra under the power law assumption, setting $\sum_{i=1}^{d} \lambda_i = 1$ without loss of generality, since all three metrics are scale-invariant with respect to the eigenvalues. While Agrawal et al. (2022) restrict their attention to $\alpha > 0$, we additionally consider $\alpha < 0$ to examine the spectral shape. For a consistent basis of comparison, we sort the eigenvalues in decreasing order. As $\alpha$

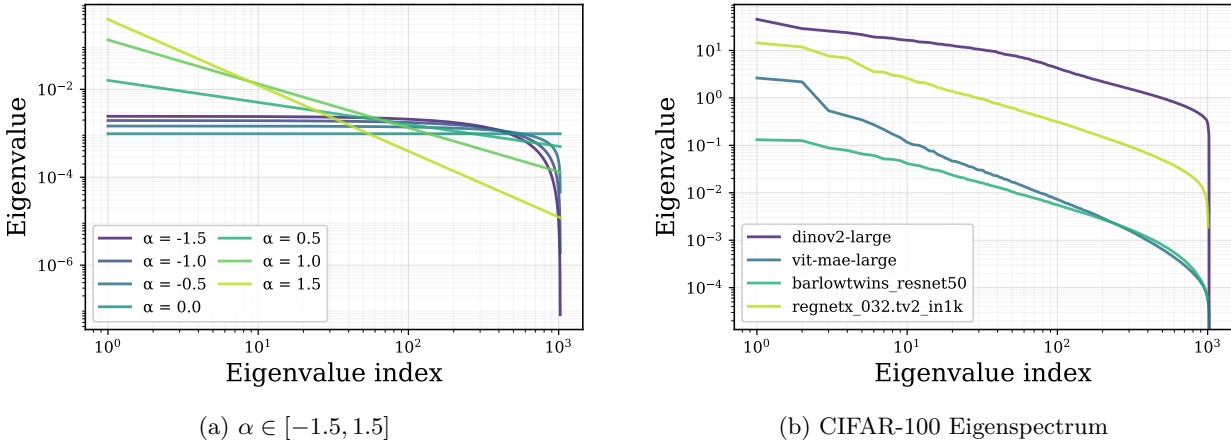

(a) $\alpha \in [-1.5, 1.5]$          (b) CIFAR-100 Eigenspectrum

Figure 2: Comparison of real and synthetic eigenspectra in log-log scale. Real backbone representations with dimension $d \in [1000, 1024]$ (Fig. 2b) exhibit spectral shapes consistent with the synthetically generated power-law family $\lambda_i \propto i^{-\alpha}$, $\sum_{i=1}^{d} \lambda_i = 1$, under negative values of $\alpha$ (Fig. 2a). This similarity justifies including $\alpha < 0$ in the synthetic sensitivity analysis

becomes increasingly negative, most of the eigenspectrum is largely flat, followed by a sharp decline (see the curves corresponding to $\alpha < 0$ in Figure 2a). We note that this closely resembles the eigenspectra observed in real representations (Figure 2b), justifying including $\alpha < 0$ in our analysis. We vary $\alpha \in [-2, 2]$ and $d \in [100, 5000]$, where the latter range spans the representation dimensions of the model families considered in Section 4.2.

We first examine the behaviour of these metrics across this parameter space. Figures 3a, 3d show that both RankMe and NE Sum are maximised at $\alpha = 0$ for all values of $d$. This is consistent with their construction, as RankMe measures spectral entropy which is maximised when the spectrum is uniform ($\alpha = 0$ generates a uniform eigenspectrum as observed in Figure 2a) and NE Sum, by construction measures features whitening which is synonymous with $\alpha = 0$. However, these metrics are asymmetric about $\alpha = 0$, which is explained by the fact that the spectrum is nearly flat for $\alpha < 0$, compared to the rapid decay for $\alpha > 0$. On the other hand, $\kappa$ is symmetric by construction, as it depends only on the ratio of the largest to the smallest singular value, which is identical for $\pm \alpha$ under the power law after sorting. $\kappa$ attains its maximum at the extremities and is unbounded (Figure 3g); as $|\alpha|$ increases, the smallest eigenvalue becomes vanishingly small. Both RankMe and NE Sum scale with $d$, while $\kappa$ displays similar behaviour, but this effect is more pronounced as $|\alpha|$ increases. To aid comparison between models with different $d$, we normalise the metrics by dividing by $d$.

The normalised surfaces (Figures 3b, 3e) still reveal qualitative asymmetry around $\alpha = 0$. The contour lines in Figures 3c and 3f reveal that for the same value of $\alpha$, RankMe$^\star$ consistently show higher values than NE Sum$^\star$ in both regimes. For $\alpha > 0$, both metrics decay with increasing $d$ even after normalisation. This indicates sub-linear scaling in $d$ in this regime. For $\alpha \leq 0$, both RankMe$^\star$ and NE Sum$^\star$ are insensitive to changes in $d$, consistent with linear scaling in that regime. NE Sum$^\star$ exhibits steeper variation near $\alpha = 0$ than RankMe$^\star$, as seen from the more tightly spaced contour lines in that region (Figures 3c and 3f). Since $\kappa$ is symmetric about $\alpha = 0$, we consider only positive values of $\alpha$ in Figure 3i. On the other hand, $\kappa^\star$ is minimised near $\alpha = 0$ and grows as $|\alpha|$ increases (Figure 3i).

These results reveal that the three metrics exhibit different sensitivities to the shape of eigenspectra. NE Sum$^\star$ exhibits high sensitivity to $\alpha$ around $\alpha = 0$ and RankMe$^\star$ exhibits low sensitivity in the same region. On the other hand, $\kappa^\star$ exhibits complementary behaviour by being most sensitive at the spectral extremes and largely uninformative near $\alpha = 0$. This suggests that no single spectral metric provides uniformly consistent sensitivity across the full range of spectral shapes encountered in practice, and care is needed when interpreting these metric values.

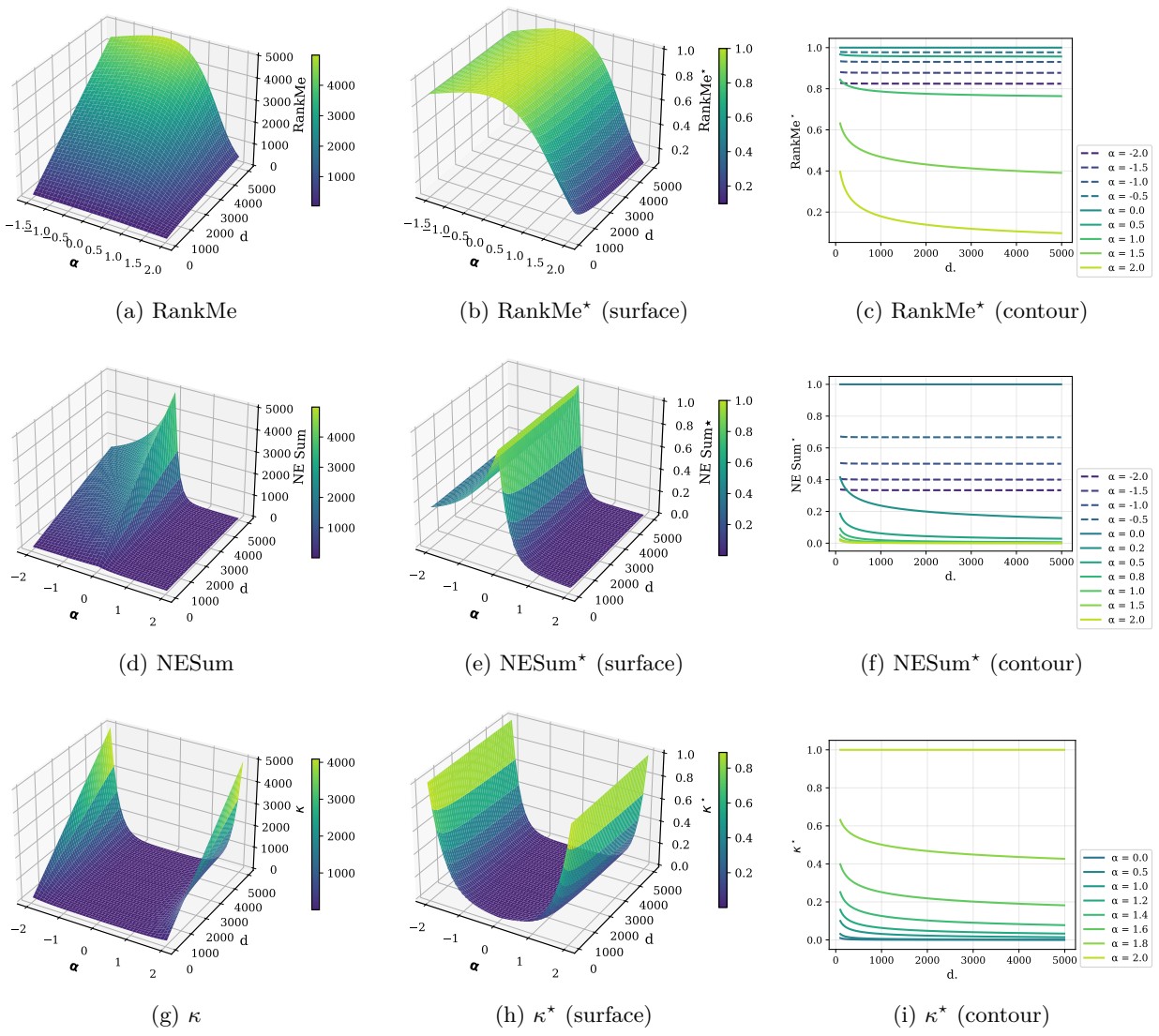

Figure 3: Sensitivity of spectral metrics RankMe, NE Sum, and $\kappa$ to spectral shape ($\alpha$) and representation dimension ($d$), where eigenspectra are synthetically generated as $\lambda_i \propto i^{-\alpha}$ with $\sum_{i=1}^{d} \lambda_i = 1$. Each row shows the un-normalised metric (col. 1), the dimension-normalised surface (col. 2), and the normalised metric as a function of $d$ for fixed values of $\alpha$ (col. 3). Normalisation is performed by dividing by $d$, following the normalisation proposed in Tsitsulin et al. (2023) for RankMe, which we extend to all three metrics. In col. 3, dashed lines correspond to $\alpha < 0$ and solid lines to $\alpha \geq 0$. Since $\kappa$ is symmetric about $\alpha = 0$ (Fig. 3h), only $\alpha \geq 0$ is shown in Fig. 3i.

## 4.2  Correlation of Metrics with Test Accuracy

The synthetic analysis in the previous section, by construction, decoupled the effects of architecture, training objective, and dataset from spectral shape. However, these factors also influence the geometry of real-world representations. In this subsection, we assess how well the metrics listed in Table 1 correlate with downstream performance by grouping the backbone models according to architecture class and training objective.

**Experimental Setup**  We consider (i) supervised models and (ii) Self-Supervised models. In total, we consider 260 backbone models for our analysis (details of the models, training data, and training methods

are provided in the Appendix C). Across all experiments, we extract features from the final layer of the backbone. We consider Vision Transformers (ViT) (Dosovitskiy et al., 2021), Residual Networks (He et al., 2016), ConvNeXT (Liu et al., 2022), and RegNet (Radosavovic et al. (2020)) model architectures. Following standard practice, we use the [CLS] token as the representation for ViT models, and for CNN-based architectures (ResNet, ConvNeXT, RegNet), we apply global average pooling over the final convolutional feature maps (He et al., 2016).

We evaluate on six image classification benchmarks spanning generic object classification: CIFAR-100 (Krizhevsky, 2009) and ImageNet-1k (Deng et al., 2009); fine-grained object classification: Flowers-102 (Nilsback & Zisserman, 2008) and Food-101 (Bossard et al., 2014); and specialised domains: Places-365 (Zhou et al., 2018) and EuroSAT (Helber et al., 2019). ImageNet-1k results should be interpreted with caution, as it appears in the training pipeline of most backbones. Results on a broader set of ten datasets, including fine-grained and remote-sensing benchmarks, are reported in Appendix D.

To enable faster and more stable evaluation, we fit a closed-form regularised least-squares classifier: $W = (X^\top X + \lambda I)^{-1} X^\top Y$, where $X$ are the $\ell_2$-normalised backbone features, $Y$ the one-hot encoded targets, and the predicted class is $\arg\max XW$. We use $\lambda = 10^{-3}$ across all models and datasets. As a complement to the OLS linear probe, we report $k$-nearest-neighbour ($k$-NN) accuracy with $k = 1$, following Balestriero et al. (2023). The full training split serves as the retrieval index; each test point is assigned the plurality label of its $k$ nearest neighbours under cosine similarity, requiring no learned parameters.

We use Spearman's correlation between the metrics considered and the test accuracy for our comparative analysis. Spearman's correlation is preferred over Pearson's as it does not assume a linear relationship between metric and accuracy. To account for the non-independence of the model pool, we compute Spearman rank correlations ($\rho$) using a cluster-robust bootstrap procedure with 1,000 resamples. Models are grouped into clusters based on their base architecture, representation dimension, and training objective. The resampling is performed at the cluster level with replacement to derive standard errors and $p$-values that account for within-cluster dependence. To correct for multiple comparisons, we apply Benjamini–Hochberg FDR correction (Benjamini & Hochberg, 1995) jointly across all metric–dataset cells in a given analysis table, including the datasets reported in Appendix D.

**Observation**   In Table 2a, ID is the only reliable predictor for all datasets, showing negative correlation with test accuracy. It shows the strongest correlation for the generic object classification tasks ($\rho \in [-0.63, -0.60]$), mild to strong correlation on fine-grained object classification tasks ($\rho \in [-0.76, -0.37]$), and weaker correlation on the scene-recognition (Places-365, $\rho = -0.18$) and remote-sensing (EuroSAT, $\rho = -0.21$) datasets. This observation shows that ID as a metric is most reliable for tasks requiring object classification, and is a comparatively weaker estimator for scene-recognition and remote-sensing tasks (see Figure 4). Table 2b removes any reservations about results being influenced by choice of probe as a similar trend is observed using KNN probe.

The relational metrics (Self-Cluster and DSE) display mild but statistically significant correlation on fine-grained object classification tasks and on the Places-365 task (Table 2a), and are insignificant on the natural and EuroSAT datasets. However, with the KNN probe they show moderate to strong correlation across all six datasets. This is expected because the KNN probe operates on the nearest-neighbour principle and relational metrics capture this behaviour. In Section 3.2, we illustrated the relationship between these two metrics and how they capture the same phenomenon but in opposite directions. Figure 5 illustrates this through an almost mirror-image relationship for these two metrics in all datasets. The Spearman correlation between the two metrics ranges from $-0.999$ to $-0.998$ (consistent between probes), across all six datasets, provide a quantitative justification for this. This is expected since DSE measures the entropy of the diffusion process over the representation graph, which is maximised when representations are spread uniformly. On the other hand, Self-Cluster explicitly measures deviation from a uniform spherical distribution.

The OLS correlation results indicate that $\kappa$ shows a statistically significant, mild-to-moderate positive correlation in CIFAR-100, Food-101, and Places-365 and EuroSAT, with the strongest correlation observed on EuroSAT ($\rho = 0.55$) (see Table 2a). Since $\kappa$ has been shown to directly govern the conditioning of the OLS solution (Belsley et al., 1980), most of these correlations do not survive under the KNN probe (Table 2b).

Table 2: Spearman correlation ($\rho$) between representation quality metrics and downstream accuracy across six evaluation datasets. Metrics are organised by construction: spectral (top), relation-based (middle), and manifold-based (bottom). Statistical significance is considered after Benjamini–Hochberg correction. * $p < 0.05$, ** $p < 0.01$, *** $p < 0.001$. Non-significant cells are greyed out. – are cells with either no models or one model

(a) OLS linear probe

|  | CIFAR-100 | ImageNet‡ | Food-101 | Flowers-102 | Places-365 | EuroSAT |
|---|---|---|---|---|---|---|
| $\alpha < 1.0$ | −0.5381** | −0.347** | – | – | −0.3235** | – |
| $\alpha \geq 1.0$ | 0.06995 | 0.1688 | -0.03956 | 0.3184** | 0.062 | 0.2965** |
| $\alpha$-ReQ | 0.03393 | -0.01968 | -0.1169 | -0.02593 | 0.3184** | −0.1824* |
| RankMe | 0.2443* | 0.2166* | 0.3823** | 0.0921 | 0.4602** | 0.4242** |
| RankMe* | -0.1362 | -0.05487 | -0.09887 | -0.1966 | 0.006721 | −0.548** |
| NE-Sum | 0.2309* | 0.1048 | 0.4116** | 0.4107** | 0.3007** | −0.2561** |
| $\kappa$ | 0.3331** | 0.1599 | 0.2716* | 0.1459 | 0.3707** | 0.5456** |
| Self-Cluster | -0.1803 | -0.09359 | −0.3015** | −0.2602* | −0.2692** | 0.0856 |
| DSE | 0.1967 | 0.1096 | 0.3099** | 0.2643* | 0.2872** | -0.09127 |
| ID | −0.5981** | −0.6254** | −0.3683** | −0.7569** | −0.1815* | −0.2128* |

(b) KNN probe ($k = 1$)

|  | CIFAR-100 | ImageNet‡ | Food-101 | Flowers-102 | Places-365 | EuroSAT |
|---|---|---|---|---|---|---|
| $\alpha < 1.0$ | −0.5769** | −0.4409** | – | – | −0.3284** | – |
| $\alpha \geq 1.0$ | −0.2066** | 0.2081 | -0.07289 | 0.1727* | −0.2072** | -0.04101 |
| $\alpha$-ReQ | −0.1678* | −0.2962** | −0.3546** | -0.06906 | 0.1727* | −0.4247** |
| RankMe | 0.3398** | 0.3726** | 0.4698** | 0.3069** | 0.5385** | 0.5585** |
| RankMe* | 0.1439 | 0.1961* | -0.05627 | −0.2627** | 0.2843** | −0.2179* |
| NE-Sum | 0.4971** | 0.3962** | 0.5801** | 0.6028** | 0.539** | 0.03467 |
| $\kappa$ | 0.04084 | -0.03046 | 0.02121 | 0.1905* | 0.03862 | 0.3332** |
| Self-Cluster | −0.5331** | −0.4385** | −0.5826** | −0.5452** | −0.5969** | −0.3621** |
| DSE | 0.5487** | 0.4505** | 0.5921** | 0.5533** | 0.6109** | 0.3484** |
| ID | −0.5502** | −0.6739** | −0.3688** | −0.6896** | −0.2109** | −0.2516** |

‡ Correlations of ImageNet-1k should be interpreted with caution as it is used in the training pipeline of most backbones

Other spectral metrics, like RankMe and NE Sum, show mild to moderate correlation on the OLS probe ($\rho \in [0.22, 0.46]$), but the correlation strengthens with KNN probe Table 2b). As argued in Section 3.1, $\alpha$-ReQ is unstable and requires careful tuning to determine which range the line is fit to. Unsurprisingly the correlation scores are insignificant for almost all datasets.

Overall, the mild to moderate levels of correlation suggest that these metrics are insufficient as stand-alone indicators of downstream performance. Since architecture class and training objective jointly influence representation geometry, we conduct a more fine-grained analysis by grouping models accordingly. However, we drop $\alpha$-ReQ and $\kappa$ from future analysis. $\alpha$-ReQ is unstable, and $\kappa$'s correlation with downstream accuracy is an artefact of the OLS solver's numerical conditioning than as a measure of representation quality. We also consider only Self-Cluster from the relational metrics, since it is highly correlated with DSE as analytically shown in Section 3.2 and empirically observed. We utilise the results from the OLS probe for further analysis, since the relational metrics (Self-Cluster and DSE) are themselves pairwise-similarity based, which is confounded by the KNN probe (magnitude nearly triples).

**Model architectures' impact on metrics** Grouping models by architecture reveals that metric reliability varies considerably across architecture classes. In total, we consider 234 models for this analysis

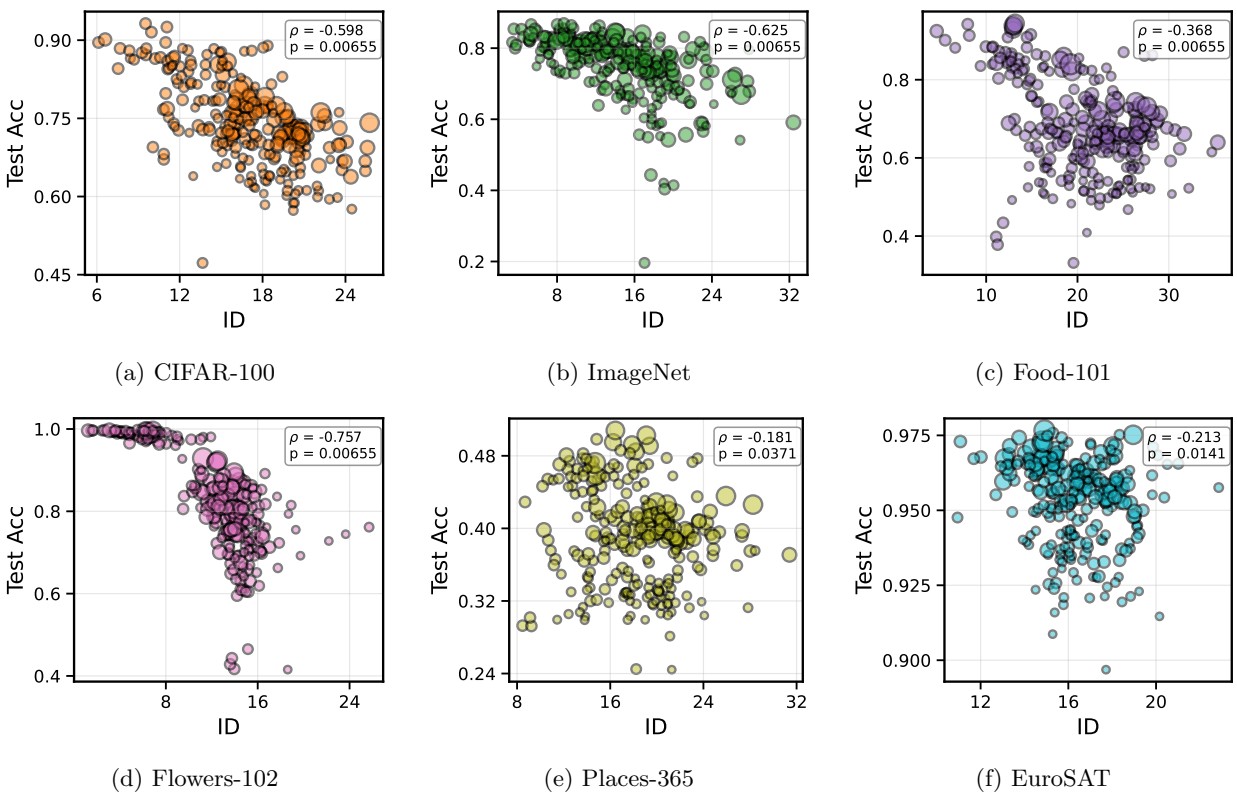

Figure 4: Comparison of ID against OLS probe test accuracy across six datasets. The $\rho$ denotes the Spearman correlation and the corresponding $p$-value is annotated in each subplot. Marker size is proportional to each model's representation dimensionality, scaled linearly relative to the highest-dimensional model.

(ConvNeXT: 52, RegNet: 63, ResNet: 64, ViT: 55 models). Table 3 shows that RankMe is a strong positive predictor at the architecture level, with significant correlations in all but one of the twenty-four cells (the exception being ViT on ImageNet), in contrast to the mild correlation it exhibits in the aggregate (Table 2a). This indicates that architecture class is a significant confounding factor in the aggregate results. Thus, models from different architecture classes should not be ranked jointly using these metrics.

ID remains a reliable predictor on the generic object classification datasets (CIFAR-100, ImageNet) and on Flowers-102, but its reliability declines on Food-101 and on the specialised datasets, becoming significant in only one of four architecture classes on EuroSAT. ID shows insignificant correlation for ResNet on four of the six datasets (CIFAR-100, Food-101, Places-365, EuroSAT), with significant correlations retained only on ImageNet and Flowers-102.

Self-Cluster is reliable for ResNet on five of the six datasets, the sole exception being EuroSAT, and is insignificant for ViT across all datasets. The insignificant correlations for Self-Cluster for ViT across all datasets can be attributed to the points that have Self-Cluster $\sim$ **1** across all datasets as observed in Figure 5. These collapsed values originate predominantly in ViT models, as illustrated in the top row of Figure 6. Furthermore, the bottom row of Figure 6 reveals that within ViT models, the near-unit Self-Cluster values arise predominantly from supervised and SSL+Sup models rather than SSL models, consistent with the finding in Section 4.2 that Self-Cluster is unreliable for supervised models, with the same pattern observed for DSE (Appendix E).

**Influence of training objective on the metrics** Another factor that influences how representations relate to each other is the training objective. We consider two main groups: models trained in a supervised fashion (Supervised, $n = 209$) and models trained with SSL objectives (SSL, $n = 32$).

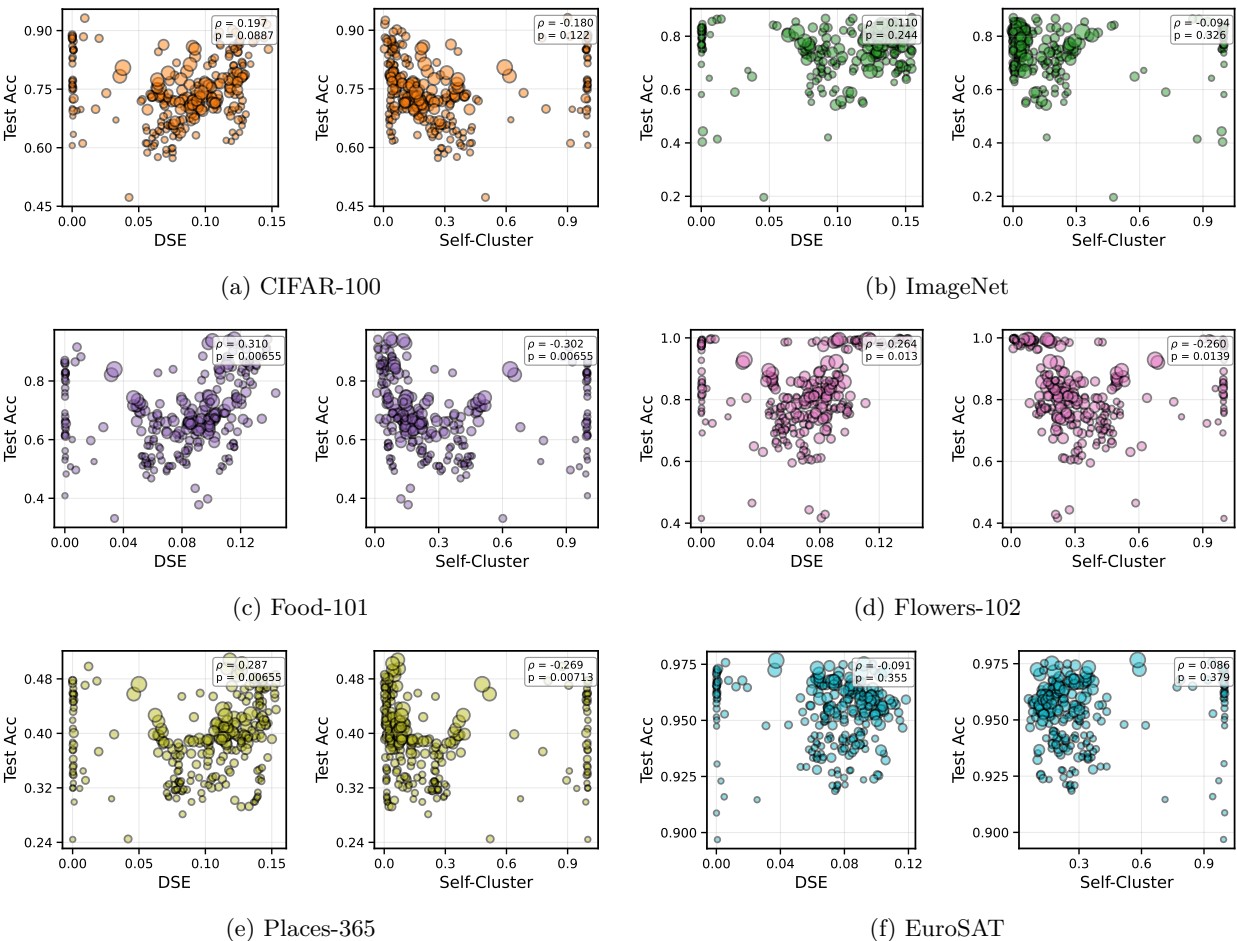

Figure 5: Comparison of DSE and Self-Cluster against OLS linear-probe test accuracy across six datasets. The $\rho$ denotes the Spearman correlation and the corresponding $p$-value is annotated in each subplot. Marker size is proportional to each model's representation dimensionality, scaled linearly relative to the highest-dimensional model.

From Table 4, ID is a significant predictor for SSL models across all five object-classification datasets ($\rho \in [-0.88, -0.46]$), the only exception being the EuroSAT dataset. For supervised models, ID is significant only on the generic object classification datasets and Flowers-102 (three of six). Notably, ID is therefore at least as reliable for SSL models as for supervised models, extending the original supervised-only analysis of Ansuini et al. (2019) to self-supervised representations.

Self-Cluster shows strong, significant negative correlations for SSL models across the same five object datasets ($\rho \in [-0.91, -0.63]$) but is insignificant for every supervised group. This is consistent with the broader argument that representations approaching maximal entropy on the hypersphere correlate with downstream performance (Wang & Isola, 2020).

Of the spectral metrics, RankMe shows a mild to moderate positive correlation on five of six datasets on the supervised models, with Flowers-102 being the only exception, and is reliable only on Places-365 and EuroSAT for the SSL objective. NE-Sum shows a reliable moderate to strong correlation for the SSL objective across all six datasets, but is significant for only half the datasets under the supervised objective (Food-101, Flowers-102, EuroSAT). This is consistent with He & Ozay (2022), who motivate NE-Sum specifically as a measure of feature whitening, a phenomenon associated with avoiding representation collapse in SSL training, which explains why the metric is informative for SSL.

Table 3: Spearman correlation ($\rho$) between representation metrics and OLS test accuracy, grouped by architecture class, across the six main datasets. Statistical significance is considered after Benjamini–Hochberg correction. $^{*}$ $p < 0.05$, $^{**}$ $p < 0.01$, $^{***}$ $p < 0.001$. Non-significant cells are greyed out.

| | | ConvNeXT | RegNet | ResNet | ViT |
|---|---|---|---|---|---|
| **CIFAR-100** | RankMe | $0.8189^{**}$ | $0.8952^{**}$ | $0.8425^{**}$ | $0.5985^{**}$ |
| | RankMe$^{\star}$ | 0.2661 | $-0.5361^{**}$ | -0.4239 | 0.1086 |
| | NE-Sum | -0.2015 | $0.4731^{**}$ | $0.5276^{**}$ | $0.4809^{**}$ |
| | Self-Cluster | 0.1316 | $-0.3331^{*}$ | $-0.7623^{**}$ | -0.143 |
| | ID | $-0.8087^{**}$ | $-0.5637^{**}$ | -0.06056 | $-0.5633^{**}$ |
| **ImageNet** | RankMe | $0.3897^{*}$ | $0.7538^{**}$ | $0.6047^{**}$ | 0.1835 |
| | RankMe$^{\star}$ | 0.1452 | 0.1151 | 0.136 | -0.1315 |
| | NE-Sum | -0.03091 | $0.5665^{**}$ | 0.2135 | 0.1395 |
| | Self-Cluster | $0.542^{**}$ | $-0.5665^{**}$ | $-0.5677^{**}$ | 0.1932 |
| | ID | $-0.4361^{*}$ | $-0.64^{**}$ | $-0.4489^{**}$ | $-0.5089^{**}$ |
| **Food-101** | RankMe | $0.8168^{**}$ | $0.874^{**}$ | $0.7219^{**}$ | $0.6275^{**}$ |
| | RankMe$^{\star}$ | 0.2391 | $-0.4341^{**}$ | $-0.4802^{*}$ | 0.08788 |
| | NE-Sum | $0.6755^{**}$ | $0.5504^{**}$ | $0.6168^{**}$ | $0.6099^{**}$ |
| | Self-Cluster | $-0.3117^{*}$ | -0.272 | $-0.6201^{**}$ | -0.2651 |
| | ID | $-0.7834^{**}$ | -0.2955 | 0.2845 | $-0.468^{**}$ |
| **Flowers-102** | RankMe | $0.3618^{**}$ | $0.703^{**}$ | $0.6503^{**}$ | $0.41^{**}$ |
| | RankMe$^{\star}$ | $-0.3568^{**}$ | $-0.8161^{**}$ | $-0.5788^{**}$ | -0.2895 |
| | NE-Sum | $0.8002^{**}$ | $0.3632^{*}$ | $0.3015^{*}$ | $0.5721^{**}$ |
| | Self-Cluster | $-0.6228^{**}$ | -0.09754 | $-0.5086^{**}$ | -0.3035 |
| | ID | $-0.9479^{**}$ | $-0.5794^{**}$ | $-0.6206^{**}$ | $-0.9146^{**}$ |
| **Places-365** | RankMe | $0.815^{**}$ | $0.9386^{**}$ | $0.8208^{**}$ | $0.6463^{**}$ |
| | RankMe$^{\star}$ | $0.3893^{**}$ | -0.2845 | -0.3035 | 0.1025 |
| | NE-Sum | 0.04653 | $0.6026^{**}$ | $0.4664^{**}$ | $0.4416^{**}$ |
| | Self-Cluster | -0.08768 | $-0.2782^{*}$ | $-0.7027^{**}$ | -0.1927 |
| | ID | -0.04636 | $-0.2563^{**}$ | 0.1671 | $-0.3843^{*}$ |
| **EuroSAT** | RankMe | $0.7766^{**}$ | $0.7987^{**}$ | $0.5526^{*}$ | $0.6053^{**}$ |
| | RankMe$^{\star}$ | -0.3913 | $-0.8258^{**}$ | $-0.7666^{**}$ | $-0.3928^{**}$ |
| | NE-Sum | $-0.6439^{**}$ | -0.1598 | 0.2322 | -0.04633 |
| | Self-Cluster | 0.02041 | 0.182 | -0.394 | -0.08389 |
| | ID | $-0.5166^{**}$ | -0.2524 | -0.08959 | -0.05596 |

From these observations, the training objective plays a strong role in determining the reliability of the spectral and relational metrics, whereas ID remains the most consistent metric across both SSL and supervised objectives on the object-classification datasets.

## 5 Discussion

Based on the taxonomy in Section 3 and the analysis in Section 4, we observe how current label-free metrics follow some notable patterns in behaviour, which we further discuss in this section, along with some of the important limitations of our study.

**Spectral metrics are generally not reliable**  $\alpha$-ReQ is an unstable metric since it is dependent on which range of the log-log eigenspectrum the line is fit to. Moreover, the hypothesis of Agrawal et al. (2022) that $\alpha \approx 1$ indicates high-quality representations does not generalise in our evaluation, as in the $\alpha < 1$ regime, $\alpha$-ReQ correlates significantly *negatively* with accuracy on CIFAR-100, ImageNet, and Places-365, directly contradicting the hypothesis. In the aggregate results from Table 2a, RankMe shows a weak-to-moderate correlation ($\rho \in [0.09, 0.46]$), but architecture-level grouping reveals near-uniformly significant correlations (23 of 24 cells, Table 3). We hypothesise that RankMe's entropy-based construction cannot

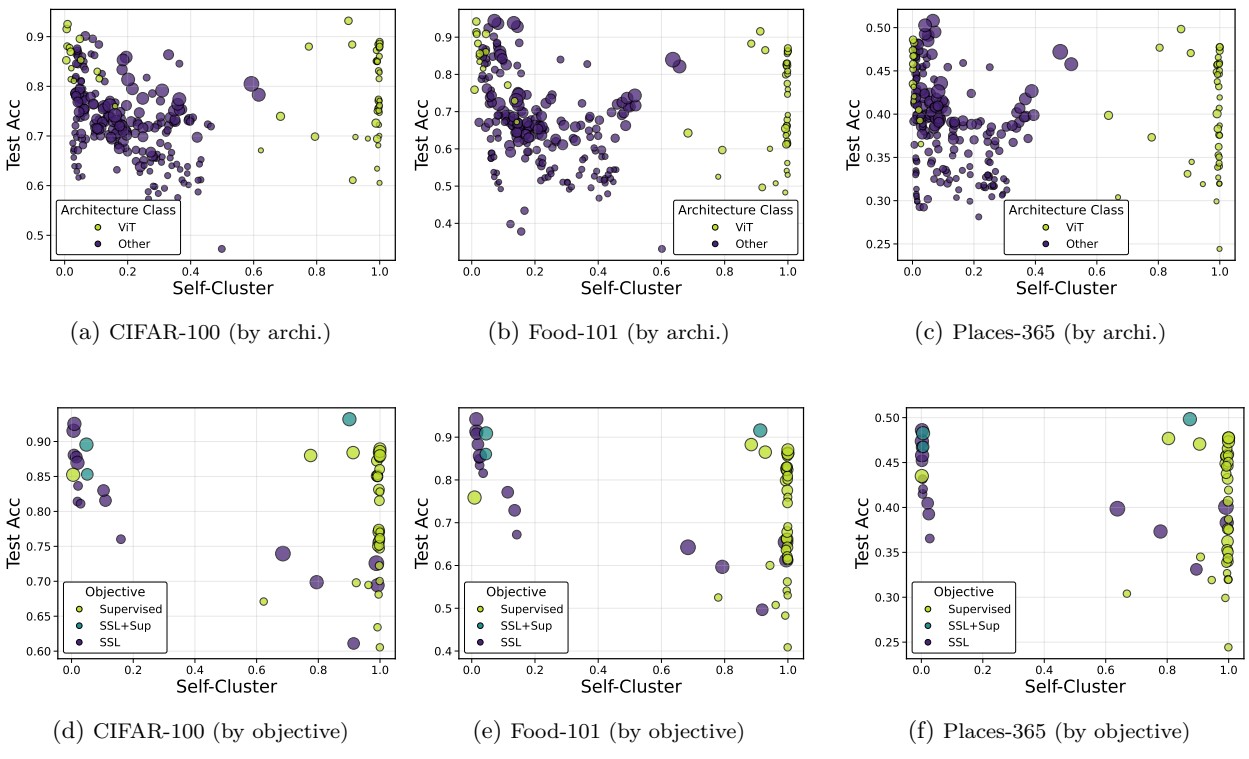

Figure 6: Self-Cluster metric values plotted against test accuracy across datasets. The top row shows all models coloured by architecture class, revealing that the models with extreme values predominantly belong to the ViT architecture class. The bottom row shows only ViT models coloured by training objective, illustrating that the near unit Self-Cluster values originate predominantly from Supervised and SSL+Sup models rather than SSL models. This suggests that the Self-Cluster metric may be a reliable predictor for SSL models (though the sample size is small), but is unreliable for Supervised and SSL+Sup models.

distinguish task-relevant variance from noise, so spreading variance across additional dimensions raises the metric regardless of whether those dimensions carry useful information. RankMe⋆ supports this concern, on most datasets, models span nearly the full accuracy range even as the normalised dimension approaches 1 (Figure 7). NE Sum is a reliable predictor specifically for SSL models (significant on all six datasets; Table 4), but is unreliable under the supervised objective. This is expected, as He & Ozay (2022) motivate NE Sum specifically as a measure of feature whitening, the degree to which eigenvalues are evenly distributed, a property associated with SSL objectives that encourage representations to avoid dimensional collapse; no equivalent whitening incentive exists under supervised cross-entropy training.

**Relation-based metrics are effective for SSL-based representations**  The near-perfect negative correlation between DSE and Self-Cluster ($\rho \approx -0.999$) confirms the theoretical prediction established in Section 3: under $\ell_2$-normalisation and the default $\sigma$, both metrics are functions of $\tilde{X}\tilde{X}^\top$. Self-Cluster is a strong predictor for SSL models (significant on all six datasets) but uninformative for supervised models (Table 4). This is consistent with contrastive and self-distillation objectives explicitly encouraging representations to spread uniformly over the hypersphere to avoid collapse (Assran et al., 2023), which is not an objective of supervised cross-entropy training. At the architecture level, Self-Cluster is most reliable for ResNet (5 of 6 datasets), partially reliable for ConvNeXT and RegNet (3 of 6 each), and uninformative for ViT (0 of 6): Self-Cluster values are near zero for ViT models trained with the SSL objective, and near one for ViT models trained with the supervised and SSL+Sup objectives, consistent with the SSL-uniformity motivation above. This suggests that these metrics are sensitive to interactions between architecture and training objective.

Table 4: Spearman correlation ($\rho$) between representation metrics and OLS test accuracy, grouped by training objective, across the six main datasets. Statistical significance is considered after Benjamini–Hochberg correction. $^{*}$ $p < 0.05$, $^{**}$ $p < 0.01$, $^{***}$ $p < 0.001$. Non-significant cells are greyed out.

| | | SSL | Supervised |
|---|---|---|---|
| CIFAR-100 | RankMe | 0.2258 | 0.3044$^{*}$ |
| | RankMe$^{*}$ | 0.6103$^{**}$ | $-0.3593^{**}$ |
| | NE-Sum | 0.6617$^{**}$ | 0.1204 |
| | Self-Cluster | $-0.8134^{**}$ | -0.02742 |
| | ID | $-0.6855^{**}$ | $-0.5586^{**}$ |
| ImageNet | RankMe | 0.331 | 0.2411$^{*}$ |
| | RankMe$^{*}$ | 0.5033$^{**}$ | -0.1564 |
| | NE-Sum | 0.4296$^{*}$ | 0.08246 |
| | Self-Cluster | $-0.7302^{**}$ | -0.1091 |
| | ID | $-0.4567^{**}$ | $-0.5948^{**}$ |
| Food-101 | RankMe | 0.1488 | 0.4572$^{**}$ |
| | RankMe$^{*}$ | 0.2804 | -0.1865 |
| | NE-Sum | 0.7474$^{**}$ | 0.3605$^{**}$ |
| | Self-Cluster | $-0.9062^{**}$ | -0.1227 |
| | ID | $-0.7265^{**}$ | -0.2311 |
| Flowers-102 | RankMe | -0.1755 | 0.1756 |
| | RankMe$^{*}$ | 0.01852 | -0.2528 |
| | NE-Sum | 0.887$^{**}$ | 0.2883$^{**}$ |
| | Self-Cluster | $-0.8921^{**}$ | -0.02264 |
| | ID | $-0.882^{**}$ | $-0.6634^{**}$ |
| Places-365 | RankMe | 0.4084$^{*}$ | 0.5174$^{**}$ |
| | RankMe$^{*}$ | 0.5587$^{**}$ | -0.1519 |
| | NE-Sum | 0.5161$^{**}$ | 0.2198 |
| | Self-Cluster | $-0.6294^{**}$ | -0.1524 |
| | ID | $-0.5183^{**}$ | -0.1054 |
| EuroSAT | RankMe | 0.6216$^{**}$ | 0.4761$^{**}$ |
| | RankMe$^{*}$ | -0.0009171 | $-0.5907^{**}$ |
| | NE-Sum | 0.37$^{*}$ | $-0.2768^{*}$ |
| | Self-Cluster | -0.246 | 0.0915 |
| | ID | 0.2788 | -0.1916 |

**Manifold-based metrics show the best overall generalisation** Ansuini et al. (2019) used ID to validate the claim that the intrinsic dimensionality of representations in convolutional models progressively decreases through the layers, and that lower ID at the final layer correlates with better generalisation. Tables 3 and 4 show that ID remains a reliable predictor across architecture classes and training objectives, with correlation scores in the range $\rho \in [-0.95, -0.26]$ indicating a moderate to strong negative relationship with test accuracy. ID's reliability does not depend on the choice of probe, its correlation magnitude is comparable under OLS probe and KNN probe (mean ratio 1.04, no change in significance across any of the six datasets) This is in contrast to the relational metrics, which roughly triple in magnitude under KNN (Section 4.2). Although their analysis focused on supervised models, Table 4 shows that ID is also a reliable predictor of performance on SSL models. However, ID's reliability is not uniform across tasks: it is strongest for datasets with clear, coarse object-category structure (CIFAR-100, ImageNet, Flowers-102), and weaker for Food-101 and the scene-recognition dataset Places-365. To assess whether ID's predictive ability is confounded by model capacity, we compute partial Spearman correlations controlling for log parameter count (Appendix B.4). ID remains a significant predictor on five of the six datasets after this correction, the only exception being Places-365, the dataset where ID weakest correlation.

Taken together, these observations suggest that neither the full utilisation of the representation space (RankMe, NE Sum) nor the degree to which representations are spread (Self-Cluster, DSE) is indicative

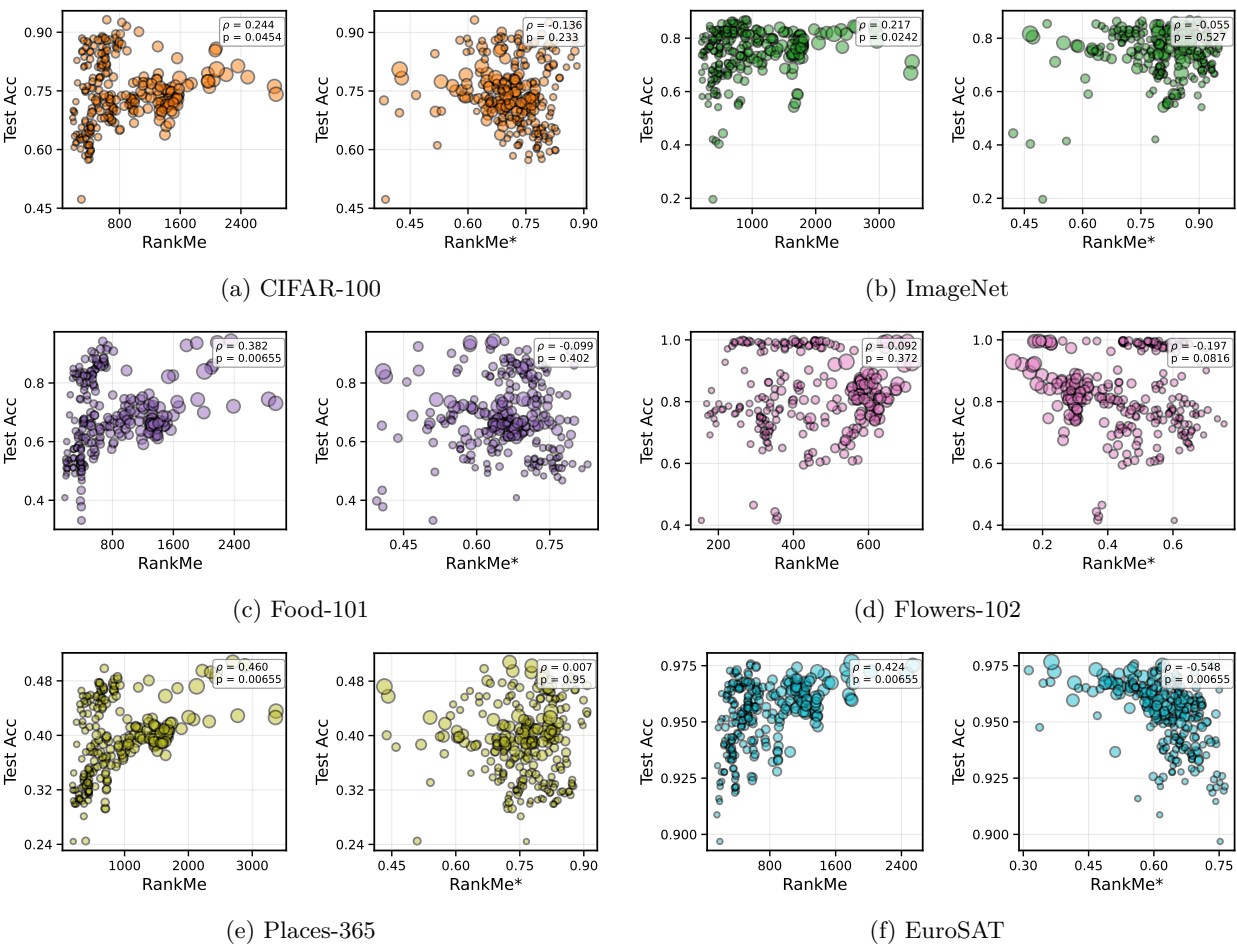

Figure 7: Comparison of RankMe and RankMe$^\star$ against OLS linear-probe test accuracy across six datasets. The $\rho$ denotes the Spearman correlation and the corresponding $p$-value is annotated in each subplot. Marker size is proportional to each model's representation dimensionality, scaled linearly relative to the highest-dimensional model.

of representation quality, except where the backbone is SSL-trained; rather, it is low intrinsic dimensionality of the representation manifold that is the most consistent indicator of representation quality in our evaluation.

**Limitations** We use an OLS fit to circumvent the long training time that linear probes take. Although this removes the stochasticity in test accuracies, converting a classification task into a regression task is a strong assumption. Also, OLS assesses only linear separability of representations, and thus the relationship between these metrics and performance with a non-linear head remains to be analysed. We consider only 19 SSL+Sup models, which makes conclusions about this group unreliable. A detailed study with a larger and more balanced model pool would be needed to better characterise their relationship with both SSL and supervised models. Finally, our experiments are conducted solely on vision models and image classification tasks, and their correlation with other modalities or task types needs to be explored.

# 6 Conclusion

Our work provides a systematic comparative evaluation of label-free representation quality metrics. We catalogue and group the metrics based on their construction into (i) spectral-based, (ii) pairwise relation-

based, and (iii) manifold-based families and reformulate them where possible to highlight similarities. We further investigate the sensitivity of spectral metrics to spectral shape and representation dimension. Our analysis of 260 models shows that the reliability of these metrics is strongly influenced by architecture class and training objective, which were obscured in the aggregate results (see Table 2a). Across most groupings, **ID** is the only metric that is consistently reliable across architecture classes and training objectives, for both SSL and supervised models, and is robust to the choice of evaluation probe (OLS vs. KNN). Its reliability degrades as tasks move away from coarse object-category classification, suggesting it tracks the separability of the representation manifold with respect to the target label structure. Spectral and relational metrics, by contrast, are reliable only within specific regimes. Since the different types of metrics capture different aspects of the representation space, their complementary nature could potentially be explored by combining different metrics, such as in CLID (Lu et al., 2023), and we consider this as an interesting direction for future work.

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

## Appendix

## A    Approximation of DSE

For $\ell_2$-normalised representations, $\|x_i - x_j\|^2 = 2 - 2x_i \cdot x_j$, so:

$$G_{ij} = \frac{1}{\sigma\sqrt{2\pi}} \exp\left(-\frac{1 - x_i \cdot x_j}{\sigma^2}\right) = C \cdot \exp\left(\frac{x_i \cdot x_j}{\sigma^2}\right)$$

where $C = \frac{1}{\sigma\sqrt{2\pi}} \exp\left(-\frac{1}{\sigma^2}\right)$ is a constant. We work with a default setting of $\sigma = 10$ for which the first-order Taylor expansion is $\exp\left(\frac{x_i \cdot x_j}{\sigma^2}\right) \approx 1 + \frac{x_i \cdot x_j}{\sigma^2}$, which is a good approximation, since $\frac{|x_i \cdot x_j|}{\sigma^2} \leq 0.01$ and $|x_i \cdot x_j| \leq 1$.

$$G \approx C\left(\mathbf{1}_{N \times N} + \frac{1}{\sigma^2}\tilde{X}\tilde{X}^\top\right)$$

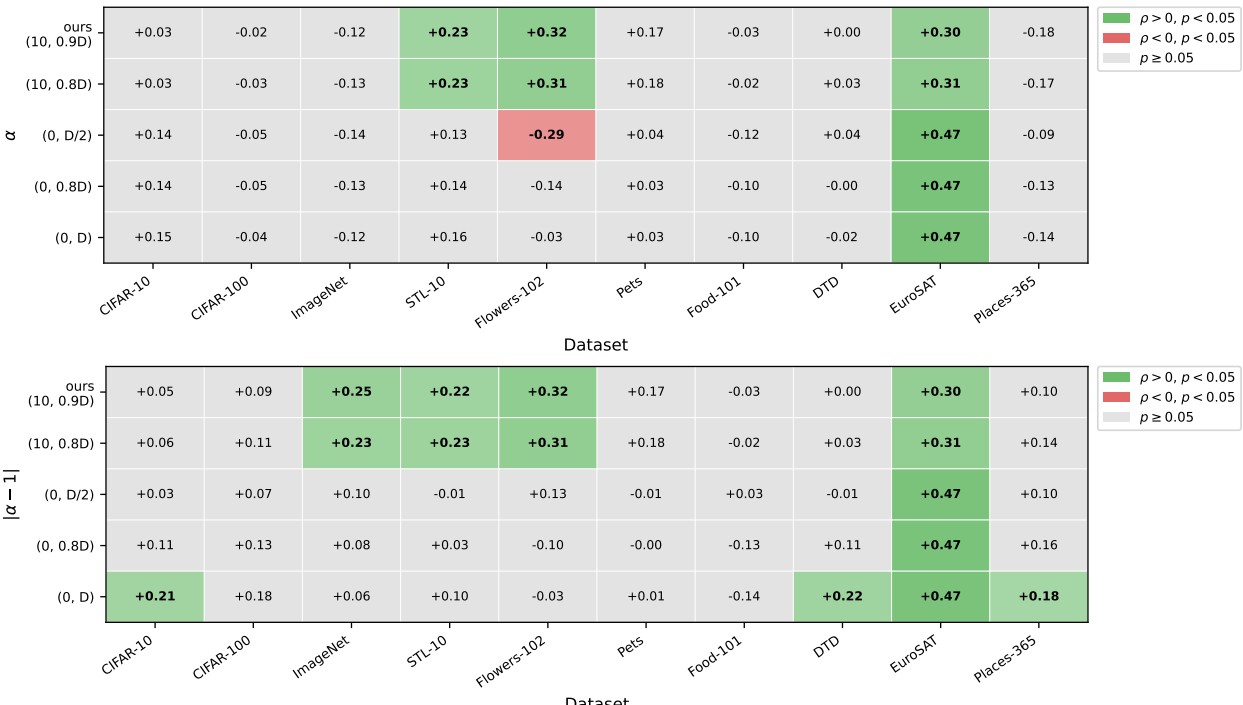

Figure 8: $\alpha$-**ReQ fitting-range sensitivity.** Spearman correlation between $\alpha$ (*top*) and $|\alpha-1|$ (*bottom*) with OLS test accuracy, across five fitting ranges and ten datasets. Green: significant positive; red: significant negative; grey: $p \geq 0.05$.

# B  Metric Estimation Stability

## B.1  $\alpha$-ReQ Fitting-Range Sensitivity

Figure 8 reports the Spearman correlation between $\alpha$ (left) and $|\alpha - 1|$ (right) with OLS test accuracy, computed across five fitting ranges: our paper euristic $(10, 0.9d)$, a narrower variant $(10, 0.8d)$, and three other ranges : $(0, d/2)$, $(0, 0.8d)$, and $(0, d)$. The two paper-adjacent ranges $(10, 0.9d)$ and $(10, 0.8d)$ yield near-identical results across all ten datasets, suggesting the tail truncation is not a material choice. However, including the dominant head eigenvalues produces substantial changes. On Flowers-102, the correlation with $\alpha$ flips sign from $+0.31$ (significant) under the paper range to $-0.29$ (significant, opposite direction) under the $(0, d/2)$ range. However, these ranges are arbitrary and there is no clear theoretical justification to choose one over the other. Consequently, the $|\alpha - 1| \approx 0$ hypothesis will depend on the choice of fitting range.

| Metric | Range | CIFAR-100 | ImageNet | Food-101 | Flowers-102 | Places-365 | EuroSAT |
|---|---|---|---|---|---|---|---|
| $\alpha$ | ours $(10, 0.9d)$ | -0.02 | -0.12 | -0.03 | $0.32^*$ | -0.18 | $0.30^*$ |
| | ours $(10, 0.8d)$ | -0.03 | -0.13 | -0.02 | $0.31^*$ | -0.17 | $0.31^*$ |
| | $(10, d//2)$ | -0.05 | -0.14 | -0.12 | $-0.29^*$ | -0.09 | $0.47^*$ |
| | $(0, 0.8d)$ | -0.05 | -0.13 | -0.10 | -0.14 | -0.13 | $0.47^*$ |
| | $(0, d)$ | -0.04 | -0.12 | -0.10 | -0.03 | -0.14 | $0.47^*$ |
| $|\alpha - 1|$ | ours $(10, 0.9d)$ | 0.09 | $0.25^*$ | -0.03 | $0.32^*$ | 0.10 | $0.30^*$ |
| | ours $(10, 0.8d)$ | 0.11 | $0.23^*$ | -0.02 | $0.31^*$ | 0.14 | $0.31^*$ |
| | $(10, d//2)$ | 0.07 | 0.10 | 0.03 | 0.13 | 0.10 | $0.47^*$ |
| | $(0, 0.8d)$ | 0.13 | 0.08 | -0.13 | -0.10 | 0.16 | $0.47^*$ |
| | $(0, d)$ | 0.18 | 0.06 | -0.14 | -0.03 | $0.18^*$ | $0.47^*$ |

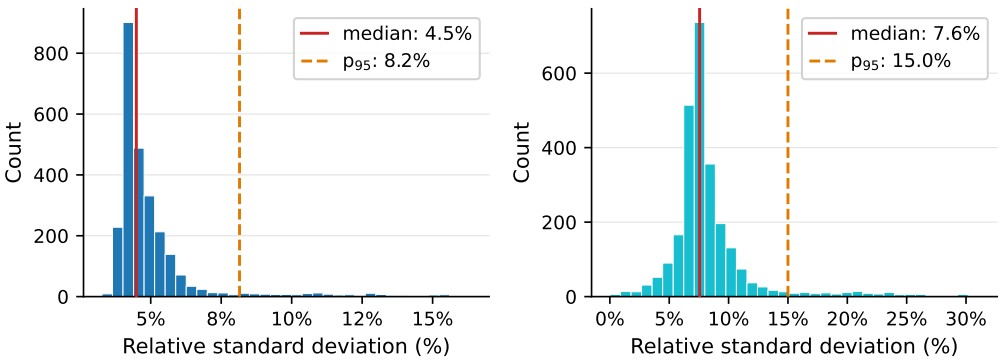

Figure 9: **TwoNN intrinsic dimensionality (ID) estimation noise across 260 models and 10 datasets (2,600 model–dataset pairs).** *Left*: Intra-batch subsample noise which is the relative standard deviation of ID across 20 within-batch subsamples (Number of repetition = 20, 90% subsampling rate within each batch). *Right*: Across-batch noise which is the relative standard deviation of the per-batch mean ID across batches. Median intra-batch and across-batch noise are 4.5% and 7.6% respectively, with 95th percentiles at 8.2% and 15.2%.

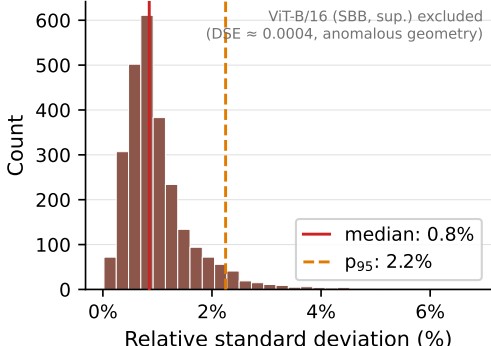

Figure 10: **DSE estimation noise across 260 models and 10 datasets.** Relative standard deviation of DSE across batches.

## B.2 ID (TwoNN) Estimation Noise

Figure 9 reports the relative standard deviation of the TwoNN ID estimate across all 260 models and 10 datasets (2,600 model–dataset pairs). The intra-batch noise — the variability of ID across 20 independent 90%-subsamples within a single batch — has a median of 4.5% and a 95th percentile of 8.2%. The across-batch noise — the variability of the per-batch mean ID across batches — has a median of 7.6% and a 95th percentile of 15.2%. Both distributions are right-skewed, with the majority of model–dataset pairs showing well under 10% relative noise. Given that the inter-model ID differences driving the correlation results span a factor of several units (e.g. $\rho \in [-0.95, -0.26]$ across the main analysis), the estimation noise is small relative to the signal.

## B.3 DSE Estimation Noise

Figure 10 reports the relative standard deviation of the DSE estimate across batches. The median relative standard deviation is 0.8%, with a 95th percentile of 2.2%, confirming that the batched DSE estimator is stable across the model pool.

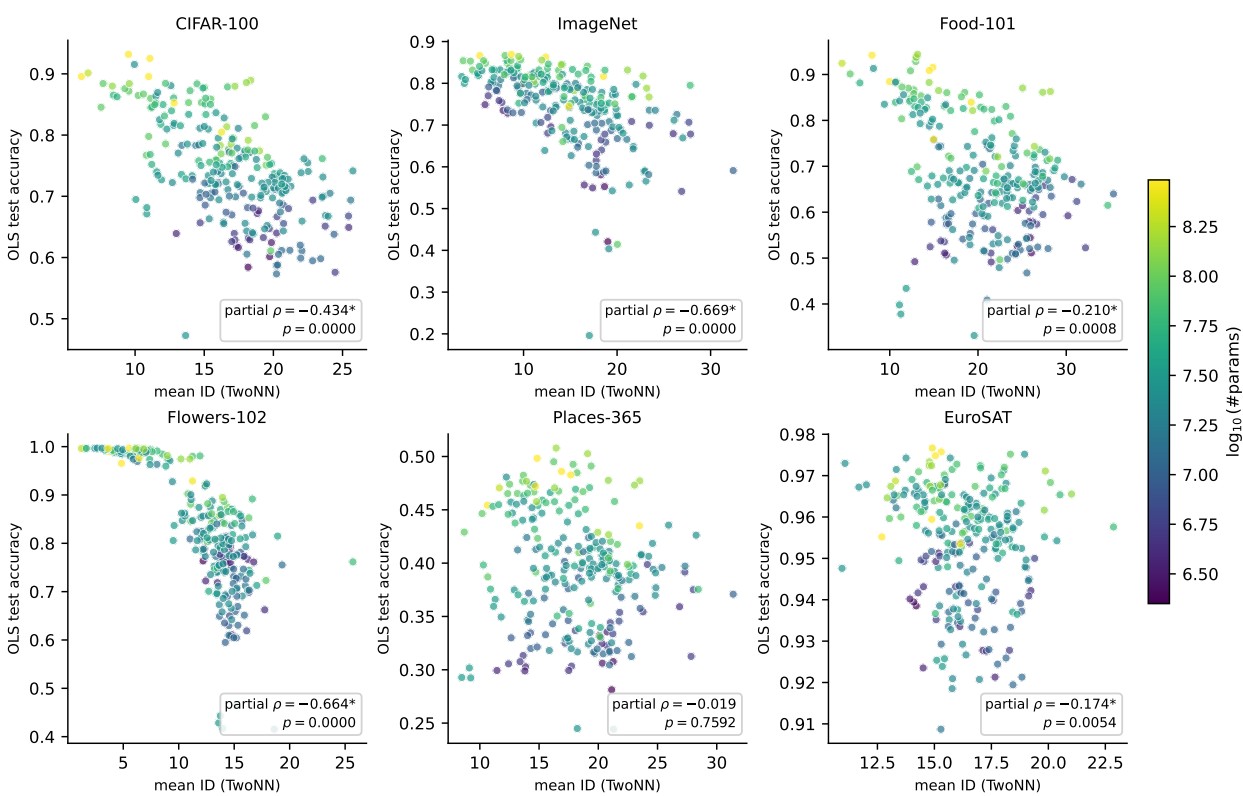

Figure 11: **ID vs OLS test accuracy, coloured by** $\log_{10}$ **parameter count, across six main datasets.** Partial Spearman correlations controlling for log parameter count are annotated in each panel ($n = 255$; 5 models excluded due to unavailable parameter counts). Asterisk (*) denotes $p < 0.05$.

### B.4 ID Partial Correlation Controlling for Model Capacity

Figure 11 shows ID against OLS test accuracy across the six main datasets, with points coloured by $\log_{10}$ parameter count. Parameter counts were obtained for a total of 255 models except `timm/vit_dlittle_patch16_reg1_gap_256.sbb_nadamuon_in1k`, `timm/vit_dpwee_patch16_reg1_gap_256.sbb_in1k`, `timm/vit_dpwee_patch16_reg1_gap_256.sbb_nadamuon_in1k`, `timm/vit_dwee_patch16_reg1_gap_256.sbb_in1k`, `timm/vit_dwee_patch16_reg1_gap_256.sbb_nadamuon_in1k`. We compute the partial Spearman correlation between ID and accuracy after removing the shared variance with log parameter count. ID remains a significant predictor on five of the six datasets after this correction: CIFAR-100 (partial $\rho = -0.434$, $p < 0.001$), ImageNet (partial $\rho = -0.669$, $p < 0.001$), Flowers-102 (partial $\rho = -0.664$, $p < 0.001$), and Food-101 (partial $\rho = -0.210$, $p = 0.001$). Only on Places-365 (partial $\rho = -0.019$, $p = 0.759$) the partial correlation is not significant. Places-365 is the dataset datasets where ID already showed its weakest pooled correlation in the main analysis (Table 2a).

## C Models used

Table 5: Summary of torch hub models and their characteristics. Params/GMACs are for the frozen feature backbone (classifier head removed), computed with `timm+fvcore`; GMACs are at the listed input resolution. Arrows denote a pretrain → fine-tune chain. LVD-142M/1689M: DINOv2/v3 curated web data; WIT-400M: OpenAI CLIP data; LAION: web image–text; IG: Instagram; YFCC-100M: Flickr; MetaCLIP-2B: Web-SSL web pool.

| GitHub Repo | Model Name | $d$ | Arch. | Objective | Params (M) | GMACs | Training data |
|---|---|---|---|---|---|---|---|
| facebookresearch/barlowtwins:main | barlowtwins | 1000 | ResNet | SSL | 25.6 | 4.11 | ImageNet-1k |
| facebookresearch/vicregl:main | resnet50_alpha0p9 | 2048 | ResNet | SSL | 23.5 | 4.11 | ImageNet-1k |
| facebookresearch/vicregl:main | resnet50_alpha0p75 | 2048 | ResNet | SSL | 23.5 | 4.11 | ImageNet-1k |
| facebookresearch/vicregl:main | convnext_small_alpha0p9 | 768 | ConvNeXT | SSL | 49.4 | 8.70 | ImageNet-1k |
| facebookresearch/vicregl:main | convnext_small_alpha0p75 | 768 | ConvNeXT | SSL | 49.4 | 8.70 | ImageNet-1k |
| facebookresearch/vicregl:main | convnext_base_alpha0p9 | 1024 | ConvNeXT | SSL | 87.5 | 15.38 | ImageNet-1k |
| facebookresearch/vicregl:main | convnext_base_alpha0p75 | 1024 | ConvNeXT | SSL | 87.5 | 15.38 | ImageNet-1k |
| facebookresearch/vicreg:main | resnet50 | 2048 | ResNet | SSL | 23.5 | 4.11 | ImageNet-1k |
| facebookresearch/vicreg:main | resnet50x2 | 4096 | ResNet | SSL | 93.9 | 16.16 | ImageNet-1k |
| facebookresearch/vicreg:main | resnet200x2 | 4096 | ResNet | SSL | 250.1 | 59.91 | ImageNet-1k |

Table 6: Summary of Hugging Face (Wolf et al., 2020) and `timm` models and their characteristics. Params/GMACs are for the frozen feature backbone (classifier head removed), computed with `timm+fvcore`; GMACs are at the listed input resolution. Arrows denote a pretrain → fine-tune chain. LVD-142M/1689M: DINOv2/v3 curated web data; WIT-400M: OpenAI CLIP data; LAION: web image–text; IG: Instagram; YFCC-100M: Flickr; MetaCLIP-2B: Web-SSL web pool.

| Model Name | $d$ | Arch. | Objective | Params (M) | GMACs | Training data |
|---|---|---|---|---|---|---|
| facebook/dinov2-small | 384 | ViT | SSL | 22.1 | 29.46 | LVD-142M |
| timm/vit_small_patch16_224.dino | 384 | ViT | SSL | 21.7 | 4.25 | ImageNet-1k |
| timm/vit_small_patch16_dinov3_qkvb.lvd1689m | 384 | ViT | SSL | 21.6 | 5.63 | LVD-1689M (web) |
| timm/vit_small_plus_patch16_dinov3_qkvb.lvd1689m | 384 | ViT | SSL | 28.7 | 7.48 | LVD-1689M (web) |
| facebook/dinov2-base | 768 | ViT | SSL | 86.6 | 117.11 | LVD-142M |
| timm/vit_base_patch16_224.dino | 768 | ViT | SSL | 85.8 | 16.87 | ImageNet-1k |
| timm/vit_base_patch16_dinov3_qkvb.lvd1689m | 768 | ViT | SSL | 85.7 | 22.34 | LVD-1689M (web) |
| timm/vit_base_patch8_224.dino | 768 | ViT | SSL | 85.8 | 66.86 | ImageNet-1k |
| facebook/vit-mae-base | 768 | ViT | SSL | 85.8 | 4.37 | ImageNet-1k |
| timm/vit_so150m_patch16_reg4_gap_256.sbb_e250_in12k_ft_in1k | 896 | ViT | SSL | 133.2 | 34.57 | ImageNet-12k → ImageNet-1k |
| timm/vit_so150m_patch16_reg4_gap_384.sbb_e250_in12k_ft_in1k | 896 | ViT | SSL | 133.5 | 77.12 | ImageNet-12k → ImageNet-1k |
| facebook/dinov2-large | 1024 | ViT | SSL | 304.4 | 414.89 | LVD-142M |
| facebook/webssl-dino300m-full2b-224 | 1024 | ViT | SSL | 303.7 | 77.93 | MetaCLIP-2B (web) |
| timm/vit_large_patch16_dinov3_qkvb.lvd1689m | 1024 | ViT | SSL | 303.1 | 79.09 | LVD-1689M (web) |
| facebook/vit-mae-large | 1024 | ViT | SSL | 303.3 | 15.27 | ImageNet-1k |
| facebook/webssl-mae300m-full2b-224 | 1024 | ViT | SSL | 304.4 | 59.70 | MetaCLIP-2B (web) |
| facebook/vit-mae-huge | 1280 | ViT | SSL | 630.8 | 41.11 | ImageNet-1k |
| facebook/webssl-mae700m-full2b-224 | 1280 | ViT | SSL | 632.4 | 161.99 | MetaCLIP-2B (web) |
| timm/eva_giant_patch14_224.clip_ft_in1k | 1408 | ViT | SSL | 1011.1 | 259.74 | LAION-400M (CLIP) → ImageNet-1k |
| timm/convnext_small.dinov3_lvd1689m | 768 | ConvNeXT | SSL | 49.5 | 8.70 | LVD-1689M (web) |
| timm/convnext_tiny.dinov3_lvd1689m | 768 | ConvNeXT | SSL | 27.8 | 4.47 | LVD-1689M (web) |
| timm/convnextv2_tiny.fcmae | 768 | ConvNeXT | SSL | 27.9 | 4.47 | ImageNet-1k |
| timm/convnext_base.dinov3_lvd1689m | 1024 | ConvNeXT | SSL | 87.6 | 15.38 | LVD-1689M (web) |
| timm/convnext_large.dinov3_lvd1689m | 1536 | ConvNeXT | SSL | 196.2 | 34.40 | LVD-1689M (web) |
| timm/vicregl_convnext_xlarge_alpha0p75 | 2048 | ConvNeXT | SSL | 348.1 | 60.97 | ImageNet-1k |
| timm/vit_base_patch16_clip_384.laion2b_ft_in12k_in1k | 768 | ViT | SSL+Sup | 86.1 | 49.40 | LAION-2B → ImageNet-12k → ImageNet-1k |
| timm/eva_large_patch14_196.in22k_ft_in22k_in1k | 1024 | ViT | SSL+Sup | 303.1 | 59.66 | ImageNet-22k → ImageNet-1k |
| timm/vit_large_patch14_clip_224.openai_ft_in12k_in1k | 1024 | ViT | SSL+Sup | 303.2 | 77.83 | WIT-400M → ImageNet-12k → ImageNet-1k |
| timm/convnextv2_atto.fcmae_ft_in1k | 320 | ConvNeXT | SSL+Sup | 3.4 | 0.55 | ImageNet-1k |
| timm/convnextv2_femto.fcmae_ft_in1k | 384 | ConvNeXT | SSL+Sup | 4.8 | 0.78 | ImageNet-1k |
| timm/convnextv2_pico.fcmae_ft_in1k | 512 | ConvNeXT | SSL+Sup | 8.6 | 1.37 | ImageNet-1k |
| timm/convnextv2_nano.fcmae_ft_in1k | 640 | ConvNeXT | SSL+Sup | 15.0 | 2.45 | ImageNet-1k |
| timm/convnextv2_nano.fcmae_ft_in22k_in1k | 640 | ConvNeXT | SSL+Sup | 15.0 | 2.45 | ImageNet-22k → ImageNet-1k |
| timm/convnextv2_nano.fcmae_ft_in22k_in1k_384 | 640 | ConvNeXT | SSL+Sup | 15.0 | 7.21 | ImageNet-22k → ImageNet-1k |
| timm/convnextv2_tiny.fcmae_ft_in1k | 768 | ConvNeXT | SSL+Sup | 27.9 | 4.47 | ImageNet-1k |
| timm/convnextv2_tiny.fcmae_ft_in22k_in1k | 768 | ConvNeXT | SSL+Sup | 27.9 | 4.47 | ImageNet-22k → ImageNet-1k |
| timm/convnextv2_tiny.fcmae_ft_in22k_in1k_384 | 768 | ConvNeXT | SSL+Sup | 27.9 | 13.14 | ImageNet-22k → ImageNet-1k |
| timm/convnextv2_base.fcmae_ft_in1k | 1024 | ConvNeXT | SSL+Sup | 87.7 | 15.38 | ImageNet-1k |
| timm/convnextv2_base.fcmae_ft_in22k_in1k | 1024 | ConvNeXT | SSL+Sup | 87.7 | 15.38 | ImageNet-22k → ImageNet-1k |
| timm/convnextv2_base.fcmae_ft_in22k_in1k_384 | 1024 | ConvNeXT | SSL+Sup | 87.7 | 45.21 | ImageNet-22k → ImageNet-1k |
| timm/convnext_base.clip_laion2b_augreg_ft_in12k_in1k | 1024 | ConvNeXT | SSL+Sup | 87.6 | 20.09 | LAION-2B → ImageNet-12k → ImageNet-1k |
| timm/convnext_base.clip_laion2b_augreg_ft_in12k_in1k_384 | 1024 | ConvNeXT | SSL+Sup | 87.6 | 45.21 | LAION-2B → ImageNet-12k → ImageNet-1k |
| timm/convnext_base.clip_laion2b_augreg_ft_in1k | 1024 | ConvNeXT | SSL+Sup | 87.6 | 20.09 | LAION-2B → ImageNet-1k |
| timm/convnext_base.clip_laiona_augreg_ft_in1k_384 | 1024 | ConvNeXT | SSL+Sup | 87.6 | 45.21 | LAION-Aesthetic → ImageNet-1k |
| timm/vit_dpwee_patch16_reg1_gap_256.sbb_in1k | 256 | ViT | Supervised | 15.0 | 3.83 | ImageNet-1k |

| Model Name | d | Arch. | Objective | Params (M) | GMACs | Training data |
|---|---|---|---|---|---|---|
| timm/vit_dpwee_patch16_reg1_gap_256.sbb_nadamuon_in1k | 256 | ViT | Supervised | 15.0 | 3.83 | ImageNet-1k |
| timm/vit_dwee_patch16_reg1_gap_256.sbb_in1k | 256 | ViT | Supervised | 13.2 | 3.36 | ImageNet-1k |
| timm/vit_dwee_patch16_reg1_gap_256.sbb_nadamuon_in1k | 256 | ViT | Supervised | 13.2 | 3.36 | ImageNet-1k |
| timm/vit_pwee_patch16_reg1_gap_256.sbb_in1k | 256 | ViT | Supervised | 15.0 | 3.83 | ImageNet-1k |
| timm/vit_wee_patch16_reg1_gap_256.sbb_in1k | 256 | ViT | Supervised | 13.2 | 3.36 | ImageNet-1k |
| timm/vit_dlittle_patch16_reg1_gap_256.sbb_nadamuon_in1k | 320 | ViT | Supervised | 22.2 | 5.67 | ImageNet-1k |
| timm/vit_little_patch16_reg1_gap_256.sbb_in12k | 320 | ViT | Supervised | 22.2 | 5.67 | ImageNet-12k |
| timm/vit_little_patch16_reg1_gap_256.sbb_in12k_ft_in1k | 320 | ViT | Supervised | 22.2 | 5.67 | ImageNet-12k → ImageNet-1k |
| timm/vit_little_patch16_reg4_gap_256.sbb_in1k | 320 | ViT | Supervised | 22.2 | 5.74 | ImageNet-1k |
| timm/vit_medium_patch16_reg1_gap_256.sbb_in1k | 512 | ViT | Supervised | 38.4 | 9.82 | ImageNet-1k |
| timm/vit_medium_patch16_reg4_gap_256.sbb_in12k | 512 | ViT | Supervised | 38.4 | 9.93 | ImageNet-12k |
| timm/vit_medium_patch16_reg4_gap_256.sbb_in12k_ft_in1k | 512 | ViT | Supervised | 38.4 | 9.93 | ImageNet-12k → ImageNet-1k |
| timm/vit_medium_patch16_reg4_gap_256.sbb_in1k | 512 | ViT | Supervised | 38.4 | 9.93 | ImageNet-1k |
| timm/vit_medium_patch16_rope_reg1_gap_256.sbb_in1k | 512 | ViT | Supervised | 38.2 | 9.82 | ImageNet-1k |
| timm/vit_mediumd_patch16_reg4_gap_256.sbb2_e200_in12k | 512 | ViT | Supervised | 63.6 | 16.49 | ImageNet-12k |
| timm/vit_mediumd_patch16_reg4_gap_256.sbb2_e200_in12k_ft_in1k | 512 | ViT | Supervised | 63.6 | 16.49 | ImageNet-12k → ImageNet-1k |
| timm/vit_mediumd_patch16_reg4_gap_256.sbb_in12k | 512 | ViT | Supervised | 63.6 | 16.49 | ImageNet-12k |
| timm/vit_mediumd_patch16_reg4_gap_256.sbb_in12k_ft_in1k | 512 | ViT | Supervised | 63.6 | 16.49 | ImageNet-12k → ImageNet-1k |
| timm/vit_mediumd_patch16_reg4_gap_384.sbb2_e200_in12k_ft_in1k | 512 | ViT | Supervised | 63.8 | 36.78 | ImageNet-12k → ImageNet-1k |
| timm/vit_mediumd_patch16_rope_reg1_gap_256.sbb_in1k | 512 | ViT | Supervised | 63.4 | 16.30 | ImageNet-1k |
| timm/vit_betwixt_patch16_reg1_gap_256.sbb_in1k | 640 | ViT | Supervised | 59.8 | 15.30 | ImageNet-1k |
| timm/vit_betwixt_patch16_reg4_gap_256.sbb2_e200_in12k | 640 | ViT | Supervised | 59.8 | 15.48 | ImageNet-12k |
| timm/vit_betwixt_patch16_reg4_gap_256.sbb2_e200_in12k_ft_in1k | 640 | ViT | Supervised | 59.8 | 15.48 | ImageNet-12k → ImageNet-1k |
| timm/vit_betwixt_patch16_reg4_gap_256.sbb_in12k | 640 | ViT | Supervised | 59.8 | 15.48 | ImageNet-12k |
| timm/vit_betwixt_patch16_reg4_gap_256.sbb_in12k_ft_in1k | 640 | ViT | Supervised | 59.8 | 15.48 | ImageNet-12k → ImageNet-1k |
| timm/vit_betwixt_patch16_reg4_gap_256.sbb_in1k | 640 | ViT | Supervised | 59.8 | 15.48 | ImageNet-1k |
| timm/vit_betwixt_patch16_reg4_gap_384.sbb2_e200_in12k_ft_in1k | 640 | ViT | Supervised | 60.0 | 34.54 | ImageNet-12k → ImageNet-1k |
| timm/vit_betwixt_patch16_rope_reg4_gap_256.sbb_in1k | 640 | ViT | Supervised | 59.6 | 15.48 | ImageNet-1k |
| timm/vit_base_patch16_rope_reg1_gap_256.sbb_in1k | 768 | ViT | Supervised | 85.7 | 22.00 | ImageNet-1k |
| timm/vit_so150m2_patch16_reg1_gap_256.sbb_e200_in12k | 832 | ViT | Supervised | 135.2 | 34.69 | ImageNet-12k |
| timm/vit_so150m2_patch16_reg1_gap_256.sbb_e200_in12k_ft_in1k | 832 | ViT | Supervised | 135.2 | 34.69 | ImageNet-12k → ImageNet-1k |
| timm/vit_so150m2_patch16_reg1_gap_384.sbb_e200_in12k_ft_in1k | 832 | ViT | Supervised | 135.5 | 77.89 | ImageNet-12k → ImageNet-1k |
| timm/vit_large_patch32_224.orig_in21k | 1024 | ViT | Supervised | 305.5 | 15.27 | ImageNet-21k |
| timm/resnet10t.c3_in1k | 512 | ResNet | Supervised | 4.9 | 0.70 | ImageNet-1k |
| timm/resnet18.a1_in1k | 512 | ResNet | Supervised | 11.2 | 1.82 | ImageNet-1k |
| timm/resnet18.a2_in1k | 512 | ResNet | Supervised | 11.2 | 1.82 | ImageNet-1k |
| timm/resnet18.a3_in1k | 512 | ResNet | Supervised | 11.2 | 0.93 | ImageNet-1k |
| timm/resnet18.fb_ssl_yfcc100m_ft_in1k | 512 | ResNet | Supervised | 11.2 | 1.82 | YFCC-100M → ImageNet-1k |
| timm/resnet18.fb_swsl_ig1b_ft_in1k | 512 | ResNet | Supervised | 11.2 | 1.82 | IG-1B → ImageNet-1k |
| timm/resnet18.gluon_in1k | 512 | ResNet | Supervised | 11.2 | 1.82 | ImageNet-1k |
| timm/resnet18.tv_in1k | 512 | ResNet | Supervised | 11.2 | 1.82 | ImageNet-1k |
| timm/resnet18d.ra2_in1k | 512 | ResNet | Supervised | 11.2 | 2.06 | ImageNet-1k |
| timm/resnet18d.ra4_e3600_r224_in1k | 512 | ResNet | Supervised | 11.2 | 2.06 | ImageNet-1k |
| timm/resnet34.a1_in1k | 512 | ResNet | Supervised | 21.3 | 3.67 | ImageNet-1k |
| timm/resnet34.a2_in1k | 512 | ResNet | Supervised | 21.3 | 3.67 | ImageNet-1k |
| timm/resnet34.a3_in1k | 512 | ResNet | Supervised | 21.3 | 1.87 | ImageNet-1k |
| timm/resnet34.bt_in1k | 512 | ResNet | Supervised | 21.3 | 3.67 | ImageNet-1k |
| timm/resnet34.gluon_in1k | 512 | ResNet | Supervised | 21.3 | 3.67 | ImageNet-1k |
| timm/resnet34.ra4_e3600_r224_in1k | 512 | ResNet | Supervised | 21.3 | 3.67 | ImageNet-1k |
| timm/resnet34.tv_in1k | 512 | ResNet | Supervised | 21.3 | 3.67 | ImageNet-1k |
| timm/resnet34d.ra2_in1k | 512 | ResNet | Supervised | 21.3 | 3.91 | ImageNet-1k |
| timm/resnetv2_18.ra4_e3600_r224_in1k | 512 | ResNet | Supervised | 11.2 | 1.82 | ImageNet-1k |
| timm/resnetv2_18d.ra4_e3600_r224_in1k | 512 | ResNet | Supervised | 11.2 | 2.06 | ImageNet-1k |
| timm/resnetv2_34.ra4_e3600_r224_in1k | 512 | ResNet | Supervised | 21.3 | 3.67 | ImageNet-1k |
| timm/resnetv2_34d.ra4_e3600_r224_in1k | 512 | ResNet | Supervised | 21.3 | 3.91 | ImageNet-1k |
| timm/resnet33ts.ra2_in1k | 1280 | ResNet | Supervised | 18.4 | 4.75 | ImageNet-1k |
| timm/resnet32ts.ra2_in1k | 1536 | ResNet | Supervised | 16.4 | 4.63 | ImageNet-1k |
| timm/resnet101.a1h_in1k | 2048 | ResNet | Supervised | 42.5 | 7.83 | ImageNet-1k |
| timm/resnet101.a3_in1k | 2048 | ResNet | Supervised | 42.5 | 4.00 | ImageNet-1k |
| timm/resnet101.tv2_in1k | 2048 | ResNet | Supervised | 42.5 | 4.92 | ImageNet-1k |
| timm/resnet14t.c3_in1k | 2048 | ResNet | Supervised | 8.0 | 1.07 | ImageNet-1k |
| timm/resnet26.bt_in1k | 2048 | ResNet | Supervised | 13.9 | 2.36 | ImageNet-1k |
| timm/resnet26d.bt_in1k | 2048 | ResNet | Supervised | 14.0 | 2.60 | ImageNet-1k |
| timm/resnet26t.ra2_in1k | 2048 | ResNet | Supervised | 14.0 | 3.35 | ImageNet-1k |
| timm/resnet50.a1_in1k | 2048 | ResNet | Supervised | 23.5 | 4.11 | ImageNet-1k |
| timm/resnet50.a1h_in1k | 2048 | ResNet | Supervised | 23.5 | 2.62 | ImageNet-1k |
| timm/resnet50.a2_in1k | 2048 | ResNet | Supervised | 23.5 | 4.11 | ImageNet-1k |
| timm/resnet50.a3_in1k | 2048 | ResNet | Supervised | 23.5 | 2.10 | ImageNet-1k |
| timm/resnet50.am_in1k | 2048 | ResNet | Supervised | 23.5 | 4.11 | ImageNet-1k |
| timm/resnet50.b1k_in1k | 2048 | ResNet | Supervised | 23.5 | 4.11 | ImageNet-1k |
| timm/resnet50.b2k_in1k | 2048 | ResNet | Supervised | 23.5 | 4.11 | ImageNet-1k |
| timm/resnet50.bt_in1k | 2048 | ResNet | Supervised | 23.5 | 4.11 | ImageNet-1k |
| timm/resnet50.c1_in1k | 2048 | ResNet | Supervised | 23.5 | 4.11 | ImageNet-1k |
| timm/resnet50.c2_in1k | 2048 | ResNet | Supervised | 23.5 | 4.11 | ImageNet-1k |
| timm/resnet50.d_in1k | 2048 | ResNet | Supervised | 23.5 | 4.11 | ImageNet-1k |
| timm/resnet50.fb_ssl_yfcc100m_ft_in1k | 2048 | ResNet | Supervised | 23.5 | 4.11 | YFCC-100M → ImageNet-1k |
| timm/resnet50.fb_swsl_ig1b_ft_in1k | 2048 | ResNet | Supervised | 23.5 | 4.11 | IG-1B → ImageNet-1k |
| timm/resnet50.gluon_in1k | 2048 | ResNet | Supervised | 23.5 | 4.11 | ImageNet-1k |
| timm/resnet50.ra_in1k | 2048 | ResNet | Supervised | 23.5 | 4.11 | ImageNet-1k |
| timm/resnet50.ram_in1k | 2048 | ResNet | Supervised | 23.5 | 4.11 | ImageNet-1k |
| timm/resnet50.tv2_in1k | 2048 | ResNet | Supervised | 23.5 | 2.62 | ImageNet-1k |
| timm/resnet50.tv_in1k | 2048 | ResNet | Supervised | 23.5 | 4.11 | ImageNet-1k |
| timm/resnet50_gn.a1h_in1k | 2048 | ResNet | Supervised | 23.5 | 4.14 | ImageNet-1k |
| timm/resnet50c.gluon_in1k | 2048 | ResNet | Supervised | 23.5 | 4.35 | ImageNet-1k |
| timm/resnet50d.a1_in1k | 2048 | ResNet | Supervised | 23.5 | 4.35 | ImageNet-1k |
| timm/resnet50d.a2_in1k | 2048 | ResNet | Supervised | 23.5 | 4.35 | ImageNet-1k |
| timm/resnet50d.a3_in1k | 2048 | ResNet | Supervised | 23.5 | 2.22 | ImageNet-1k |
| timm/resnet50d.gluon_in1k | 2048 | ResNet | Supervised | 23.5 | 4.35 | ImageNet-1k |
| timm/resnet50d.ra2_in1k | 2048 | ResNet | Supervised | 23.5 | 4.35 | ImageNet-1k |
| timm/resnet50d.ra4_e3600_r224_in1k | 2048 | ResNet | Supervised | 23.5 | 4.35 | ImageNet-1k |
| timm/resnetrs50.tf_in1k | 2048 | ResNet | Supervised | 33.6 | 2.29 | ImageNet-1k |
| timm/regnetx_002.pycls_in1k | 368 | RegNet | Supervised | 2.3 | 0.20 | ImageNet-1k |

| Model Name | $d$ | Arch. | Objective | Params (M) | GMACs | Training data |
|---|---|---|---|---|---|---|
| timm/regnety_002.pycls_in1k | 368 | RegNet | Supervised | 2.8 | 0.20 | ImageNet-1k |
| timm/regnetx_004.pycls_in1k | 384 | RegNet | Supervised | 4.8 | 0.40 | ImageNet-1k |
| timm/regnetx_004_tv.tv2_in1k | 400 | RegNet | Supervised | 5.1 | 0.42 | ImageNet-1k |
| timm/regnety_004.pycls_in1k | 440 | RegNet | Supervised | 3.9 | 0.41 | ImageNet-1k |
| timm/regnety_004.tv2_in1k | 440 | RegNet | Supervised | 3.9 | 0.41 | ImageNet-1k |
| timm/regnetx_006.pycls_in1k | 528 | RegNet | Supervised | 5.7 | 0.61 | ImageNet-1k |
| timm/regnetz_040.ra3_in1k | 528 | RegNet | Supervised | 26.6 | 4.06 | ImageNet-1k |
| timm/regnety_006.pycls_in1k | 608 | RegNet | Supervised | 5.5 | 0.61 | ImageNet-1k |
| timm/regnetx_008.pycls_in1k | 672 | RegNet | Supervised | 6.6 | 0.81 | ImageNet-1k |
| timm/regnetx_008.tv2_in1k | 672 | RegNet | Supervised | 6.6 | 0.81 | ImageNet-1k |
| timm/regnety_008.pycls_in1k | 768 | RegNet | Supervised | 5.5 | 0.81 | ImageNet-1k |
| timm/regnety_008_tv.tv2_in1k | 784 | RegNet | Supervised | 5.7 | 0.84 | ImageNet-1k |
| timm/regnety_016.pycls_in1k | 888 | RegNet | Supervised | 10.3 | 1.63 | ImageNet-1k |
| timm/regnety_016.tv2_in1k | 888 | RegNet | Supervised | 10.3 | 1.63 | ImageNet-1k |
| timm/regnetx_016.pycls_in1k | 912 | RegNet | Supervised | 8.3 | 1.62 | ImageNet-1k |
| timm/regnetx_016.tv2_in1k | 912 | RegNet | Supervised | 8.3 | 1.62 | ImageNet-1k |
| timm/regnetx_032.pycls_in1k | 1008 | RegNet | Supervised | 14.3 | 3.20 | ImageNet-1k |
| timm/regnetx_032.tv2_in1k | 1008 | RegNet | Supervised | 14.3 | 3.20 | ImageNet-1k |
| timm/regnetv_040.ra3_in1k | 1088 | RegNet | Supervised | 19.6 | 4.00 | ImageNet-1k |
| timm/regnety_040.pycls_in1k | 1088 | RegNet | Supervised | 19.6 | 4.00 | ImageNet-1k |
| timm/regnety_040.ra3_in1k | 1088 | RegNet | Supervised | 19.6 | 4.00 | ImageNet-1k |
| timm/regnetv_064.ra3_in1k | 1296 | RegNet | Supervised | 29.3 | 6.38 | ImageNet-1k |
| timm/regnety_064.pycls_in1k | 1296 | RegNet | Supervised | 29.3 | 6.39 | ImageNet-1k |
| timm/regnety_064.ra3_in1k | 1296 | RegNet | Supervised | 29.3 | 6.39 | ImageNet-1k |
| timm/regnetx_040.pycls_in1k | 1360 | RegNet | Supervised | 20.8 | 3.99 | ImageNet-1k |
| timm/regnety_032.pycls_in1k | 1512 | RegNet | Supervised | 17.9 | 3.20 | ImageNet-1k |
| timm/regnety_032.ra_in1k | 1512 | RegNet | Supervised | 17.9 | 3.20 | ImageNet-1k |
| timm/regnety_032.tv2_in1k | 1512 | RegNet | Supervised | 17.9 | 3.20 | ImageNet-1k |
| timm/regnetz_040_h.ra3_in1k | 1536 | RegNet | Supervised | 27.4 | 4.12 | ImageNet-1k |
| timm/regnetz_b16.ra3_in1k | 1536 | RegNet | Supervised | 8.2 | 1.45 | ImageNet-1k |
| timm/regnetz_c16.ra3_in1k | 1536 | RegNet | Supervised | 11.9 | 2.51 | ImageNet-1k |
| timm/regnetz_c16_evos.ch_in1k | 1536 | RegNet | Supervised | 11.9 | 2.47 | ImageNet-1k |
| timm/regnetx_064.pycls_in1k | 1624 | RegNet | Supervised | 24.6 | 6.49 | ImageNet-1k |
| timm/regnetz_d32.ra3_in1k | 1792 | RegNet | Supervised | 25.8 | 5.98 | ImageNet-1k |
| timm/regnetz_d8.ra3_in1k | 1792 | RegNet | Supervised | 21.6 | 3.97 | ImageNet-1k |
| timm/regnetz_d8_evos.ch_in1k | 1792 | RegNet | Supervised | 21.7 | 4.50 | ImageNet-1k |
| timm/regnetx_080.pycls_in1k | 1920 | RegNet | Supervised | 37.6 | 8.02 | ImageNet-1k |
| timm/regnetx_080.tv2_in1k | 1920 | RegNet | Supervised | 37.6 | 8.02 | ImageNet-1k |
| timm/regnety_080.pycls_in1k | 2016 | RegNet | Supervised | 37.2 | 8.00 | ImageNet-1k |
| timm/regnety_080.ra3_in1k | 2016 | RegNet | Supervised | 37.2 | 8.00 | ImageNet-1k |
| timm/regnety_080_tv.tv2_in1k | 2016 | RegNet | Supervised | 37.4 | 8.51 | ImageNet-1k |
| timm/regnetx_160.pycls_in1k | 2048 | RegNet | Supervised | 52.2 | 15.99 | ImageNet-1k |
| timm/regnetx_160.tv2_in1k | 2048 | RegNet | Supervised | 52.2 | 15.99 | ImageNet-1k |
| timm/regnetz_e8.ra3_in1k | 2048 | RegNet | Supervised | 55.6 | 9.90 | ImageNet-1k |
| timm/regnetx_120.pycls_in1k | 2240 | RegNet | Supervised | 43.9 | 12.13 | ImageNet-1k |
| timm/regnety_120.pycls_in1k | 2240 | RegNet | Supervised | 49.6 | 12.13 | ImageNet-1k |
| timm/regnety_120.sw_in12k_ft_in1k | 2240 | RegNet | Supervised | 49.6 | 12.13 | ImageNet-12k → ImageNet-1k |
| timm/regnetx_320.pycls_in1k | 2520 | RegNet | Supervised | 105.3 | 31.81 | ImageNet-1k |
| timm/regnetx_320.tv2_in1k | 2520 | RegNet | Supervised | 105.3 | 31.81 | ImageNet-1k |
| timm/regnety_160.deit_in1k | 3024 | RegNet | Supervised | 80.6 | 15.95 | ImageNet-1k |
| timm/regnety_160.lion_in12k_ft_in1k | 3024 | RegNet | Supervised | 80.6 | 15.95 | ImageNet-12k → ImageNet-1k |
| timm/regnety_160.pycls_in1k | 3024 | RegNet | Supervised | 80.6 | 15.95 | ImageNet-1k |
| timm/regnety_160.sw_in12k_ft_in1k | 3024 | RegNet | Supervised | 80.6 | 15.95 | ImageNet-12k → ImageNet-1k |
| timm/regnety_160.swag_ft_in1k | 3024 | RegNet | Supervised | 80.6 | 46.87 | IG-3.6B → ImageNet-1k |
| timm/regnety_160.swag_lc_in1k | 3024 | RegNet | Supervised | 80.6 | 15.95 | IG-3.6B → ImageNet-1k |
| timm/regnety_160.tv2_in1k | 3024 | RegNet | Supervised | 80.6 | 15.95 | ImageNet-1k |
| timm/regnety_320.pycls_in1k | 3712 | RegNet | Supervised | 141.3 | 32.34 | ImageNet-1k |
| timm/regnety_320.seer_ft_in1k | 3712 | RegNet | Supervised | 141.3 | 95.00 | IG-1B → ImageNet-1k |
| timm/regnety_320.swag_ft_in1k | 3712 | RegNet | Supervised | 141.3 | 95.00 | IG-3.6B → ImageNet-1k |
| timm/regnety_320.swag_lc_in1k | 3712 | RegNet | Supervised | 141.3 | 32.34 | IG-3.6B → ImageNet-1k |
| timm/regnety_320.tv2_in1k | 3712 | RegNet | Supervised | 141.3 | 32.34 | ImageNet-1k |
| timm/regnety_640.seer_ft_in1k | 4920 | RegNet | Supervised | 276.5 | 188.47 | IG-1B → ImageNet-1k |
| timm/mobilenetv4_conv_small_050.e3000_r224_in1k | 480 | others | Supervised | 1.0 | 0.07 | ImageNet-1k |
| timm/convformer_m36.sail_in22k_ft_in1k | 576 | others | Supervised | 53.4 | 12.88 | ImageNet-22k → ImageNet-1k |
| timm/mobilenetv4_conv_aa_large.e230_r384_in12k_ft_in1k | 960 | others | Supervised | 31.3 | 7.07 | ImageNet-12k → ImageNet-1k |
| timm/mobilenetv4_conv_aa_large.e230_r448_in12k_ft_in1k | 960 | others | Supervised | 31.3 | 9.62 | ImageNet-12k → ImageNet-1k |
| timm/mobilenetv4_conv_aa_large.e600_r384_in1k | 960 | others | Supervised | 31.3 | 7.07 | ImageNet-1k |
| timm/mobilenetv4_conv_blur_medium.e500_r224_in1k | 960 | others | Supervised | 8.4 | 1.22 | ImageNet-1k |
| timm/mobilenetv4_conv_large.e500_r256_in1k | 960 | others | Supervised | 31.3 | 2.86 | ImageNet-1k |
| timm/mobilenetv4_conv_large.e600_r384_in1k | 960 | others | Supervised | 31.3 | 6.43 | ImageNet-1k |
| timm/mobilenetv4_conv_medium.e500_r224_in1k | 960 | others | Supervised | 8.4 | 0.84 | ImageNet-1k |
| timm/mobilenetv4_conv_medium.e500_r256_in1k | 960 | others | Supervised | 8.4 | 1.09 | ImageNet-1k |
| timm/mobilenetv4_conv_small.e1200_r224_in1k | 960 | others | Supervised | 2.5 | 0.19 | ImageNet-1k |
| timm/mobilenetv4_conv_small.e2400_r224_in1k | 960 | others | Supervised | 2.5 | 0.19 | ImageNet-1k |
| timm/beitv2_large_patch16_224.in1k_ft_in22k_in1k | 1024 | others | Supervised | 303.4 | 59.69 | ImageNet-1k → ImageNet-22k → ImageNet-1k |
| timm/mobilenetv1_100.ra4_e3600_r224_in1k | 1024 | others | Supervised | 3.2 | 0.58 | ImageNet-1k |
| timm/mobilenetv1_100h.ra4_e3600_r224_in1k | 1024 | others | Supervised | 4.3 | 0.63 | ImageNet-1k |
| timm/mobilenetv1_125.ra4_e3600_r224_in1k | 1280 | others | Supervised | 5.0 | 0.89 | ImageNet-1k |
| timm/mobilenetv3_large_100.ra4_e3600_r224_in1k | 1280 | others | Supervised | 4.2 | 0.22 | ImageNet-1k |
| timm/mobilenetv3_large_100.ra_in1k | 1280 | others | Supervised | 4.2 | 0.22 | ImageNet-1k |
| timm/mobilenetv3_large_150d.ra4_e3600_r256_in1k | 1280 | others | Supervised | 13.3 | 1.03 | ImageNet-1k |
| timm/mobilenetv4_hybrid_large.e600_r384_in1k | 1280 | others | Supervised | 36.5 | 7.43 | ImageNet-1k |
| timm/mobilenetv4_hybrid_large.ix_e600_r384_in1k | 1280 | others | Supervised | 36.5 | 7.43 | ImageNet-1k |
| timm/mobilenetv4_hybrid_medium.e200_r256_in12k_ft_in1k | 1280 | others | Supervised | 9.8 | 1.24 | ImageNet-12k → ImageNet-1k |
| timm/mobilenetv4_hybrid_medium.e500_r224_in1k | 1280 | others | Supervised | 9.8 | 0.95 | ImageNet-1k |
| timm/mobilenetv4_hybrid_medium.ix_e550_r256_in1k | 1280 | others | Supervised | 9.8 | 1.24 | ImageNet-1k |
| timm/mobilenetv4_hybrid_medium.ix_e550_r384_in1k | 1280 | others | Supervised | 9.8 | 2.80 | ImageNet-1k |
| timm/mobilenet_edgetpu_v2_m.ra4_e3600_r224_in1k | 1344 | others | Supervised | 7.1 | 1.85 | ImageNet-1k |
| timm/convnext_atto.d2_in1k | 320 | ConvNeXT | Supervised | 3.4 | 0.55 | ImageNet-1k |
| timm/convnext_atto_ols.a2_in1k | 320 | ConvNeXT | Supervised | 3.4 | 0.58 | ImageNet-1k |
| timm/convnext_femto.d1_in1k | 384 | ConvNeXT | Supervised | 4.8 | 0.78 | ImageNet-1k |

*Continued on next page*

| Model Name | $d$ | Arch. | Objective | Params (M) | GMACs | Training data |
|---|---|---|---|---|---|---|
| timm/convnext_femto_ols.d1_in1k | 384 | ConvNeXT | Supervised | 4.8 | 0.82 | ImageNet-1k |
| timm/convnext_pico.d1_in1k | 512 | ConvNeXT | Supervised | 8.5 | 1.37 | ImageNet-1k |
| timm/convnext_pico_ols.d1_in1k | 512 | ConvNeXT | Supervised | 8.6 | 1.43 | ImageNet-1k |
| timm/convnext_nano.d1h_in1k | 640 | ConvNeXT | Supervised | 14.9 | 2.45 | ImageNet-1k |
| timm/convnext_nano.in12k | 640 | ConvNeXT | Supervised | 14.9 | 2.45 | ImageNet-12k |
| timm/convnext_nano.in12k_ft_in1k | 640 | ConvNeXT | Supervised | 14.9 | 2.45 | ImageNet-12k → ImageNet-1k |
| timm/convnext_nano_ols.d1h_in1k | 640 | ConvNeXT | Supervised | 15.0 | 2.65 | ImageNet-1k |
| timm/convnext_small.fb_in1k | 768 | ConvNeXT | Supervised | 49.5 | 8.70 | ImageNet-1k |
| timm/convnext_small.fb_in22k_ft_in1k | 768 | ConvNeXT | Supervised | 49.5 | 8.70 | ImageNet-22k → ImageNet-1k |
| timm/convnext_small.fb_in22k_ft_in1k_384 | 768 | ConvNeXT | Supervised | 49.5 | 25.58 | ImageNet-22k → ImageNet-1k |
| timm/convnext_small.in12k | 768 | ConvNeXT | Supervised | 49.5 | 8.70 | ImageNet-12k |
| timm/convnext_small.in12k_ft_in1k | 768 | ConvNeXT | Supervised | 49.5 | 8.70 | ImageNet-12k → ImageNet-1k |
| timm/convnext_small.in12k_ft_in1k_384 | 768 | ConvNeXT | Supervised | 49.5 | 25.58 | ImageNet-12k → ImageNet-1k |
| timm/convnext_tiny.fb_in1k | 768 | ConvNeXT | Supervised | 27.8 | 4.47 | ImageNet-1k |
| timm/convnext_tiny.fb_in22k_ft_in1k | 768 | ConvNeXT | Supervised | 27.8 | 4.47 | ImageNet-22k → ImageNet-1k |
| timm/convnext_tiny.fb_in22k_ft_in1k_384 | 768 | ConvNeXT | Supervised | 27.8 | 13.14 | ImageNet-22k → ImageNet-1k |
| timm/convnext_tiny.in12k | 768 | ConvNeXT | Supervised | 27.8 | 4.47 | ImageNet-12k |
| timm/convnext_tiny.in12k_ft_in1k | 768 | ConvNeXT | Supervised | 27.8 | 4.47 | ImageNet-12k → ImageNet-1k |
| timm/convnext_tiny.in12k_ft_in1k_384 | 768 | ConvNeXT | Supervised | 27.8 | 13.14 | ImageNet-12k → ImageNet-1k |
| timm/convnext_tiny_hnf.a2h_in1k | 768 | ConvNeXT | Supervised | 27.8 | 4.47 | ImageNet-1k |
| timm/convnext_base.fb_in1k | 1024 | ConvNeXT | Supervised | 87.6 | 15.38 | ImageNet-1k |
| timm/convnext_base.fb_in22k_ft_in1k | 1024 | ConvNeXT | Supervised | 87.6 | 15.38 | ImageNet-22k → ImageNet-1k |
| timm/convnext_base.fb_in22k_ft_in1k_384 | 1024 | ConvNeXT | Supervised | 87.6 | 45.21 | ImageNet-22k → ImageNet-1k |
| facebook/dinov2-giant | 1536 | ViT | SSL | 1136.5 | 1553.56 | LVD-142M |
| timm/vit_small_patch14_dinov2.lvd142m | 384 | ViT | SSL | 22.1 | 29.46 | LVD-142M |
| timm/vit_base_patch14_dinov2.lvd142m | 768 | ViT | SSL | 86.6 | 117.11 | LVD-142M |
| timm/vit_large_patch14_dinov2.lvd142m | 1024 | ViT | SSL | 304.4 | 414.89 | LVD-142M |
| facebook/webssl-dino1b-full2b-224 | 1536 | ViT | SSL | 1134.8 | 291.43 | MetaCLIP-2B (web) |
| timm/eva02_base_patch14_448.mim_in22k_ft_in22k_in1k | 768 | ViT | SSL+Sup | 86.3 | 87.74 | ImageNet-22k → ImageNet-1k |
| timm/vit_so150m2_patch16_reg1_gap_448.sbb_e200_in12k_ft_in1k | 882 | ViT | Supervised | 135.7 | 105.97 | ImageNet-12k → ImageNet-1k |
| timm/efficientnet_b0.ra4_e3600_r224_in1k | 1280 | others | Supervised | 4.0 | 0.40 | ImageNet-1k |
| timm/efficientnet_b1.ra4_e3600_r240_in1k | 1280 | others | Supervised | 6.5 | 0.71 | ImageNet-1k |

# D   Results on a Broader Dataset

Spearman correlations are reported for a broader set of datasets; the main-text datasets are included for ease of comparison. All datasets use their officially provided train and test splits, with two exceptions. EuroSAT (Helber et al., 2019) provides no official partition, so we construct a deterministic stratified 80/20 split seeded with `random.Random(0)`, ensuring disjoint and reproducible train/test sets. For Places-365 (Zhou et al., 2018), images are decoded at 256×256 resolution using PIL's draft mode to reduce I/O cost; we use the standard train and validation splits.

## D.1   OLS linear probe

We also include the stratified results on the OLS linear probe.

Table 7: Spearman correlation ($\rho$) between representation quality metrics and OLS test accuracy across 10 evaluation datasets, grouped by task type. Metrics are organised by construction: spectral (top), relation-based (middle), and manifold-based (bottom). Statistical significance is considered after Benjamini–Hochberg correction. $^{*}$ $p < 0.05$, $^{**}$ $p < 0.01$, $^{***}$ $p < 0.001$. Non-significant cells are greyed out.

(a) Generic Object Detection

|  | CIFAR-10 | CIFAR-100 | ImageNet$^{\ddagger}$ | STL-10 |
|---|---|---|---|---|
| $\alpha < 1.0$ | -0.3242 | $-0.5381^{**}$ | $-0.347^{**}$ | 0.3095 |
| $\alpha \geq 1.0$ | 0.1156 | 0.06995 | 0.1688 | $0.1984^{*}$ |
| $\alpha$-ReQ | 0.03393 | -0.01968 | -0.1169 | $0.2267^{*}$ |
| RankMe | 0.1093 | $0.2443^{*}$ | $0.2166^{*}$ | 0.09793 |
| RankMe$^{\star}$ | -0.1725 | -0.1362 | -0.05487 | $-0.2319^{*}$ |
| NE-Sum | 0.1991 | $0.2309^{*}$ | 0.1048 | 0.1783 |
| $\kappa$ | 0.2196 | $0.3331^{**}$ | 0.1599 | $0.3003^{**}$ |
| Self-Cluster | -0.1636 | -0.1803 | -0.09359 | -0.205 |
| DSE | 0.1712 | 0.1967 | 0.1096 | $0.2092^{*}$ |
| ID | $-0.5451^{**}$ | $-0.5981^{**}$ | $-0.6254^{**}$ | $-0.5654^{**}$ |

(b) Fine-grained Object Detection

|  | Food-101 | Pets | Flowers-102 |
|---|---|---|---|
| $\alpha < 1.0$ | – | – | – |
| $\alpha \geq 1.0$ | -0.03956 | $0.1717^{*}$ | $0.3184^{**}$ |
| $\alpha$-ReQ | -0.02593 | $0.1717^{*}$ | $0.3184^{**}$ |
| RankMe | $0.3823^{**}$ | $0.1838^{*}$ | 0.0921 |
| RankMe$^{\star}$ | -0.09887 | -0.1032 | -0.1966 |
| NE-Sum | $0.4116^{**}$ | $0.2429^{**}$ | $0.4107^{**}$ |
| $\kappa$ | $0.2716^{*}$ | 0.06511 | 0.1459 |
| Self-Cluster | $-0.3015^{**}$ | $-0.1988^{*}$ | $-0.2602^{*}$ |
| DSE | $0.3099^{**}$ | $0.2034^{*}$ | $0.2643^{*}$ |
| ID | $-0.3683^{**}$ | $-0.3086^{**}$ | $-0.7569^{**}$ |

(c) Specialised

|  | DTD | Places-365 | EuroSAT |
|---|---|---|---|
| $\alpha < 1.0$ | – | $-0.3235^{**}$ | – |
| $\alpha \geq 1.0$ | 0.009892 | 0.062 | $0.2965^{**}$ |
| $\alpha$-ReQ | 0.002006 | $-0.1824^{*}$ | $0.2965^{**}$ |
| RankMe | $-0.403^{*}$ | $0.4602^{**}$ | $0.4242^{**}$ |
| RankMe$^{\star}$ | $0.2843^{*}$ | 0.006721 | $-0.548^{**}$ |
| NE-Sum | $-0.2804^{*}$ | $0.3007^{**}$ | $-0.2561^{**}$ |
| $\kappa$ | -0.2502 | $0.3707^{**}$ | $0.5456^{**}$ |
| Self-Cluster | 0.103 | $-0.2692^{**}$ | 0.0856 |
| DSE | -0.1052 | $0.2872^{**}$ | -0.09127 |
| ID | $-0.2849^{**}$ | $-0.1815^{*}$ | $-0.2128^{*}$ |

$^{\ddagger}$ Correlations of ImageNet-1k should be interpreted with caution as it is used in the training pipeline of most backbones

### D.2 KNN probe

We also include the stratified results on the KNN linear probe.

Table 8: Spearman correlation ($\rho$) between representation quality metrics and OLS test accuracy, stratified by architecture class, across natural image datasets. Metrics are organised by construction: spectral (top), relation-based (middle), and manifold-based (bottom). Arrows indicate the preferred direction: ↑ (higher is better), ↓ (lower is better), → 1 (closer to 1 is better). Statistical significance is considered after Benjamini–Hochberg correction. $^{*}$ $p < 0.05$, $^{**}$ $p < 0.01$, $^{***}$ $p < 0.001$. Non-significant cells are greyed out.

| | | ConvNeXT | RegNet | ResNet | ViT |
|---|---|---|---|---|---|
| **CIFAR-10** | **$\alpha$-ReQ** | $-0.469^{**}$ | -0.08933 | -0.1038 | -0.2193 |
| | **RankMe** | $0.7983^{**}$ | $0.779^{**}$ | $0.7668^{**}$ | $0.5649^{**}$ |
| | **RankMe$^{\star}$** | 0.1358 | $-0.3692^{*}$ | -0.3669 | 0.1175 |
| | **NE-Sum** | 0.2707 | $0.4884^{**}$ | 0.3144 | $0.3697^{*}$ |
| | **$\kappa$** | 0.3292 | $0.6438^{**}$ | $0.6204^{**}$ | $0.409^{**}$ |
| | **Self-Cluster** | 0.181 | $-0.5331^{**}$ | $-0.7515^{**}$ | -0.1731 |
| | **DSE** | -0.1278 | $0.5481^{**}$ | $0.7579^{**}$ | 0.1714 |
| | **ID** | $-0.5375^{**}$ | $-0.5716^{**}$ | -0.1554 | -0.2191 |
| **CIFAR-100** | **$\alpha$-ReQ** | $-0.5233^{**}$ | 0.08324 | -0.02837 | -0.1791 |
| | **RankMe** | $0.8189^{**}$ | $0.8952^{**}$ | $0.8425^{**}$ | $0.5985^{**}$ |
| | **RankMe$^{\star}$** | 0.2661 | $-0.5361^{**}$ | -0.4239 | 0.1086 |
| | **NE-Sum** | -0.2015 | $0.4731^{**}$ | $0.5276^{**}$ | $0.4809^{**}$ |
| | **$\kappa$** | $0.3774^{*}$ | $0.7608^{**}$ | $0.675^{**}$ | $0.3554^{*}$ |
| | **Self-Cluster** | 0.1316 | $-0.3331^{*}$ | $-0.7623^{**}$ | -0.143 |
| | **DSE** | -0.000555 | $0.3578^{**}$ | $0.7743^{**}$ | 0.1401 |
| | **ID** | $-0.8087^{**}$ | $-0.5637^{**}$ | -0.06056 | $-0.5633^{**}$ |
| **ImageNet$^{\ddagger}$** | **$\alpha$-ReQ** | -0.233 | $-0.3788^{**}$ | $-0.4998^{**}$ | 0.1701 |
| | **RankMe** | $0.3897^{*}$ | $0.7538^{**}$ | $0.6047^{**}$ | 0.1835 |
| | **RankMe$^{\star}$** | 0.1452 | 0.1151 | 0.136 | -0.1315 |
| | **NE-Sum** | -0.03091 | $0.5665^{**}$ | 0.2135 | 0.1395 |
| | **$\kappa$** | 0.006318 | $0.3854^{*}$ | 0.4237 | -0.238 |
| | **Self-Cluster** | $0.542^{**}$ | $-0.5665^{**}$ | $-0.5677^{**}$ | 0.1932 |
| | **DSE** | $-0.5304^{**}$ | $0.5717^{**}$ | $0.5864^{**}$ | -0.1783 |
| | **ID** | $-0.4361^{*}$ | $-0.64^{**}$ | $-0.4489^{**}$ | $-0.5089^{**}$ |
| **STL-10** | **$\alpha$-ReQ** | $-0.5577^{**}$ | 0.2052 | 0.1373 | -0.214 |
| | **RankMe** | $0.7421^{**}$ | $0.6964^{**}$ | 0.3914 | $0.4746^{**}$ |
| | **RankMe$^{\star}$** | 0.2709 | $-0.3829^{*}$ | -0.3397 | 0.1051 |
| | **NE-Sum** | -0.1921 | $0.7508^{**}$ | 0.192 | 0.3222 |
| | **$\kappa$** | 0.2363 | $0.5701^{**}$ | 0.4931 | 0.1615 |
| | **Self-Cluster** | 0.2918 | $-0.535^{**}$ | $-0.6102^{**}$ | -0.2333 |
| | **DSE** | -0.229 | $0.5592^{**}$ | $0.6037^{**}$ | 0.2262 |
| | **ID** | $-0.3375^{*}$ | $-0.6566^{**}$ | $-0.4953^{**}$ | $-0.4797^{**}$ |

$^{\ddagger}$ Correlations of ImageNet-1k should be interpreted with caution as it is used in the training pipeline of most backbones.

# E   The curious case of ViT models

Table 9: Spearman correlation ($\rho$) between representation quality metrics and OLS test accuracy, stratified by architecture class, across fine-grained datasets. Metrics are organised by construction: spectral (top), relation-based (middle), and manifold-based (bottom). Arrows indicate the preferred direction: $\uparrow$ (higher is better), $\downarrow$ (lower is better), $\rightarrow 1$ (closer to 1 is better). Statistical significance is considered after Benjamini–Hochberg correction. $^{*}$ $p < 0.05$, $^{**}$ $p < 0.01$, $^{***}$ $p < 0.001$. Non-significant cells are greyed out.

| | | ConvNeXT | RegNet | ResNet | ViT |
|---|---|---|---|---|---|
| **Food-101** | **$\alpha$-ReQ** | $-0.5099^{**}$ | $-0.1406$ | $-0.1182$ | $-0.1822$ |
| | **RankMe** | $0.8168^{**}$ | $0.874^{**}$ | $0.7219^{**}$ | $0.6275^{**}$ |
| | **RankMe$^{\star}$** | $0.2391$ | $-0.4341^{**}$ | $-0.4802^{*}$ | $0.08788$ |
| | **NE-Sum** | $0.6755^{**}$ | $0.5504^{**}$ | $0.6168^{**}$ | $0.6099^{**}$ |
| | **$\kappa$** | $0.1634$ | $0.6421^{**}$ | $0.585^{*}$ | $0.1812$ |
| | **Self-Cluster** | $-0.3117^{*}$ | $-0.272$ | $-0.6201^{**}$ | $-0.2651$ |
| | **DSE** | $0.3401^{*}$ | $0.2819^{*}$ | $0.6346^{**}$ | $0.2657$ |
| | **ID** | $-0.7834^{**}$ | $-0.2955$ | $0.2845$ | $-0.468^{**}$ |
| **Pets** | **$\alpha$-ReQ** | $-0.2543$ | $0.1161$ | $0.1713$ | $-0.01147$ |
| | **RankMe** | $0.5569^{**}$ | $0.4319^{**}$ | $0.1805$ | $0.3651^{*}$ |
| | **RankMe$^{\star}$** | $0.1412$ | $-0.2774$ | $-0.201$ | $-0.0166$ |
| | **NE-Sum** | $0.4825^{**}$ | $0.37^{*}$ | $0.09779$ | $0.4711^{**}$ |
| | **$\kappa$** | $0.1231$ | $0.3437^{**}$ | $0.1897$ | $-0.04254$ |
| | **Self-Cluster** | $-0.01793$ | $-0.192$ | $-0.5201^{**}$ | $-0.2375$ |
| | **DSE** | $0.031$ | $0.2065$ | $0.4999^{**}$ | $0.2509$ |
| | **ID** | $0.1293$ | $-0.3511^{**}$ | $-0.1962$ | $-0.495^{**}$ |
| **Flowers-102** | **$\alpha$-ReQ** | $0.1857$ | $0.3003$ | $0.03789$ | $0.4101^{*}$ |
| | **RankMe** | $0.3618^{**}$ | $0.703^{**}$ | $0.6503^{**}$ | $0.41^{**}$ |
| | **RankMe$^{\star}$** | $-0.3568^{**}$ | $-0.8161^{**}$ | $-0.5788^{**}$ | $-0.2895$ |
| | **NE-Sum** | $0.8002^{**}$ | $0.3632^{*}$ | $0.3015^{*}$ | $0.5721^{**}$ |
| | **$\kappa$** | $0.3483^{*}$ | $0.6188^{**}$ | $0.4646^{*}$ | $0.1629$ |
| | **Self-Cluster** | $-0.6228^{**}$ | $-0.09754$ | $-0.5086^{**}$ | $-0.3035$ |
| | **DSE** | $0.6254^{**}$ | $0.1111$ | $0.5277^{**}$ | $0.3038$ |
| | **ID** | $-0.9479^{**}$ | $-0.5794^{**}$ | $-0.6206^{**}$ | $-0.9146^{**}$ |

## F Other solvers

### F.1 SGD results

We freeze the backbone and train only the linear probe using SGD with a momentum of 0.9 and without weight decay, following standard practice for linear evaluation. We use Optuna Akiba et al. (2019) to find the optimal learning rate in the range $[1 \times 10^{-4}, 3 \times 10^{-1}]$, performing three rounds of hyper-parameter search to determine the final learning rate. We schedule the learning rate using a Cosine Scheduler and train for a total of 100 epochs. We perform $\ell_2$ normalisation on the features as is commonly done in foundation models (Balestriero et al. (2023)). We train only $\sim 120$ models, a subset of models in Tab. C for this analysis.

### F.2 Summary of the results

Table 10: Spearman correlation ($\rho$) between representation quality metrics and OLS test accuracy, stratified by architecture class, across specialised datasets. Metrics are organised by construction: spectral (top), relation-based (middle), and manifold-based (bottom). Arrows indicate the preferred direction: $\uparrow$ (higher is better), $\downarrow$ (lower is better), $\rightarrow 1$ (closer to 1 is better). Statistical significance is considered after Benjamini–Hochberg correction. $^{*}$ $p < 0.05$, $^{**}$ $p < 0.01$, $^{***}$ $p < 0.001$. Non-significant cells are greyed out.

| | | ConvNeXT | RegNet | ResNet | ViT |
|---|---|---|---|---|---|
| **DTD** | **α-ReQ** | $-0.4812^{**}$ | -0.181 | -0.153 | 0.03269 |
| | **RankMe** | $0.6054^{**}$ | 0.02952 | $-0.6883^{**}$ | $0.4478^{**}$ |
| | **RankMe$^{\star}$** | 0.1516 | -0.1294 | 0.4217 | -0.1118 |
| | **NE-Sum** | -0.1059 | -0.1169 | $-0.4944^{*}$ | $0.4943^{**}$ |
| | **κ** | $0.3355^{*}$ | 0.04234 | $-0.5242^{*}$ | 0.05048 |
| | **Self-Cluster** | 0.03795 | 0.2564 | $0.6252^{**}$ | -0.225 |
| | **DSE** | 0.1291 | -0.2608 | $-0.6339^{**}$ | 0.225 |
| | **ID** | $-0.3986^{*}$ | -0.0921 | -0.2663 | $-0.7644^{**}$ |
| **Places-365** | **α-ReQ** | $-0.6259^{**}$ | -0.08679 | -0.04473 | -0.1908 |
| | **RankMe** | $0.815^{**}$ | $0.9386^{**}$ | $0.8208^{**}$ | $0.6463^{**}$ |
| | **RankMe$^{\star}$** | $0.3893^{**}$ | -0.2845 | -0.3035 | 0.1025 |
| | **NE-Sum** | 0.04653 | $0.6026^{**}$ | $0.4664^{**}$ | $0.4416^{**}$ |
| | **κ** | 0.2575 | $0.7926^{**}$ | $0.6394^{**}$ | 0.2613 |
| | **Self-Cluster** | -0.08768 | $-0.2782^{*}$ | $-0.7027^{**}$ | -0.1927 |
| | **DSE** | 0.1902 | $0.2987^{**}$ | $0.7347^{**}$ | 0.1991 |
| | **ID** | -0.04636 | $-0.2563^{**}$ | 0.1671 | $-0.3843^{*}$ |
| **EuroSAT** | **α-ReQ** | -0.1854 | $0.5266^{**}$ | $0.5074^{**}$ | 0.1938 |
| | **RankMe** | $0.7766^{**}$ | $0.7987^{**}$ | $0.5526^{*}$ | $0.6053^{**}$ |
| | **RankMe$^{\star}$** | -0.3913 | $-0.8258^{**}$ | $-0.7666^{**}$ | $-0.3928^{**}$ |
| | **NE-Sum** | $-0.6439^{**}$ | -0.1598 | 0.2322 | -0.04633 |
| | **κ** | $0.7291^{**}$ | $0.759^{**}$ | $0.6948^{**}$ | $0.3927^{**}$ |
| | **Self-Cluster** | 0.02041 | 0.182 | -0.394 | -0.08389 |
| | **DSE** | -0.08056 | -0.1601 | 0.4104 | 0.0801 |
| | **ID** | $-0.5166^{**}$ | -0.2524 | -0.08959 | -0.05596 |

Table 11: Spearman correlation ($\rho$) between representation quality metrics and OLS test accuracy, stratified by training objective, across natural image datasets. Metrics are organised by construction: spectral (top), relation-based (middle), and manifold-based (bottom). Arrows indicate the preferred direction: ↑ (higher is better), ↓ (lower is better), → 1 (closer to 1 is better). Statistical significance is considered after Benjamini–Hochberg correction. $^{*}$ $p < 0.05$, $^{**}$ $p < 0.01$, $^{***}$ $p < 0.001$. Non-significant cells are greyed out.

| | | SSL | Supervised |
|---|---|---|---|
| **CIFAR-10** | **α-ReQ** (→ 1) | $-0.4703^{**}$ | $0.1877^{*}$ |
| | **RankMe** (↑ ) | 0.09054 | 0.1503 |
| | **RankMe$^{\star}$** (↑ ) | $0.4985^{**}$ | $-0.3683^{**}$ |
| | **NE-Sum** (↑ ) | $0.5942^{**}$ | 0.0295 |
| | **$\kappa$** (↑ ) | -0.06452 | 0.294 |
| | **Self-Cluster** (↓) | $-0.7885^{**}$ | 0.003576 |
| | **DSE** (↑ ) | $0.7903^{**}$ | 0.006315 |
| | **ID** (↓) | $-0.5249^{**}$ | $-0.5387^{**}$ |
| **CIFAR-100** | **α-ReQ** (→ 1) | $-0.6162^{**}$ | 0.1634 |
| | **RankMe** (↑ ) | 0.2258 | $0.3044^{*}$ |
| | **RankMe$^{\star}$** (↑ ) | $0.6103^{**}$ | $-0.3593^{**}$ |
| | **NE-Sum** (↑ ) | $0.6617^{**}$ | 0.1204 |
| | **$\kappa$** (↑ ) | -0.07808 | $0.3992^{**}$ |
| | **Self-Cluster** (↓) | $-0.8134^{**}$ | -0.02742 |
| | **DSE** (↑ ) | $0.8167^{**}$ | 0.04236 |
| | **ID** (↓) | $-0.6855^{**}$ | $-0.5586^{**}$ |
| **ImageNet$^{\ddagger}$** | **α-ReQ** (→ 1) | $-0.7038^{**}$ | -0.05376 |
| | **RankMe** (↑ ) | 0.331 | $0.2411^{*}$ |
| | **RankMe$^{\star}$** (↑ ) | $0.5033^{**}$ | -0.1564 |
| | **NE-Sum** (↑ ) | $0.4296^{*}$ | 0.08246 |
| | **$\kappa$** (↑ ) | 0.1554 | $0.4054^{**}$ |
| | **Self-Cluster** (↓) | $-0.7302^{**}$ | -0.1091 |
| | **DSE** (↑ ) | $0.7551^{**}$ | 0.1267 |
| | **ID** (↓) | $-0.4567^{**}$ | $-0.5948^{**}$ |
| **STL-10** | **α-ReQ** (→ 1) | $-0.3985^{*}$ | $0.3978^{**}$ |
| | **RankMe** (↑ ) | 0.01613 | 0.1485 |
| | **RankMe$^{\star}$** (↑ ) | $0.4637^{**}$ | $-0.4337^{**}$ |
| | **NE-Sum** (↑ ) | $0.5315^{**}$ | 0.06483 |
| | **$\kappa$** (↑ ) | -0.2863 | $0.4675^{**}$ |
| | **Self-Cluster** (↓) | $-0.8992^{**}$ | -0.02404 |
| | **DSE** (↑ ) | $0.9032^{**}$ | 0.02919 |
| | **ID** (↓) | $-0.7467^{**}$ | $-0.5336^{**}$ |

$^{\ddagger}$ Correlations of ImageNet-1k should be interpreted with caution as it is used in the training pipeline of most backbones.

Table 12: Spearman correlation ($\rho$) between representation quality metrics and OLS test accuracy, stratified by training objective, across fine-grained datasets. Metrics are organised by construction: spectral (top), relation-based (middle), and manifold-based (bottom). Arrows indicate the preferred direction: $\uparrow$ (higher is better), $\downarrow$ (lower is better), $\rightarrow 1$ (closer to 1 is better). Statistical significance is considered after Benjamini–Hochberg correction. $^{*}$ $p < 0.05$, $^{**}$ $p < 0.01$, $^{***}$ $p < 0.001$. Non-significant cells are greyed out.

| | | SSL | Supervised |
|---|---|---|---|
| **Food-101** | **$\alpha$-ReQ ($\rightarrow 1$)** | -0.07001 | -0.01501 |
| | **RankMe ($\uparrow$ )** | 0.1488 | 0.4572$^{**}$ |
| | **RankMe$^{\star}$ ($\uparrow$ )** | 0.2804 | -0.1865 |
| | **NE-Sum ($\uparrow$ )** | 0.7474$^{**}$ | 0.3605$^{**}$ |
| | **$\kappa$ ($\uparrow$ )** | -0.2929 | 0.3314$^{*}$ |
| | **Self-Cluster ($\downarrow$)** | $-0.9062^{**}$ | -0.1227 |
| | **DSE ($\uparrow$ )** | 0.9161$^{**}$ | 0.1318 |
| | **ID ($\downarrow$)** | $-0.7265^{**}$ | -0.2311 |
| **Pets** | **$\alpha$-ReQ ($\rightarrow 1$)** | 0.1952 | 0.2222$^{*}$ |
| | **RankMe ($\uparrow$ )** | -0.2189 | 0.2774$^{**}$ |
| | **RankMe$^{\star}$ ($\uparrow$ )** | 0.2348 | $-0.2586^{*}$ |
| | **NE-Sum ($\uparrow$ )** | 0.6003$^{**}$ | 0.1854 |
| | **$\kappa$ ($\uparrow$ )** | -0.2458 | 0.2018 |
| | **Self-Cluster ($\downarrow$)** | $-0.9266^{**}$ | -0.0395 |
| | **DSE ($\uparrow$ )** | 0.9141$^{**}$ | 0.04942 |
| | **ID ($\downarrow$)** | $-0.6692^{**}$ | $-0.2295^{*}$ |
| **Flowers-102** | **$\alpha$-ReQ ($\rightarrow 1$)** | 0.2881 | 0.261$^{*}$ |
| | **RankMe ($\uparrow$ )** | -0.1755 | 0.1756 |
| | **RankMe$^{\star}$ ($\uparrow$ )** | 0.01852 | -0.2528 |
| | **NE-Sum ($\uparrow$ )** | 0.887$^{**}$ | 0.2883$^{**}$ |
| | **$\kappa$ ($\uparrow$ )** | -0.2663 | 0.1992 |
| | **Self-Cluster ($\downarrow$)** | $-0.8921^{**}$ | -0.02264 |
| | **DSE ($\uparrow$ )** | 0.9003$^{**}$ | 0.02849 |
| | **ID ($\downarrow$)** | $-0.882^{**}$ | $-0.6634^{**}$ |

Table 13: Spearman correlation ($\rho$) between representation quality metrics and OLS test accuracy, stratified by training objective, across specialised datasets. Metrics are organised by construction: spectral (top), relation-based (middle), and manifold-based (bottom). Arrows indicate the preferred direction: $\uparrow$ (higher is better), $\downarrow$ (lower is better), $\rightarrow 1$ (closer to 1 is better). Statistical significance is considered after Benjamini–Hochberg correction. $^{*}\ p < 0.05$, $^{**}\ p < 0.01$, $^{***}\ p < 0.001$. Non-significant cells are greyed out.

| | | SSL | Supervised |
|---|---|---|---|
| **DTD** | **$\alpha$-ReQ** ($\rightarrow$ 1) | $-0.5796^{**}$ | 0.07191 |
| | **RankMe** ($\uparrow$ ) | -0.1747 | $-0.4853^{**}$ |
| | **RankMe$^{\star}$** ($\uparrow$ ) | $0.6599^{**}$ | 0.3193 |
| | **NE-Sum** ($\uparrow$ ) | $0.4883^{**}$ | $-0.4066^{**}$ |
| | **$\kappa$** ($\uparrow$ ) | $-0.448^{**}$ | -0.3383 |
| | **Self-Cluster** ($\downarrow$) | $-0.6756^{**}$ | $0.3718^{**}$ |
| | **DSE** ($\uparrow$ ) | $0.678^{**}$ | $-0.377^{**}$ |
| | **ID** ($\downarrow$) | $-0.4645^{**}$ | -0.1372 |
| **Places-365** | **$\alpha$-ReQ** ($\rightarrow$ 1) | $-0.706^{**}$ | -0.03565 |
| | **RankMe** ($\uparrow$ ) | $0.4084^{*}$ | $0.5174^{**}$ |
| | **RankMe$^{\star}$** ($\uparrow$ ) | $0.5587^{**}$ | -0.1519 |
| | **NE-Sum** ($\uparrow$ ) | $0.5161^{**}$ | 0.2198 |
| | **$\kappa$** ($\uparrow$ ) | -0.1756 | $0.4554^{**}$ |
| | **Self-Cluster** ($\downarrow$) | $-0.6294^{**}$ | -0.1524 |
| | **DSE** ($\uparrow$ ) | $0.7419^{**}$ | 0.1696 |
| | **ID** ($\downarrow$) | $-0.5183^{**}$ | -0.1054 |
| **EuroSAT** | **$\alpha$-ReQ** ($\rightarrow$ 1) | -0.2117 | $0.3385^{**}$ |
| | **RankMe** ($\uparrow$ ) | $0.6216^{**}$ | $0.4761^{**}$ |
| | **RankMe$^{\star}$** ($\uparrow$ ) | -0.0009171 | $-0.5907^{**}$ |
| | **NE-Sum** ($\uparrow$ ) | $0.37^{*}$ | $-0.2768^{*}$ |
| | **$\kappa$** ($\uparrow$ ) | $0.3822^{**}$ | $0.5308^{**}$ |
| | **Self-Cluster** ($\downarrow$) | -0.246 | 0.0915 |
| | **DSE** ($\uparrow$ ) | 0.2825 | -0.09816 |
| | **ID** ($\downarrow$) | 0.2788 | -0.1916 |

Table 14: Spearman correlation ($\rho$) between representation quality metrics and OLS test accuracy, stratified by representation dimension $d$, across natural image datasets. Metrics are organised by construction: spectral (top), relation-based (middle), and manifold-based (bottom). Arrows indicate the preferred direction: $\uparrow$ (higher is better), $\downarrow$ (lower is better), $\to 1$ (closer to 1 is better). Statistical significance is considered after Benjamini–Hochberg correction. * $p < 0.05$, ** $p < 0.01$, *** $p < 0.001$. Non-significant cells are greyed out.

| | | $d \leq 520$ | $520 < d \leq 960$ | $960 < d \leq 1792$ | $d > 1792$ |
|---|---|---|---|---|---|
| CIFAR-10 | $\boldsymbol{\alpha}$-**ReQ** ($\to 1$) | 0.1734 | -0.1571 | 0.04738 | 0.1315 |
| | **RankMe** ($\uparrow$) | -0.3555 | -0.02149 | -0.1327 | 0.4376 |
| | **RankMe*** ($\uparrow$) | -0.3226 | 0.08077 | -0.03157 | -0.2487 |
| | **NE-Sum** ($\uparrow$) | -0.2765 | 0.1083 | 0.4189* | 0.1896 |
| | $\boldsymbol{\kappa}$ ($\uparrow$) | 0.4829* | 0.1335 | 0.04447 | 0.04695 |
| | **Self-Cluster** ($\downarrow$) | 0.3405 | 0.03767 | $-0.583$* | -0.1373 |
| | **DSE** ($\uparrow$) | -0.3379 | -0.0316 | 0.5723* | 0.1593 |
| | **ID** ($\downarrow$) | $-0.578$* | -0.1895 | $-0.8023$* | $-0.6649$* |
| CIFAR-100 | $\boldsymbol{\alpha}$-**ReQ** ($\to 1$) | 0.1731 | -0.09751 | -0.008382 | 0.1117 |
| | **RankMe** ($\uparrow$) | -0.2754 | -0.03186 | -0.05085 | 0.5141* |
| | **RankMe*** ($\uparrow$) | -0.2871 | 0.0889 | 0.04651 | -0.2368 |
| | **NE-Sum** ($\uparrow$) | -0.101 | 0.1313 | 0.1457 | 0.1815 |
| | $\boldsymbol{\kappa}$ ($\uparrow$) | 0.5506* | 0.133 | 0.1513 | 0.03697 |
| | **Self-Cluster** ($\downarrow$) | 0.3401 | 0.01835 | $-0.5816$* | -0.0354 |
| | **DSE** ($\uparrow$) | -0.3344 | -0.01253 | 0.587* | 0.06931 |
| | **ID** ($\downarrow$) | $-0.6787$* | $-0.4224$* | $-0.8049$* | $-0.6618$* |
| ImageNet‡ | $\boldsymbol{\alpha}$-**ReQ** ($\to 1$) | 0.3707* | -0.06594 | -0.1894 | $-0.4712$* |
| | **RankMe** ($\uparrow$) | -0.1344 | -0.001592 | 0.0307 | 0.4758* |
| | **RankMe*** ($\uparrow$) | $-0.436$* | 0.02231 | 0.05674 | 0.1557 |
| | **NE-Sum** ($\uparrow$) | -0.2037 | -0.01923 | 0.08408 | 0.3476* |
| | $\boldsymbol{\kappa}$ ($\uparrow$) | 0.234 | 0.04222 | 0.02922 | -0.01129 |
| | **Self-Cluster** ($\downarrow$) | 0.2987 | 0.1644 | -0.1078 | $-0.4651$* |
| | **DSE** ($\uparrow$) | -0.2758 | -0.1461 | 0.1191 | 0.4824* |
| | **ID** ($\downarrow$) | $-0.5015$* | $-0.4952$* | $-0.8368$* | $-0.6808$* |
| STL-10 | $\boldsymbol{\alpha}$-**ReQ** ($\to 1$) | 0.2643 | -0.09379 | 0.1807 | 0.2877 |
| | **RankMe** ($\uparrow$) | -0.2544 | 0.07278 | -0.1987 | 0.1844 |
| | **RankMe*** ($\uparrow$) | -0.3679 | 0.1488 | -0.09002 | -0.3022 |
| | **NE-Sum** ($\uparrow$) | -0.2053 | 0.1135 | 0.06434 | 0.2372 |
| | $\boldsymbol{\kappa}$ ($\uparrow$) | 0.5* | 0.04954 | 0.2663 | 0.3911* |
| | **Self-Cluster** ($\downarrow$) | 0.2748 | -0.07069 | $-0.4514$* | $-0.3415$* |
| | **DSE** ($\uparrow$) | -0.2678 | 0.08441 | 0.4449* | 0.3345* |
| | **ID** ($\downarrow$) | -0.361 | -0.3003 | $-0.7833$* | $-0.7299$* |

‡ Correlations of ImageNet-1k should be interpreted with caution as it is used in the training pipeline of most backbones.

Table 15: Spearman correlation ($\rho$) between representation quality metrics and OLS test accuracy, stratified by representation dimension $d$, across fine-grained datasets. Metrics are organised by construction: spectral (top), relation-based (middle), and manifold-based (bottom). Arrows indicate the preferred direction: $\uparrow$ (higher is better), $\downarrow$ (lower is better), $\rightarrow 1$ (closer to 1 is better). Statistical significance is considered after Benjamini–Hochberg correction. $^{*}\ p < 0.05$, $^{**}\ p < 0.01$, $^{***}\ p < 0.001$. Non-significant cells are greyed out.

| | | $d \leq 520$ | $520 < d \leq 960$ | $960 < d \leq 1792$ | $d > 1792$ |
|---|---|---|---|---|---|
| **Food-101** | **α-ReQ** ($\rightarrow 1$) | 0.2028 | $-0.446^{*}$ | 0.1216 | 0.2136 |
| | **RankMe** ($\uparrow$) | -0.05324 | 0.3067 | -0.1193 | 0.4383 |
| | **RankMe$^{*}$** ($\uparrow$) | -0.1871 | $0.4453^{*}$ | -0.08247 | -0.2668 |
| | **NE-Sum** ($\uparrow$) | 0.08391 | $0.4844^{*}$ | 0.1824 | 0.3293 |
| | **$\kappa$** ($\uparrow$) | $0.4579^{*}$ | -0.002327 | 0.1213 | -0.1018 |
| | **Self-Cluster** ($\downarrow$) | 0.2196 | -0.1607 | $-0.5652^{*}$ | -0.1326 |
| | **DSE** ($\uparrow$) | -0.2184 | 0.1631 | $0.5518^{*}$ | 0.1297 |
| | **ID** ($\downarrow$) | -0.197 | -0.3246 | $-0.6238^{*}$ | -0.2413 |
| **Pets** | **α-ReQ** ($\rightarrow 1$) | $0.3928^{*}$ | -0.1464 | 0.1566 | 0.2665 |
| | **RankMe** ($\uparrow$) | 0.03452 | 0.1881 | -0.06354 | -0.009081 |
| | **RankMe$^{*}$** ($\uparrow$) | $-0.3774^{*}$ | 0.2862 | -0.04897 | -0.1598 |
| | **NE-Sum** ($\uparrow$) | 0.07808 | $0.4522^{*}$ | 0.1765 | -0.2438 |
| | **$\kappa$** ($\uparrow$) | $0.3891^{*}$ | -0.1584 | 0.02108 | -0.02806 |
| | **Self-Cluster** ($\downarrow$) | -0.04736 | -0.1164 | -0.3369 | 0.02313 |
| | **DSE** ($\uparrow$) | 0.04551 | 0.128 | 0.3246 | -0.03073 |
| | **ID** ($\downarrow$) | -0.2462 | -0.04858 | $-0.5503^{*}$ | -0.3249 |
| **Flowers-102** | **α-ReQ** ($\rightarrow 1$) | $0.5289^{*}$ | 0.047 | $0.4831^{*}$ | 0.3041 |
| | **RankMe** ($\uparrow$) | -0.3579 | -0.04978 | $-0.3645^{*}$ | 0.2327 |
| | **RankMe$^{*}$** ($\uparrow$) | -0.3004 | 0.1655 | -0.2058 | $-0.5056^{*}$ |
| | **NE-Sum** ($\uparrow$) | -0.05045 | $0.6226^{*}$ | 0.3139 | 0.3522 |
| | **$\kappa$** ($\uparrow$) | 0.3044 | -0.2483 | -0.009413 | $-0.4847^{*}$ |
| | **Self-Cluster** ($\downarrow$) | 0.12 | -0.2458 | $-0.5228^{*}$ | -0.1166 |
| | **DSE** ($\uparrow$) | -0.1175 | 0.2443 | $0.5151^{*}$ | 0.1355 |
| | **ID** ($\downarrow$) | $-0.5277^{*}$ | $-0.8358^{*}$ | $-0.7534^{*}$ | $-0.5133^{*}$ |

Table 16: Spearman correlation ($\rho$) between representation quality metrics and OLS test accuracy, stratified by representation dimension $d$, across specialised datasets. Metrics are organised by construction: spectral (top), relation-based (middle), and manifold-based (bottom). Arrows indicate the preferred direction: $\uparrow$ (higher is better), $\downarrow$ (lower is better), $\to 1$ (closer to 1 is better). Statistical significance is considered after Benjamini–Hochberg correction. $^{*}$ $p < 0.05$, $^{**}$ $p < 0.01$, $^{***}$ $p < 0.001$. Non-significant cells are greyed out.

| | | $d \leq 520$ | $520 < d \leq 960$ | $960 < d \leq 1792$ | $d > 1792$ |
|---|---|---|---|---|---|
| **DTD** | $\alpha$-**ReQ** ($\to 1$) | 0.3104 | -0.276 | 0.08673 | 0.2104 |
| | **RankMe** ($\uparrow$) | -0.1474 | -0.144 | $-0.7875^{*}$ | 0.06973 |
| | **RankMe**$^{*}$ ($\uparrow$) | -0.3189 | 0.3309 | 0.1021 | $-0.8435^{*}$ |
| | **NE-Sum** ($\uparrow$) | -0.2004 | 0.01337 | $-0.3821^{*}$ | -0.2098 |
| | $\kappa$ ($\uparrow$) | $0.4092^{*}$ | -0.0831 | -0.05862 | 0.4448 |
| | **Self-Cluster** ($\downarrow$) | 0.2162 | 0.09457 | -0.07227 | 0.2249 |
| | **DSE** ($\uparrow$) | -0.2199 | -0.05258 | 0.07967 | -0.2254 |
| | **ID** ($\downarrow$) | -0.1627 | 0.05697 | -0.4907 | -0.2296 |
| **Places-365** | $\alpha$-**ReQ** ($\to 1$) | 0.02679 | -0.1881 | -0.2842 | -0.1931 |
| | **RankMe** ($\uparrow$) | 0.113 | -0.06566 | -0.0004231 | $0.632^{*}$ |
| | **RankMe**$^{*}$ ($\uparrow$) | -0.196 | 0.2073 | 0.2801 | 0.1082 |
| | **NE-Sum** ($\uparrow$) | -0.1816 | 0.1924 | 0.1249 | $0.4749^{*}$ |
| | $\kappa$ ($\uparrow$) | $0.436^{*}$ | 0.004077 | 0.2508 | 0.09826 |
| | **Self-Cluster** ($\downarrow$) | 0.2627 | -0.07105 | $-0.6344^{*}$ | -0.2096 |
| | **DSE** ($\uparrow$) | -0.2617 | 0.08256 | $0.6381^{*}$ | 0.2543 |
| | **ID** ($\downarrow$) | -0.1479 | 0.01832 | $-0.4577^{*}$ | -0.3163 |
| **EuroSAT** | $\alpha$-**ReQ** ($\to 1$) | 0.2226 | -0.03631 | $0.6747^{*}$ | $0.5483^{*}$ |
| | **RankMe** ($\uparrow$) | -0.04265 | -0.2769 | $-0.567^{*}$ | 0.04986 |
| | **RankMe**$^{*}$ ($\uparrow$) | -0.3263 | 0.05031 | $-0.6929^{*}$ | $-0.5732^{*}$ |
| | **NE-Sum** ($\uparrow$) | $-0.3559^{*}$ | $-0.355^{*}$ | $-0.7209^{*}$ | $-0.4477^{*}$ |
| | $\kappa$ ($\uparrow$) | $0.6135^{*}$ | 0.07628 | $0.4884^{*}$ | 0.09337 |
| | **Self-Cluster** ($\downarrow$) | 0.3065 | 0.4001 | 0.2651 | $0.5282^{*}$ |
| | **DSE** ($\uparrow$) | -0.3239 | -0.4024 | -0.2853 | $-0.5329^{*}$ |
| | **ID** ($\downarrow$) | -0.1074 | $0.349^{*}$ | $-0.6469^{*}$ | $-0.5237^{*}$ |

Table 17: Spearman correlation ($\rho$) between representation quality metrics and KNN test accuracy ($k = 1$) across 10 evaluation datasets, grouped by task type. Metrics are organised by construction: spectral (top), relation-based (middle), and manifold-based (bottom). Statistical significance is considered after Benjamini–Hochberg correction. $^{*}$ $p < 0.05$, $^{**}$ $p < 0.01$, $^{***}$ $p < 0.001$. Non-significant cells are greyed out.

(a) Generic Objection Detection

| | CIFAR-10 | CIFAR-100 | ImageNet$^{\ddagger}$ | STL-10 |
|---|---|---|---|---|
| $\alpha < 1.0$ | $-0.5275^{*}$ | $-0.5769^{**}$ | $-0.4409^{**}$ | 0.5714 |
| $\alpha \geq 1.0$ | -0.07735 | $-0.2066^{**}$ | 0.2081 | 0.09299 |
| $\alpha$-ReQ | $-0.1678^{*}$ | $-0.2962^{**}$ | $-0.3546^{**}$ | 0.1224 |
| RankMe | $0.2609^{*}$ | $0.3398^{**}$ | $0.3726^{**}$ | $0.2944^{**}$ |
| RankMe$^{\star}$ | 0.03976 | 0.1439 | $0.1961^{*}$ | -0.156 |
| NE-Sum | $0.5004^{**}$ | $0.4971^{**}$ | $0.3962^{**}$ | $0.4346^{**}$ |
| $\kappa$ | -0.003066 | 0.04084 | -0.03046 | 0.05982 |
| Self-Cluster | $-0.5028^{**}$ | $-0.5331^{**}$ | $-0.4385^{**}$ | $-0.4942^{**}$ |
| DSE | $0.5098^{**}$ | $0.5487^{**}$ | $0.4505^{**}$ | $0.4977^{**}$ |
| ID | $-0.5002^{**}$ | $-0.5502^{**}$ | $-0.6739^{**}$ | $-0.6419^{**}$ |

(b) Fine-grained Objection Detection

| | Food-101 | Pets | Flowers-102 |
|---|---|---|---|
| $\alpha < 1.0$ | – | – | – |
| $\alpha \geq 1.0$ | -0.07289 | 0.1052 | $0.1727^{*}$ |
| $\alpha$-ReQ | -0.06906 | 0.1052 | $0.1727^{*}$ |
| RankMe | $0.4698^{**}$ | $0.342^{**}$ | $0.3069^{**}$ |
| RankMe$^{\star}$ | -0.05627 | -0.1124 | $-0.2627^{**}$ |
| NE-Sum | $0.5801^{**}$ | $0.4591^{**}$ | $0.6028^{**}$ |
| $\kappa$ | 0.02121 | -0.03045 | $0.1905^{*}$ |
| Self-Cluster | $-0.5826^{**}$ | $-0.5355^{**}$ | $-0.5452^{**}$ |
| DSE | $0.5921^{**}$ | $0.5366^{**}$ | $0.5533^{**}$ |
| ID | $-0.3688^{**}$ | $-0.4847^{**}$ | $-0.6896^{**}$ |

(c) Specialised

| | DTD | Places-365 | EuroSAT |
|---|---|---|---|
| $\alpha < 1.0$ | – | $-0.3284^{**}$ | – |
| $\alpha \geq 1.0$ | $-0.2585^{**}$ | $-0.2072^{**}$ | -0.04101 |
| $\alpha$-ReQ | $-0.2617^{**}$ | $-0.4247^{**}$ | -0.04101 |
| RankMe | $0.53^{**}$ | $0.5385^{**}$ | $0.5585^{**}$ |
| RankMe$^{\star}$ | -0.1564 | $0.2843^{**}$ | $-0.2179^{*}$ |
| NE-Sum | $0.3943^{**}$ | $0.539^{**}$ | 0.03467 |
| $\kappa$ | $0.2942^{**}$ | 0.03862 | $0.3332^{**}$ |
| Self-Cluster | $-0.5034^{**}$ | $-0.5969^{**}$ | $-0.3621^{**}$ |
| DSE | $0.5252^{**}$ | $0.6109^{**}$ | $0.3484^{**}$ |
| ID | $-0.3101^{**}$ | $-0.2109^{**}$ | $-0.2516^{**}$ |

$^{\ddagger}$ Correlations of ImageNet-1k should be interpreted with caution as it is used in the training pipeline of most backbones

Table 18: Spearman correlation ($\rho$) between representation quality metrics and KNN test accuracy ($k = 1$), stratified by architecture class, across natural image datasets. Metrics are organised by construction: spectral (top), relation-based (middle), and manifold-based (bottom). Statistical significance is considered after Benjamini–Hochberg correction. $^{*}$ $p < 0.05$, $^{**}$ $p < 0.01$, $^{***}$ $p < 0.001$. Non-significant cells are greyed out.

| | | ConvNeXT | RegNet | ResNet | ViT |
|---|---|---|---|---|---|
| **CIFAR-10** | **$\alpha$-ReQ** | $-0.5643^{**}$ | -0.227 | -0.2629 | -0.2075 |
| | **RankMe** | $0.7738^{**}$ | $0.6102^{**}$ | $0.5309^{*}$ | $0.5369^{**}$ |
| | **RankMe$^\star$** | 0.2589 | -0.1931 | -0.05598 | 0.2075 |
| | **NE-Sum** | $0.3157^{*}$ | $0.6241^{**}$ | 0.3101 | $0.6791^{**}$ |
| | **$\kappa$** | 0.2408 | $0.4666^{**}$ | 0.3615 | 0.2019 |
| | **Self-Cluster** | 0.1193 | $-0.6115^{**}$ | $-0.5767^{**}$ | $-0.8651^{**}$ |
| | **DSE** | -0.06232 | $0.6234^{**}$ | $0.5789^{**}$ | $0.8677^{**}$ |
| | **ID** | $-0.4905^{**}$ | $-0.5217^{**}$ | -0.3566 | $-0.5324^{**}$ |
| **CIFAR-100** | **$\alpha$-ReQ** | $-0.614^{**}$ | -0.2094 | -0.197 | -0.294 |
| | **RankMe** | $0.7737^{**}$ | $0.62^{**}$ | $0.7421^{**}$ | $0.5785^{**}$ |
| | **RankMe$^\star$** | $0.4025^{**}$ | -0.1887 | -0.1931 | 0.3015 |
| | **NE-Sum** | -0.122 | $0.801^{**}$ | $0.5692^{**}$ | $0.6242^{**}$ |
| | **$\kappa$** | 0.2709 | $0.4597^{**}$ | $0.5138^{*}$ | 0.2394 |
| | **Self-Cluster** | 0.04559 | $-0.5949^{**}$ | $-0.7037^{**}$ | $-0.8853^{**}$ |
| | **DSE** | 0.08999 | $0.608^{**}$ | $0.7154^{**}$ | $0.8893^{**}$ |
| | **ID** | $-0.8036^{**}$ | $-0.6186^{**}$ | -0.1639 | $-0.6559^{**}$ |
| **ImageNet$^{\ddagger}$** | **$\alpha$-ReQ** | -0.36 | $-0.3159^{**}$ | $-0.4293^{**}$ | $-0.4528^{**}$ |
| | **RankMe** | $0.4813^{**}$ | $0.7323^{**}$ | $0.5134^{*}$ | $0.5553^{**}$ |
| | **RankMe$^\star$** | 0.2595 | 0.06456 | 0.07582 | $0.3788^{**}$ |
| | **NE-Sum** | -0.151 | $0.5263^{**}$ | 0.1835 | $0.6805^{**}$ |
| | **$\kappa$** | 0.04089 | $0.398^{**}$ | 0.3947 | 0.0813 |
| | **Self-Cluster** | $0.4743^{**}$ | $-0.4977^{**}$ | $-0.5311^{*}$ | $-0.8353^{**}$ |
| | **DSE** | $-0.4494^{*}$ | $0.5058^{**}$ | $0.5457^{*}$ | $0.8419^{**}$ |
| | **ID** | -0.3073 | $-0.6361^{**}$ | $-0.4833^{**}$ | $-0.6616^{**}$ |
| **STL-10** | **$\alpha$-ReQ** | $-0.5858^{**}$ | 0.2031 | 0.1792 | -0.08643 |
| | **RankMe** | $0.6397^{**}$ | $0.7175^{**}$ | 0.2811 | $0.5463^{**}$ |
| | **RankMe$^\star$** | $0.3826^{**}$ | $-0.3868^{*}$ | -0.3117 | 0.1093 |
| | **NE-Sum** | -0.2151 | $0.7712^{**}$ | 0.1714 | $0.677^{**}$ |
| | **$\kappa$** | 0.03005 | $0.5895^{**}$ | 0.4051 | -0.1812 |
| | **Self-Cluster** | 0.2787 | $-0.5917^{**}$ | $-0.5464^{*}$ | $-0.8447^{**}$ |
| | **DSE** | -0.2166 | $0.616^{**}$ | $0.5392^{*}$ | $0.8504^{**}$ |
| | **ID** | -0.3028 | $-0.6716^{**}$ | $-0.5773^{**}$ | $-0.7826^{**}$ |

$^{\ddagger}$ Correlations of ImageNet-1k should be interpreted with caution as it is used in the training pipeline of most backbones.

Table 19: Spearman correlation ($\rho$) between representation quality metrics and KNN test accuracy ($k = 1$), stratified by architecture class, across fine-grained datasets. Metrics are organised by construction: spectral (top), relation-based (middle), and manifold-based (bottom). Statistical significance is considered after Benjamini–Hochberg correction. $^{*}\ p < 0.05$, $^{**}\ p < 0.01$, $^{***}\ p < 0.001$. Non-significant cells are greyed out.

| | | ConvNeXT | RegNet | ResNet | ViT |
|---|---|---|---|---|---|
| **Food-101** | **$\alpha$-ReQ** | $-0.5981^{**}$ | $-0.2761^{*}$ | -0.1346 | 0.1837 |
| | **RankMe** | $0.7632^{**}$ | $0.8486^{**}$ | $0.6684^{**}$ | $0.5275^{**}$ |
| | **RankMe$^{\star}$** | $0.3745^{**}$ | $-0.3125^{*}$ | $-0.4269^{*}$ | -0.1435 |
| | **NE-Sum** | $0.8039^{**}$ | $0.6442^{**}$ | $0.6426^{**}$ | $0.5705^{**}$ |
| | **$\kappa$** | 0.0578 | $0.6323^{**}$ | $0.5163^{*}$ | -0.1739 |
| | **Self-Cluster** | $-0.3704^{**}$ | $-0.3716^{*}$ | $-0.6934^{**}$ | $-0.914^{**}$ |
| | **DSE** | $0.4054^{**}$ | $0.3849^{**}$ | $0.7108^{**}$ | $0.916^{**}$ |
| | **ID** | $-0.7347^{**}$ | -0.2977 | 0.3305 | $-0.7847^{**}$ |
| **Pets** | **$\alpha$-ReQ** | $-0.3112^{*}$ | 0.1244 | 0.1487 | 0.2569 |
| | **RankMe** | $0.5143^{**}$ | $0.4701^{**}$ | 0.1991 | $0.3835^{*}$ |
| | **RankMe$^{\star}$** | 0.2484 | -0.2805 | -0.2059 | -0.1994 |
| | **NE-Sum** | $0.5999^{**}$ | $0.4746^{**}$ | 0.1302 | $0.4386^{*}$ |
| | **$\kappa$** | -0.004611 | $0.4258^{**}$ | 0.1883 | -0.009377 |
| | **Self-Cluster** | -0.0742 | $-0.4092^{**}$ | $-0.6065^{**}$ | $-0.869^{**}$ |
| | **DSE** | 0.08312 | $0.4177^{**}$ | $0.5795^{**}$ | $0.8732^{**}$ |
| | **ID** | 0.2077 | $-0.5766^{**}$ | -0.2008 | $-0.8571^{**}$ |
| **Flowers-102** | **$\alpha$-ReQ** | 0.06719 | $0.3443^{*}$ | -0.02418 | $0.5014^{**}$ |
| | **RankMe** | $0.4645^{**}$ | $0.5307^{**}$ | $0.5967^{**}$ | $0.3787^{*}$ |
| | **RankMe$^{\star}$** | $-0.3425^{*}$ | $-0.5689^{**}$ | -0.4193 | $-0.3351^{*}$ |
| | **NE-Sum** | $0.805^{**}$ | $0.6244^{**}$ | $0.4333^{**}$ | $0.492^{**}$ |
| | **$\kappa$** | $0.4115^{**}$ | $0.3679^{**}$ | 0.3384 | 0.1936 |
| | **Self-Cluster** | $-0.628^{**}$ | 0.02899 | $-0.5656^{**}$ | $-0.9313^{**}$ |
| | **DSE** | $0.6327^{**}$ | -0.01046 | $0.58^{**}$ | $0.9339^{**}$ |
| | **ID** | $-0.9752^{**}$ | $-0.5583^{**}$ | $-0.5375^{**}$ | $-0.7505^{**}$ |

Table 20: Spearman correlation ($\rho$) between representation quality metrics and KNN test accuracy ($k = 1$), stratified by architecture class, across specialised datasets. Metrics are organised by construction: spectral (top), relation-based (middle), and manifold-based (bottom). Statistical significance is considered after Benjamini–Hochberg correction. $^{*}$ $p < 0.05$, $^{**}$ $p < 0.01$, $^{***}$ $p < 0.001$. Non-significant cells are greyed out.

| | | ConvNeXT | RegNet | ResNet | ViT |
|---|---|---|---|---|---|
| **DTD** | $\boldsymbol{\alpha}$**-ReQ** | $-0.5196^{**}$ | 0.03956 | 0.2612 | -0.2834 |
| | **RankMe** | $0.7614^{**}$ | $0.8254^{**}$ | $0.6813^{**}$ | $0.4636^{**}$ |
| | **RankMe$^{\star}$** | 0.1018 | $-0.7366^{**}$ | $-0.6649^{**}$ | 0.1615 |
| | **NE-Sum** | -0.2564 | $0.5682^{**}$ | $0.5334^{**}$ | $0.6393^{**}$ |
| | $\boldsymbol{\kappa}$ | $0.4526^{**}$ | $0.7614^{**}$ | $0.6288^{*}$ | -0.09476 |
| | **Self-Cluster** | 0.01003 | -0.2031 | $-0.6567^{**}$ | $-0.9018^{**}$ |
| | **DSE** | 0.165 | $0.2462^{*}$ | $0.6748^{**}$ | $0.9026^{**}$ |
| | **ID** | $-0.4767^{**}$ | -0.005594 | 0.1137 | $-0.7173^{**}$ |
| **Places-365** | $\boldsymbol{\alpha}$**-ReQ** | $-0.6867^{**}$ | -0.2348 | 0.08313 | -0.3218 |
| | **RankMe** | $0.7415^{**}$ | $0.8783^{**}$ | $0.7036^{**}$ | $0.571^{**}$ |
| | **RankMe$^{\star}$** | $0.519^{**}$ | -0.1678 | -0.3772 | $0.3264^{*}$ |
| | **NE-Sum** | 0.1597 | $0.5925^{**}$ | $0.2966^{*}$ | $0.6714^{**}$ |
| | $\boldsymbol{\kappa}$ | 0.1592 | $0.754^{**}$ | $0.6405^{**}$ | -0.1803 |
| | **Self-Cluster** | -0.1646 | $-0.3782^{**}$ | $-0.7112^{**}$ | $-0.9145^{**}$ |
| | **DSE** | 0.2614 | $0.4003^{**}$ | $0.7278^{**}$ | $0.9185^{**}$ |
| | **ID** | 0.02254 | -0.2187 | 0.2459 | $-0.5505^{**}$ |
| **EuroSAT** | $\boldsymbol{\alpha}$**-ReQ** | $-0.4338^{**}$ | $0.4307^{**}$ | $0.4052^{**}$ | -0.1318 |
| | **RankMe** | $0.6431^{**}$ | $0.3518^{**}$ | $0.4698^{*}$ | $0.4742^{**}$ |
| | **RankMe$^{\star}$** | -0.06425 | $-0.5506^{**}$ | $-0.6164^{**}$ | 0.06468 |
| | **NE-Sum** | $-0.6319^{**}$ | -0.3014 | 0.1316 | $0.4889^{**}$ |
| | $\boldsymbol{\kappa}$ | $0.5129^{**}$ | $0.3792^{**}$ | $0.6073^{**}$ | 0.2751 |
| | **Self-Cluster** | -0.1829 | $0.2971^{*}$ | $-0.3721^{*}$ | $-0.9435^{**}$ |
| | **DSE** | 0.1159 | $-0.295^{*}$ | $0.3898^{*}$ | $0.9438^{**}$ |
| | **ID** | $-0.5287^{**}$ | $-0.3289^{**}$ | -0.03897 | $-0.3665^{*}$ |

Table 21: Spearman correlation ($\rho$) between representation quality metrics and KNN test accuracy ($k = 1$), stratified by training objective, across natural image datasets. Metrics are organised by construction: spectral (top), relation-based (middle), and manifold-based (bottom). Statistical significance is considered after Benjamini–Hochberg correction. $^{*}$ $p < 0.05$, $^{**}$ $p < 0.01$, $^{***}$ $p < 0.001$. Non-significant cells are greyed out.

| | | SSL | Supervised |
|---|---|---|---|
| **CIFAR-10** | **α-ReQ** | $-0.5896^{**}$ | $-0.05143$ |
| | **RankMe** | $0.2196$ | $0.3681^{**}$ |
| | **RankMe$^{\star}$** | $0.6715^{**}$ | $-0.1566$ |
| | **NE-Sum** | $0.754^{**}$ | $0.4138^{**}$ |
| | **$\kappa$** | $-0.01451$ | $0.03192$ |
| | **Self-Cluster** | $-0.9059^{**}$ | $-0.406^{**}$ |
| | **DSE** | $0.9089^{**}$ | $0.4166^{**}$ |
| | **ID** | $-0.4521^{**}$ | $-0.4649^{**}$ |
| **CIFAR-100** | **α-ReQ** | $-0.7647^{**}$ | $-0.164^{*}$ |
| | **RankMe** | $0.273$ | $0.4478^{**}$ |
| | **RankMe$^{\star}$** | $0.8011^{**}$ | $-0.06249$ |
| | **NE-Sum** | $0.776^{**}$ | $0.4557^{**}$ |
| | **$\kappa$** | $-0.06127$ | $0.04207$ |
| | **Self-Cluster** | $-0.9276^{**}$ | $-0.4535^{**}$ |
| | **DSE** | $0.9392^{**}$ | $0.4693^{**}$ |
| | **ID** | $-0.6309^{**}$ | $-0.4528^{**}$ |
| **ImageNet‡** | **α-ReQ** | $-0.919^{**}$ | $-0.296^{**}$ |
| | **RankMe** | $0.127$ | $0.4953^{**}$ |
| | **RankMe$^{\star}$** | $0.7879^{**}$ | $0.09995$ |
| | **NE-Sum** | $0.6541^{**}$ | $0.4326^{**}$ |
| | **$\kappa$** | $0.05118$ | $0.03491$ |
| | **Self-Cluster** | $-0.9154^{**}$ | $-0.4649^{**}$ |
| | **DSE** | $0.9074^{**}$ | $0.4814^{**}$ |
| | **ID** | $-0.7128^{**}$ | $-0.62^{**}$ |
| **STL-10** | **α-ReQ** | $-0.4743^{**}$ | $0.2715^{**}$ |
| | **RankMe** | $0.1224$ | $0.4244^{**}$ |
| | **RankMe$^{\star}$** | $0.5362^{**}$ | $-0.3722^{**}$ |
| | **NE-Sum** | $0.6498^{**}$ | $0.4383^{**}$ |
| | **$\kappa$** | $-0.3705^{**}$ | $0.1948$ |
| | **Self-Cluster** | $-0.9311^{**}$ | $-0.4371^{**}$ |
| | **DSE** | $0.9454^{**}$ | $0.4411^{**}$ |
| | **ID** | $-0.7666^{**}$ | $-0.616^{**}$ |

‡ Correlations of ImageNet-1k should be interpreted with caution as it is used in the training pipeline of most backbones.

Table 22: Spearman correlation ($\rho$) between representation quality metrics and KNN test accuracy ($k = 1$), stratified by training objective, across fine-grained datasets. Metrics are organised by construction: spectral (top), relation-based (middle), and manifold-based (bottom). Statistical significance is considered after Benjamini–Hochberg correction. $^{*}$ $p < 0.05$, $^{**}$ $p < 0.01$, $^{***}$ $p < 0.001$. Non-significant cells are greyed out.

| | | SSL | Supervised |
|---|---|---|---|
| **Food-101** | $\boldsymbol{\alpha}$**-ReQ** | -0.1068 | -0.08897 |
| | **RankMe** | 0.1419 | $0.6169^{**}$ |
| | **RankMe$^{\star}$** | $0.3647^{*}$ | -0.1516 |
| | **NE-Sum** | $0.8026^{**}$ | $0.5775^{**}$ |
| | $\boldsymbol{\kappa}$ | $-0.4805^{**}$ | 0.03732 |
| | **Self-Cluster** | $-0.915^{**}$ | $-0.4968^{**}$ |
| | **DSE** | $0.9184^{**}$ | $0.509^{**}$ |
| | **ID** | $-0.7381^{**}$ | -0.1821 |
| **Pets** | $\boldsymbol{\alpha}$**-ReQ** | 0.003514 | 0.1265 |
| | **RankMe** | -0.0576 | $0.5116^{**}$ |
| | **RankMe$^{\star}$** | 0.2504 | $-0.2795^{**}$ |
| | **NE-Sum** | $0.6898^{**}$ | $0.4546^{**}$ |
| | $\boldsymbol{\kappa}$ | $-0.3656^{*}$ | 0.1254 |
| | **Self-Cluster** | $-0.8741^{**}$ | $-0.5216^{**}$ |
| | **DSE** | $0.8579^{**}$ | $0.5265^{**}$ |
| | **ID** | $-0.6037^{**}$ | $-0.4758^{**}$ |
| **Flowers-102** | $\boldsymbol{\alpha}$**-ReQ** | 0.1692 | 0.1027 |
| | **RankMe** | -0.1452 | $0.4476^{**}$ |
| | **RankMe$^{\star}$** | 0.1107 | $-0.3743^{**}$ |
| | **NE-Sum** | $0.9034^{**}$ | $0.5128^{**}$ |
| | $\boldsymbol{\kappa}$ | $-0.4269^{**}$ | $0.2766^{**}$ |
| | **Self-Cluster** | $-0.9326^{**}$ | $-0.4038^{**}$ |
| | **DSE** | $0.9402^{**}$ | $0.4137^{**}$ |
| | **ID** | $-0.875^{**}$ | $-0.5422^{**}$ |

Table 23: Spearman correlation ($\rho$) between representation quality metrics and KNN test accuracy ($k = 1$), stratified by training objective, across specialised datasets. Metrics are organised by construction: spectral (top), relation-based (middle), and manifold-based (bottom). Statistical significance is considered after Benjamini–Hochberg correction. $^{*}$ $p < 0.05$, $^{**}$ $p < 0.01$, $^{***}$ $p < 0.001$. Non-significant cells are greyed out.

| | | SSL | Supervised |
|---|---|---|---|
| **DTD** | **$\alpha$-ReQ** | $-0.6826^{**}$ | $-0.229^{*}$ |
| | **RankMe** | $0.1968$ | $0.6786^{**}$ |
| | **RankMe$^{\star}$** | $0.676^{**}$ | $-0.3815^{**}$ |
| | **NE-Sum** | $0.701^{**}$ | $0.4393^{**}$ |
| | **$\kappa$** | $-0.4032^{**}$ | $0.3912^{**}$ |
| | **Self-Cluster** | $-0.849^{**}$ | $-0.4414^{**}$ |
| | **DSE** | $0.867^{**}$ | $0.4669^{**}$ |
| | **ID** | $-0.6931^{**}$ | $-0.08197$ |
| **Places-365** | **$\alpha$-ReQ** | $-0.9019^{**}$ | $-0.2916^{**}$ |
| | **RankMe** | $0.1044$ | $0.683^{**}$ |
| | **RankMe$^{\star}$** | $0.8408^{**}$ | $0.1301$ |
| | **NE-Sum** | $0.7494^{**}$ | $0.4926^{**}$ |
| | **$\kappa$** | $-0.3678^{*}$ | $0.06575$ |
| | **Self-Cluster** | $-0.8866^{**}$ | $-0.522^{**}$ |
| | **DSE** | $0.9291^{**}$ | $0.5373^{**}$ |
| | **ID** | $-0.6104^{**}$ | $-0.05613$ |
| **EuroSAT** | **$\alpha$-ReQ** | $-0.5707^{**}$ | $0.0703$ |
| | **RankMe** | $0.3451$ | $0.6347^{**}$ |
| | **RankMe$^{\star}$** | $0.5169^{**}$ | $-0.3299^{**}$ |
| | **NE-Sum** | $0.728^{**}$ | $0.01291$ |
| | **$\kappa$** | $0.3036$ | $0.2812^{*}$ |
| | **Self-Cluster** | $-0.7011^{**}$ | $-0.3997^{**}$ |
| | **DSE** | $0.7085^{**}$ | $0.3821^{**}$ |
| | **ID** | $0.5169^{**}$ | $-0.3588^{**}$ |

Table 24: Spearman correlation ($\rho$) between representation quality metrics and KNN test accuracy ($k = 1$), stratified by representation dimension $d$, across natural image datasets. Metrics are organised by construction: spectral (top), relation-based (middle), and manifold-based (bottom). Statistical significance is considered after Benjamini–Hochberg correction. $^*$ $p < 0.05$, $^{**}$ $p < 0.01$, $^{***}$ $p < 0.001$. Non-significant cells are greyed out.

| | | $d \leq 520$ | $520 < d \leq 960$ | $960 < d \leq 1792$ | $d > 1792$ |
|---|---|---|---|---|---|
| **CIFAR-10** | $\boldsymbol{\alpha}$**-ReQ** | -0.2205 | $-0.3899^{**}$ | -0.05614 | 0.04828 |
| | **RankMe** | 0.165 | 0.1998 | 0.06565 | 0.3609 |
| | **RankMe**$^\star$ | 0.3151 | $0.3395^{*}$ | 0.08716 | -0.1348 |
| | **NE-Sum** | $0.4071^{**}$ | $0.5095^{**}$ | $0.4823^{**}$ | 0.2383 |
| | $\boldsymbol{\kappa}$ | -0.2296 | -0.1643 | 0.007536 | -0.02937 |
| | **Self-Cluster** | $-0.4417^{**}$ | $-0.4743^{**}$ | $-0.6552^{**}$ | -0.2042 |
| | **DSE** | $0.4505^{**}$ | $0.4812^{**}$ | $0.6481^{**}$ | 0.2263 |
| | **ID** | -0.2194 | $-0.4175^{**}$ | $-0.7011^{**}$ | $-0.6159^{**}$ |
| **CIFAR-100** | $\boldsymbol{\alpha}$**-ReQ** | -0.2681 | $-0.4752^{**}$ | -0.1803 | -0.06054 |
| | **RankMe** | 0.1627 | 0.2308 | 0.1226 | $0.3887^{*}$ |
| | **RankMe**$^\star$ | $0.3585^{*}$ | $0.4767^{**}$ | 0.217 | -0.006555 |
| | **NE-Sum** | $0.3806^{*}$ | $0.498^{**}$ | 0.2509 | 0.3759 |
| | $\boldsymbol{\kappa}$ | -0.3294 | -0.1674 | 0.06461 | -0.0821 |
| | **Self-Cluster** | $-0.4598^{**}$ | $-0.4985^{**}$ | $-0.6312^{**}$ | $-0.2487^{*}$ |
| | **DSE** | $0.4694^{**}$ | $0.5055^{**}$ | $0.6387^{**}$ | $0.2826^{*}$ |
| | **ID** | -0.1541 | $-0.5574^{**}$ | $-0.7747^{**}$ | $-0.6316^{*}$ |
| **ImageNet**$^\ddagger$ | $\boldsymbol{\alpha}$**-ReQ** | -0.1429 | $-0.5034^{**}$ | -0.1886 | -0.2237 |
| | **RankMe** | 0.1356 | $0.3769^{*}$ | -0.006137 | 0.3336 |
| | **RankMe**$^\star$ | 0.2723 | $0.4806^{**}$ | 0.03338 | -0.08957 |
| | **NE-Sum** | $0.7256^{**}$ | $0.4374^{**}$ | 0.08168 | 0.2448 |
| | $\boldsymbol{\kappa}$ | $-0.5167^{**}$ | -0.1863 | 0.108 | 0.03709 |
| | **Self-Cluster** | $-0.7721^{**}$ | $-0.4206^{*}$ | -0.1078 | $-0.2609^{*}$ |
| | **DSE** | $0.7866^{**}$ | $0.4395^{*}$ | 0.1149 | $0.2743^{*}$ |
| | **ID** | $-0.6039^{**}$ | $-0.6981^{**}$ | $-0.8526^{**}$ | $-0.6605^{**}$ |
| **STL-10** | $\boldsymbol{\alpha}$**-ReQ** | -0.1049 | -0.121 | 0.151 | $0.3781^{*}$ |
| | **RankMe** | 0.1459 | 0.2025 | -0.1058 | 0.1845 |
| | **RankMe**$^\star$ | 0.1518 | 0.2254 | -0.06956 | $-0.398^{*}$ |
| | **NE-Sum** | $0.5517^{**}$ | $0.4915^{**}$ | 0.05242 | 0.2244 |
| | $\boldsymbol{\kappa}$ | $-0.4447^{**}$ | -0.1335 | 0.2284 | $0.3505^{*}$ |
| | **Self-Cluster** | $-0.6104^{**}$ | $-0.4807^{**}$ | $-0.4443^{**}$ | $-0.3438^{**}$ |
| | **DSE** | $0.6193^{**}$ | $0.4924^{**}$ | $0.4354^{**}$ | $0.3341^{**}$ |
| | **ID** | $-0.4592^{**}$ | $-0.6152^{**}$ | $-0.7638^{**}$ | $-0.7391^{**}$ |

$^\ddagger$ Correlations of ImageNet-1k should be interpreted with caution as it is used in the training pipeline of most backbones.

Table 25: Spearman correlation ($\rho$) between representation quality metrics and KNN test accuracy ($k = 1$), stratified by representation dimension $d$, across fine-grained datasets. Metrics are organised by construction: spectral (top), relation-based (middle), and manifold-based (bottom). Statistical significance is considered after Benjamini–Hochberg correction. $^{*}$ $p < 0.05$, $^{**}$ $p < 0.01$, $^{***}$ $p < 0.001$. Non-significant cells are greyed out.

| | | $d \le 520$ | $520 < d \le 960$ | $960 < d \le 1792$ | $d > 1792$ |
|---|---|---|---|---|---|
| **Food-101** | $\boldsymbol{\alpha}$**-ReQ** | -0.07838 | -0.2344 | 0.02227 | 0.1511 |
| | **RankMe** | 0.158 | 0.2704 | 0.01413 | $0.4386^{*}$ |
| | **RankMe$^{\star}$** | 0.238 | 0.267 | 0.04507 | -0.1973 |
| | **NE-Sum** | $0.5371^{**}$ | $0.5656^{**}$ | $0.3925^{**}$ | $0.3831^{*}$ |
| | $\boldsymbol{\kappa}$ | $-0.6333^{**}$ | -0.2241 | -0.002442 | -0.1096 |
| | **Self-Cluster** | $-0.4853^{*}$ | $-0.6011^{**}$ | $-0.6187^{**}$ | -0.2234 |
| | **DSE** | $0.4863^{*}$ | $0.6044^{**}$ | $0.6109^{**}$ | 0.2261 |
| | **ID** | 0.09847 | $-0.5978^{**}$ | $-0.5298^{**}$ | -0.2837 |
| **Pets** | $\boldsymbol{\alpha}$**-ReQ** | 0.1372 | 0.1471 | 0.09649 | 0.179 |
| | **RankMe** | 0.1272 | 0.14 | 0.04046 | 0.1671 |
| | **RankMe$^{\star}$** | 0.03904 | 0.02955 | 0.02048 | -0.09902 |
| | **NE-Sum** | $0.5584^{**}$ | $0.4913^{*}$ | $0.3139^{*}$ | -0.06825 |
| | $\boldsymbol{\kappa}$ | $-0.5936^{**}$ | -0.2191 | -0.04975 | 0.03399 |
| | **Self-Cluster** | $-0.7578^{**}$ | $-0.5976^{**}$ | $-0.4304^{**}$ | -0.2296 |
| | **DSE** | $0.7589^{**}$ | $0.6042^{**}$ | $0.4148^{**}$ | 0.2095 |
| | **ID** | $-0.5793^{**}$ | $-0.541^{**}$ | $-0.565^{**}$ | $-0.4538^{*}$ |
| **Flowers-102** | $\boldsymbol{\alpha}$**-ReQ** | -0.003907 | 0.1629 | 0.2256 | 0.1044 |
| | **RankMe** | 0.1992 | 0.1018 | -0.09509 | $0.3341^{*}$ |
| | **RankMe$^{\star}$** | 0.1636 | -0.003773 | 0.1699 | -0.2636 |
| | **NE-Sum** | $0.4765^{**}$ | $0.626^{**}$ | $0.689^{**}$ | $0.5271^{**}$ |
| | $\boldsymbol{\kappa}$ | $-0.5285^{**}$ | -0.1731 | $-0.2886^{*}$ | $-0.4494^{**}$ |
| | **Self-Cluster** | -0.4042 | $-0.6233^{**}$ | $-0.6737^{**}$ | -0.2488 |
| | **DSE** | 0.4067 | $0.6254^{**}$ | $0.671^{**}$ | 0.2715 |
| | **ID** | -0.3548 | $-0.7832^{**}$ | $-0.8095^{**}$ | $-0.4983^{**}$ |

Table 26: Spearman correlation ($\rho$) between representation quality metrics and KNN test accuracy ($k = 1$), stratified by representation dimension $d$, across specialised datasets. Metrics are organised by construction: spectral (top), relation-based (middle), and manifold-based (bottom). Statistical significance is considered after Benjamini–Hochberg correction. $^*$ $p < 0.05$, $^{**}$ $p < 0.01$, $^{***}$ $p < 0.001$. Non-significant cells are greyed out.

| | | $d \le 520$ | $520 < d \le 960$ | $960 < d \le 1792$ | $d > 1792$ |
|---|---|---|---|---|---|
| **DTD** | **α-ReQ** | $-0.3514^*$ | $-0.4515^{**}$ | -0.106 | -0.1975 |
| | **RankMe** | 0.3403 | $0.2915^*$ | 0.0454 | $0.3509^*$ |
| | **RankMe$^\star$** | $0.4068^{**}$ | $0.4418^{**}$ | 0.1551 | -0.3448 |
| | **NE-Sum** | $0.3725^*$ | $0.3289^*$ | 0.005449 | 0.1462 |
| | **κ** | $-0.4265^*$ | -0.1425 | 0.01042 | 0.195 |
| | **Self-Cluster** | $-0.4378^*$ | $-0.4512^{**}$ | $-0.5059^{**}$ | -0.05875 |
| | **DSE** | $0.4304^*$ | $0.4937^{**}$ | $0.5081^{**}$ | 0.08992 |
| | **ID** | $-0.4162^{**}$ | -0.3066 | $-0.5788^{**}$ | -0.05981 |
| **Places-365** | **α-ReQ** | $-0.4912^{**}$ | $-0.6097^{**}$ | $-0.4195^{**}$ | -0.07941 |
| | **RankMe** | $0.3319^*$ | $0.3252^*$ | 0.1573 | $0.433^*$ |
| | **RankMe$^\star$** | $0.5883^{**}$ | $0.6243^{**}$ | $0.3877^{**}$ | -0.008747 |
| | **NE-Sum** | $0.4782^*$ | $0.6324^{**}$ | 0.1873 | 0.2742 |
| | **κ** | $-0.5216^{**}$ | -0.2482 | 0.07337 | 0.1908 |
| | **Self-Cluster** | $-0.5519^{**}$ | $-0.6319^{**}$ | $-0.6644^{**}$ | -0.2103 |
| | **DSE** | $0.5437^{**}$ | $0.6405^{**}$ | $0.6658^{**}$ | 0.2362 |
| | **ID** | 0.1069 | $-0.3021^*$ | $-0.4769^{**}$ | -0.3434 |
| **EuroSAT** | **α-ReQ** | -0.1636 | -0.2966 | 0.07227 | 0.1166 |
| | **RankMe** | 0.1483 | 0.187 | 0.02651 | -0.1176 |
| | **RankMe$^\star$** | 0.294 | 0.2558 | -0.1174 | -0.1044 |
| | **NE-Sum** | 0.1268 | 0.12 | $-0.3936^*$ | -0.1562 |
| | **κ** | -0.3574 | -0.03989 | 0.2584 | 0.02616 |
| | **Self-Cluster** | $-0.4899^*$ | $-0.5335^{**}$ | -0.08774 | 0.1669 |
| | **DSE** | 0.4634 | $0.5044^{**}$ | 0.07196 | -0.1542 |
| | **ID** | $-0.3833^*$ | $-0.3834^*$ | -0.1424 | -0.1645 |

Table 27: Spearman correlation ($\rho$) between representation metrics and test accuracy on Linear Probe trained using SGD, across CIFAR-10, CIFAR-100, and ImageNet-1k. We analyze three regimes: $\alpha < 1$, $\alpha > 1$, and unconstrained (all $\alpha$). Bold underlined values indicate statistical significance ($p < 0.05$).()

| | CIFAR-10 | | CIFAR-100 | | ImageNet-1k | |
|---|---|---|---|---|---|---|
| Metric | $\rho$ | $p$ | $\rho$ | $p$ | $\rho$ | $p$ |
| $\alpha >= 1.0$ | $\underline{0.1357}$ | $\underline{3.2360 \times 10^{-2}}$ | 0.0817 | 0.2298 | 0.0675 | 0.5651 |
| $\alpha < 1.0$ | $-0.4794$ | 0.0602 | $\underline{-0.5287}$ | $\underline{1.5860 \times 10^{-4}}$ | $\underline{-0.2734}$ | $\underline{1.6470 \times 10^{-3}}$ |
| $\alpha$-Req | 0.0259 | 0.6752 | $-0.0146$ | 0.8129 | $-0.0970$ | 0.1666 |
| RankMe | 0.1138 | 0.0644 | $\underline{0.2472}$ | $\underline{4.8930 \times 10^{-5}}$ | $\underline{0.2009}$ | $\underline{3.8680 \times 10^{-3}}$ |
| RankMe$^\star$ | $\underline{-0.1565}$ | $\underline{1.0720 \times 10^{-2}}$ | $-0.1378$ | $2.5140 \times 10^{-2}$ | $-0.0225$ | 0.7493 |
| NE-Sum | $\underline{0.2169}$ | $\underline{3.7580 \times 10^{-4}}$ | $\underline{0.2162}$ | $\underline{4.0230 \times 10^{-4}}$ | 0.0588 | 0.4026 |
| $\kappa$ | $\underline{0.2663}$ | $\underline{1.1140 \times 10^{-5}}$ | $\underline{0.3458}$ | $\underline{7.9080 \times 10^{-9}}$ | 0.0903 | 0.1980 |
| Self-Cluster | $-0.1787$ | $3.5210 \times 10^{-3}$ | $-0.1797$ | $3.4000 \times 10^{-3}$ | $-0.0585$ | 0.4048 |
| DSE | $\underline{0.1856}$ | $\underline{2.4150 \times 10^{-3}}$ | $\underline{0.1960}$ | $\underline{1.3720 \times 10^{-3}}$ | 0.0740 | 0.2916 |
| ID | $\underline{-0.5608}$ | $\underline{2.3610 \times 10^{-23}}$ | $-0.6085$ | $3.9740 \times 10^{-28}$ | $-0.6189$ | $4.6090 \times 10^{-23}$ |

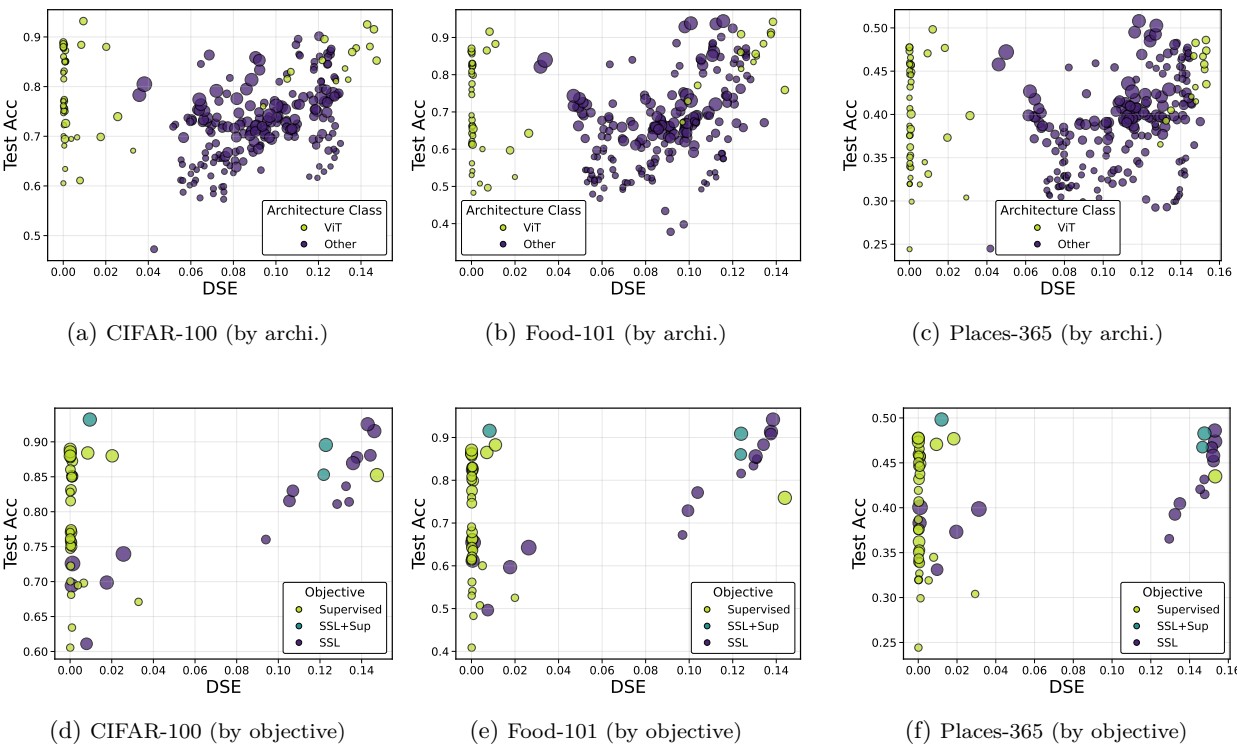

Figure 12: DSE metric values plotted against test accuracy across datasets. The top row shows all models coloured by architecture class, revealing that the models with extreme values predominantly belong to the ViT architecture class. The bottom row shows only ViT models coloured by training objective, illustrating that the near zero DSE values originate predominantly from Supervised and SSL+Sup models rather than SSL models. This suggests that the DSE metric may be a reliable predictor for SSL models (though the sample size is small), but is unreliable for Supervised and SSL+Sup models.

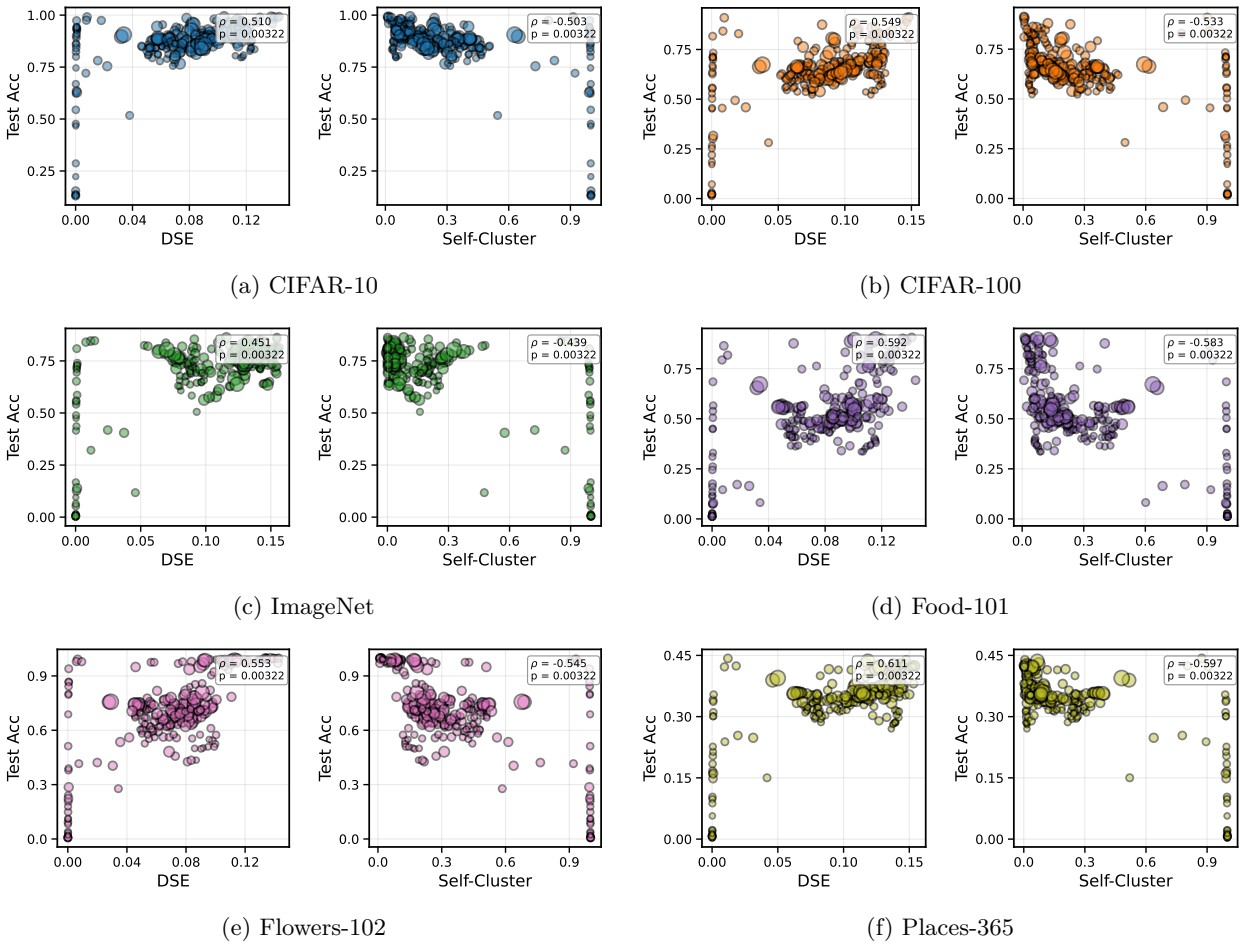

Figure 13: Comparison of DSE and Self-Cluster against KNN linear-probe test accuracy across six datasets. The $\rho$ denotes the Spearman correlation and the corresponding $p$-value is annotated in each subplot. Marker size is proportional to each model's representation dimensionality, scaled linearly relative to the highest-dimensional model.

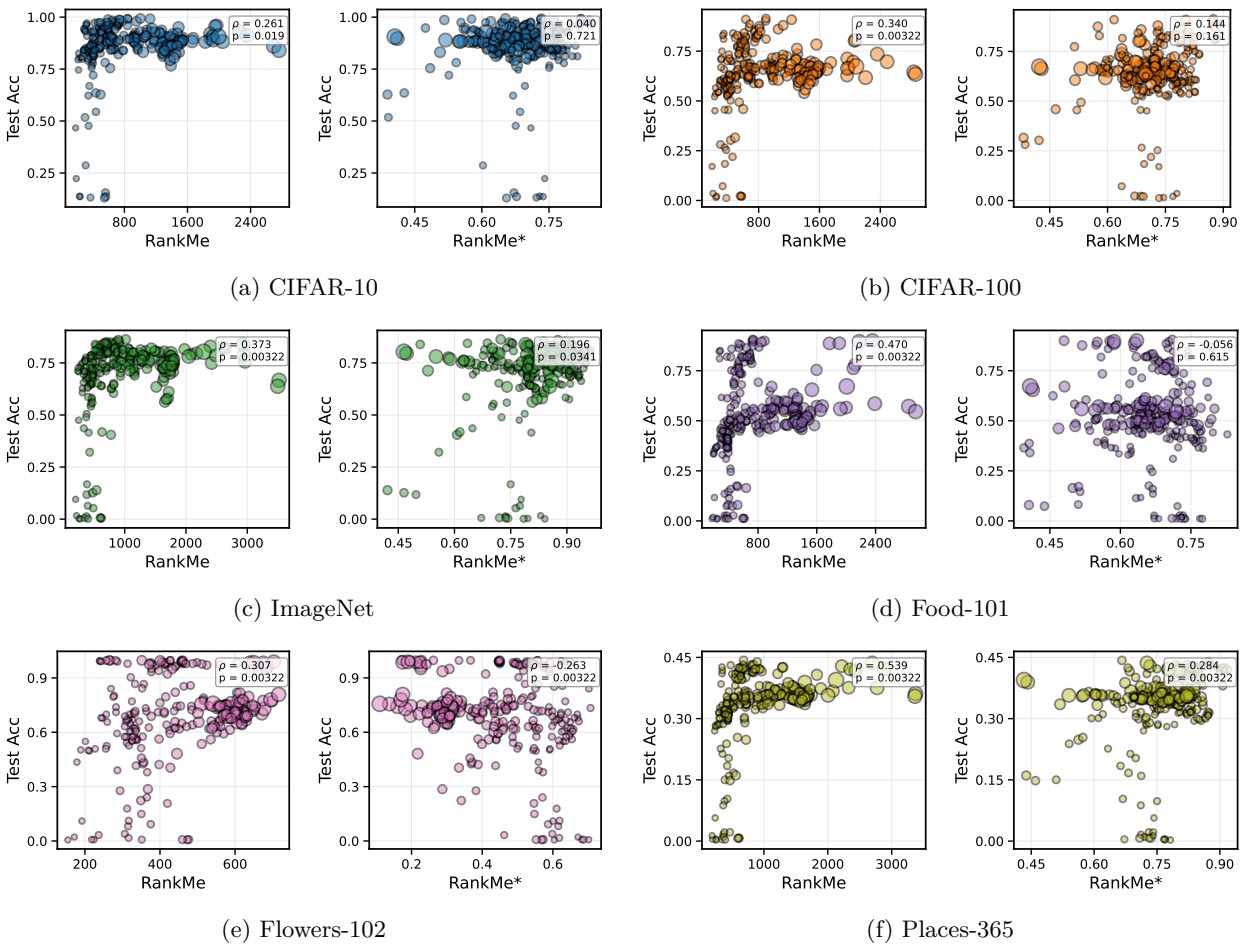

Figure 14: Comparison of RankMe and RankMe* against KNN probe test accuracy across six datasets. The $\rho$ denotes the Spearman correlation and the corresponding $p$-value is annotated in each subplot. Marker size is proportional to each model's representation dimensionality, scaled linearly relative to the highest-dimensional model.

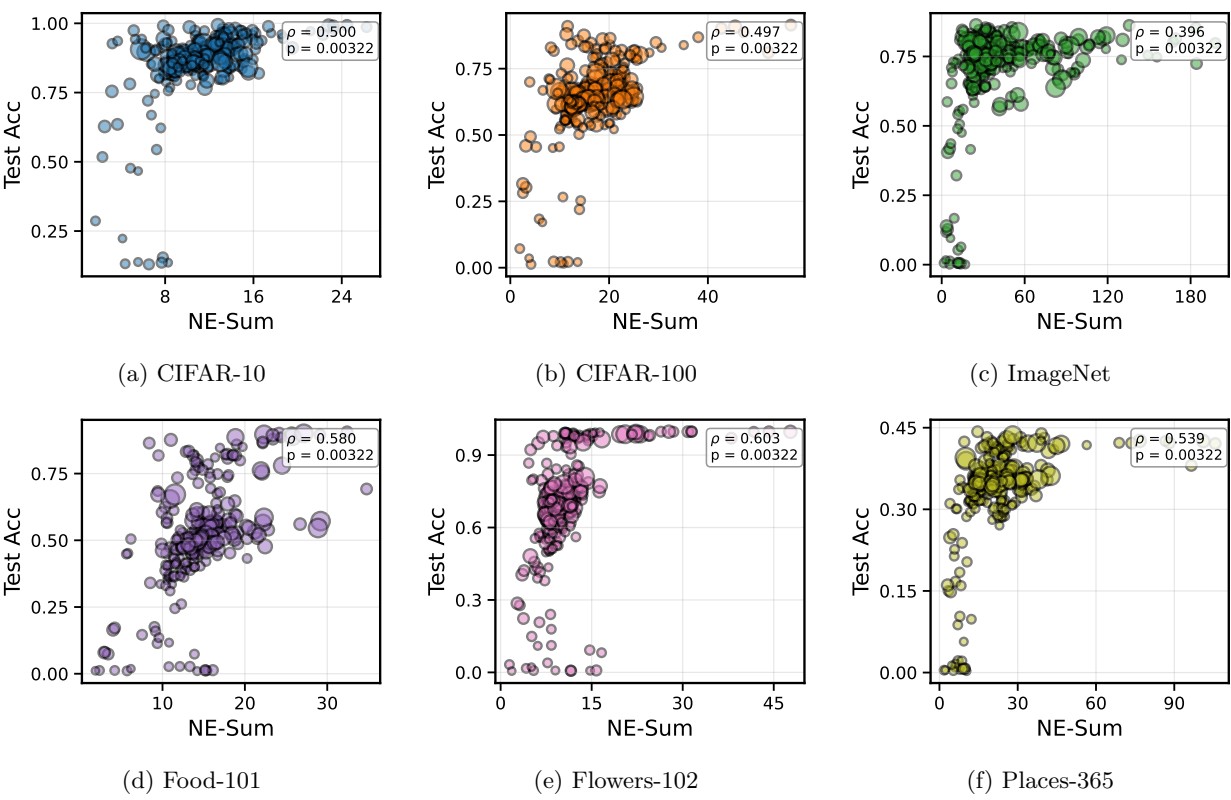

Figure 15: Comparison of NE Sum against KNN linear-probe test accuracy across six datasets. The $\rho$ denotes the Spearman correlation and the corresponding $p$-value is annotated in each subplot. Marker size is proportional to each model's representation dimensionality, scaled linearly relative to the highest-dimensional model.

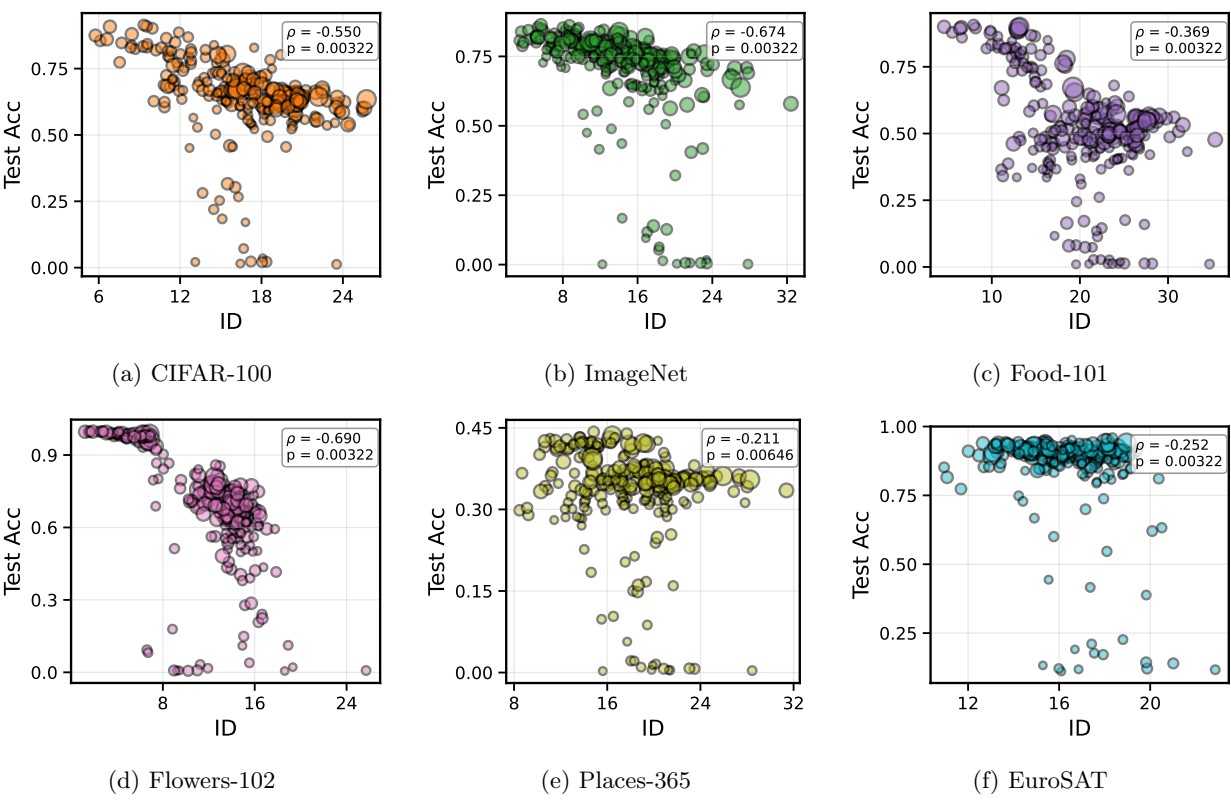

(a) CIFAR-100      (b) ImageNet      (c) Food-101

(d) Flowers-102      (e) Places-365      (f) EuroSAT

Figure 16: Comparison of ID against KNN probe test accuracy across six datasets. The $\rho$ denotes the Spearman correlation and the corresponding $p$-value is annotated in each subplot. Marker size is proportional to each model's representation dimensionality, scaled linearly relative to the highest-dimensional model.

