# OpenReview forum: "A Comparative Study of Label-free Representation Quality Metrics in Deep Learning"
_TMLR — Under review for TMLR_

### Review · Reviewer_eE6b · 2026-06-04

**Summary Of Contributions:**

### Summary of Contributions

This paper presents a systematic comparative evaluation of label-free metrics for assessing representation quality in deep neural networks. The authors make three main contributions: (1) they propose a taxonomy that groups existing metrics into three families while analytically establishing connections between metrics within families; (2) they conduct controlled synthetic experiments characterizing how spectral metrics respond to changes in eigenspectral shape and representation dimensionality; and (3) they perform a large-scale empirical evaluation across 260 vision models on CIFAR-10, CIFAR-100, and ImageNet-1K, stratifying results by architecture class, training objective, and representation dimension. The central finding is that intrinsic dimensionality (ID) is the most consistently reliable predictor of downstream performance across all settings, while other metrics' reliability is strongly moderated by architecture and training objective. This is a useful empirical study that addresses a timely and practical question. The taxonomy and stratified evaluation design are strong contributions. The finding that ID is the most reliable metric is interesting but would be more compelling with mechanistic understanding and validation under standard evaluation protocols. The paper would benefit from addressing the confounding role of model capacity and from a more thorough treatment of the statistical analysis. With the suggested revisions, this could become a valuable reference for the community.

### Key Strengths

1. **Breadth and rigor of evaluation**: The study is commendably comprehensive, covering 260 models across multiple architecture families (ViT, ResNet, ConvNeXT, RegNet), training paradigms (supervised, SSL, SSL+Sup), and datasets of varying complexity. The stratification by architecture, objective, and representation dimension is a significant methodological strength that reveals important nuances hidden in aggregate results.

2. **Useful taxonomy and analytical connections**: The grouping of metrics into three families with a unified notation is pedagogically valuable. The analytical connection between DSE and Self-Cluster (showing they are nearly the same metric in practice with ρ ≈ −0.999), and the equivalence between NE Sum and Stable Rank under centering, are useful contributions that simplify the landscape for practitioners.

3. **Synthetic sensitivity analysis**: The controlled experiments decoupling spectral shape (α) from representation dimensionality (d) provide genuine insight into the behavior of spectral metrics. The observation that no single spectral metric provides uniformly consistent sensitivity across all spectral shapes is a valuable finding.

4. **Practical relevance**: The paper addresses a genuinely important problem—how to select pretrained models without expensive evaluation—and the finding that metrics should be computed on backbone outputs rather than projector embeddings (which are typically discarded) is practically important.


### Key Weaknesses

1. **Choice of downstream evaluation protocol**: The paper uses an OLS (Ordinary Least Squares) regression fit with one-hot targets as the downstream evaluation. This is a significant methodological concern:
   - Converting classification to regression is a strong assumption that may not reflect the actual utility of representations.
   - The authors acknowledge this limitation but do not validate whether their conclusions hold under more standard protocols (linear probing with SGD, KNN). While Appendix E provides SGD results on ~120 models, the main narrative is built entirely on OLS. It would be more convincing if the paper demonstrated consistency across evaluation protocols more prominently, or justified why OLS is a reliable proxy.
   - Under OLS with ℓ₂-normalized features, the solution is essentially a pseudoinverse, which means the condition number κ directly affects the solution quality. This creates a confound: κ may correlate with OLS accuracy not because it measures representation quality, but because it affects the OLS solver itself.

2. **Limited scope of tasks and modalities**: All experiments are restricted to image classification on three datasets. The paper claims to study "representation quality" broadly, but:
   - No evaluation on detection, segmentation, or other downstream tasks where representation quality matters differently.
   - No text or multimodal models are considered, limiting generalizability claims.
   - The datasets span limited diversity (CIFAR-10/100 are relatively simple; ImageNet is a single complex benchmark).

3. **Causal interpretation is lacking**: The paper establishes correlations between metrics and test accuracy but does not establish causality or provide mechanistic understanding of *why* ID is the best predictor. Is low ID truly a cause of good generalization, or is it merely a confound correlated with model size, training data, or other factors? The paper would benefit from controlling for model capacity and training data volume.

4. **Imbalanced model groups**: The SSL group contains only 32 models and SSL+Sup only 19 models, versus 209 supervised models. This imbalance means that:
   - Aggregate correlations are dominated by supervised models.
   - Conclusions about SSL metrics are drawn from relatively small samples, reducing statistical power and potentially inflating correlation estimates.

5. **Missing analysis of metric combinations**: Given that different metrics capture complementary aspects of representation geometry, an obvious question is whether combining metrics (e.g., via multivariate regression) would yield better predictions. The paper mentions CLID as future work but does not attempt even a simple combination analysis.

6. **Incomplete treatment of hyperparameter sensitivity**:
   - For α-ReQ, the fitting range is described as a "consistent heuristic" but no sensitivity analysis is provided for this choice.
   - For DSE, σ=10 is used as default, but the paper acknowledges sensitivity to σ without exploring it empirically.
   - For ID, batch size effects are mentioned but not systematically studied.

**Audience:**

Yes

**Audience Explanation:**

This is a useful empirical study that addresses a timely and practical question. The taxonomy and stratified evaluation design are strong contributions. Should be of interest by TMLR audiences.

**Broader Impact Concerns:**

No broader impact concerns.

**Claims And Evidence:**

Yes

**Claims Explanation:**

This paper compares existing families of label-free representation quality metrics through the lens of sensitivity and downstream task performance. The paper concludes that ID is the most reliable predictor across all settings. This claim is well supported by aggregated experimental results in Table 2, with the experimental settings well justified by other tables across differnet model architectures and training settings.

**Requested Changes:**

### Requested Changes

**Major:**

1. **Validate conclusions under standard linear probing**: Either promote the SGD results from Appendix E to the main paper or demonstrate statistically that OLS rankings are consistent with linear probe rankings across the full 260-model set.

2. **Address the κ-OLS confound**: Provide analysis or discussion of whether κ's correlation with OLS accuracy is artifactual (due to the solver) rather than reflecting genuine representation quality.

3. **Control for model capacity**: Include model parameter count or FLOP count as a covariate in the correlation analysis to assess whether ID (or other metrics) predict performance beyond what model size already explains.

4. **Provide confidence intervals**: Especially for the stratified analyses where group sizes are small.

**Minor:**

5. Include a brief analysis of metric combinations (e.g., can ID + RankMe together outperform ID alone?).

6. Expand the discussion of why ID works—connect to existing theoretical work on generalization bounds and intrinsic dimensionality.

7. Add a practical recommendation section: given the findings, what should a practitioner actually do when selecting a model from HuggingFace?

8. Discuss computational cost comparison more explicitly—how much cheaper are these metrics compared to linear probing or KNN evaluation?

---

> ### Comment · Action_Editor_6GUx · 2026-06-04
> **Please update your review**
>
> Dear Reviewer eE6b,
>
> Thank you for submitting the review in time.
>
> For the following questions and your answers
> > Are the claims made in the submission supported by accurate, convincing and clear evidence?: Yes
> > Explain your answer above:
> > Experiments are detailed and well documented.
>
> , we need more details as the current answers do not discuss one of the key acceptance criteria for TMLR:
> > Are the claims made in the submission supported by accurate, convincing and clear evidence?
>
>
> Could you please clarify
> 1. what claims made in the submission are (or NOT) supported by accurate, convincing and clear evidence and
> 2. what the evidence is
> ?
>
> Thank you,
> Your AE

---

> ### Author Response · Authors · 2026-07-01
> **Response to Reviewer eE6b**
>
> We thank the reviewer for their thorough and insightful comments, which have led to meaningful improvements in the paper.
>
> ### Validate conclusions under standard linear probing; SGD results in Appendix E should be promoted or OLS/SGD consistency demonstrated across full 260-model set
>
> Addressed. To verify that our conclusions do not depend on the OLS solver, we introduce a KNN probe ($k=1$) as Table 2b in the main paper. Unlike OLS, the KNN probe requires no linear algebra routines susceptible to numerical conditioning, making it a clean test of whether OLS-based findings reflect genuine representation quality rather than solver artefacts. Across all six datasets, ID retains the strongest and most consistent negative correlation under KNN ($\rho \in [-0.67, -0.55]$), confirming that the
> primary claim is not an OLS-specific result. Both OLS and KNN results are presented in Table 2.
>
> ### $\kappa$-OLS confound: $\kappa$ may correlate with OLS accuracy artifactually via the solver's numerical sensitivity
>
> Agreed entirely. This concern is now explicitly addressed in three places: (1) a Note in Section 3.1 identifies $\kappa$ as solver-dependent; (2) Table 2b shows that $\kappa$'s OLS correlations largely vanish under the KNN probe, providing direct empirical confirmation; and (3) Section 4.2 explicitly states that "$\kappa$'s correlation with downstream accuracy is a result of the OLS solver's numerical conditioning rather than a measure of representation quality" and drops $\kappa$ from all subsequent fine-grained analyses.
>
> ### Control for model capacity (parameter count or FLOPs) to assess whether ID predicts beyond model size
>
> We thank the reviewer for highlighting this important potential confound. To directly address it, we performed a partial correlation analysis controlling for model parameter count (Appendix B.4). The results show that ID remains significantly correlated with downstream performance on five of the six datasets after controlling for model capacity (CIFAR-100: $\rho = -0.43$, $p < 0.001$; ImageNet: $\rho = -0.67$, $p < 0.001$; Food-101: $\rho = -0.21$, $p = 0.001$; Flowers-102: $\rho = -0.66$, $p < 0.001$; EuroSAT: $\rho = -0.17$, $p = 0.005$). Only Places-365 becomes non-significant ($\rho = -0.02$, $p = 0.759$). While model capacity explains part of the observed relationship, these results demonstrate that the predictive power of ID is still present even after accounting for parameter count. We added our finding in the discussion section.
>
> ### Provide confidence intervals, especially for stratified analyses with small group sizes
>
> Addressed via the cluster-robust bootstrap that was done based on the changes suggested by Reviewer kVhE. The bootstrap naturally yields standard errors, and all significance stars in revised Tables 2–4 reflect these corrected, cluster-aware $p$-values. We also explicitly acknowledge in Section 4.2 that the SSL group ($n=32$) is comparatively small and that conclusions about SSL-specific metrics carry lower statistical power than those for the supervised group ($n=209$).
>
> ###  Metric combination analysis (can ID + RankMe outperform ID alone?)
>
> We thank the reviewer for this valuable suggestion. We agree that combining metrics is a promising direction; however, a meaningful analysis would require a more in-depth study of the interactions between metric families. Since our goal is to provide a systematic comparison of individual metrics, we leave this investigation to future work.
>
> ### Expand discussion of why ID works; connect to generalisation bounds and intrinsic dimensionality theory
>
> Addressed. We now state the motivating hypotheses for each metric in Sections 3.1–3.3 and use these hypotheses throughout the discussion section to interpret the empirical results. This provides a principled explanation of why certain metrics succeed or fail in different settings and what the findings imply about the properties captured by each metric.

---

### Review · Reviewer_8w65 · 2026-06-14

**Summary Of Contributions:**

This paper evaluates several model quality metrics for how well they correspond with linear classification probe.  Extensive evaluation data indicate that Intrinsic Dimension (ID) correlates fairly decently with label accuracy metrics, while other methods have little or inconsistent correlation; DSE and Self-Cluster also correlate for SSL methods.

**Additional Comments:**

- sec 3.1, 3.2, 3.3:  These are good summaries, but mostly just on the computation of each metric, with a just a touch on motivational observation.  But they don't supply an intuition of what they capture or why they might be related to downstream performance.  For example, "α-ReQ: Building on the idea that the eigenspectrum of neural population responses in mammals follows a power law, λi ∝ i−α, ...".  This states a motivational observation, but doesn't say anything about why it might be a good property for downstream performance.  Similarly with most of the other methods and sections.  While the mechanical descriptions are nice and clear, I think it would be better to also say something about why these properties might matter here as well.  Without this, the various methods start to blend together as just a bunch of numbers without enough grounding in their goals.  Briefly relating each measure to the goal of downstream performance --- at least an intuition of why it might be the case, whether or not it is obtained --- would help a lot in distinguishing them, as well as creating hypotheses for what to probe in measurements.


- ID sec 3.3 description is not detailed enough, especially compared to the other methods, that have clear formula for what they compute.  while the others lack explanation relating to perf goals, ID is the opposite:  it has a good sentence saying "Lower ID suggests that the learnt representation lies on a lower-dimensional manifold, which the authors show is associated with better generalisation performance.", but no details on how to actually compute it.


- for each model studied (both ssl and supervised), what dataset(s) was it trained on?  how many on imagenet (which was an eval dataset)?  some were at least fine-tuned on imagenet, it looks like.  for example, sec 4.2: compared on imagenet, but it looks like many of the models were finetuned on imagenet (even some of the ssl ones, according to the legend in fig 2b)

- are there any other datasets that are entirely held-out from the set of data studied, e.g.: SUN397, openimages (maybe bbox crops of openimages?), places365


- groupings by model arch and dimension d are some of the more interesting breakdowns.  but they are very verbose and come late in the paper.

- what do the sizes of the circles in the plots in fig 4 mean?

- table 2:  if this is showing correlations, I'm not sure why there are higher/lower-is-better arrows (or closer to 1 arrow), which are a little confusing.  this doesn't show the actual values of the metrics, but the correlations with test acc, from my reading here.  if that's the case, then larger abs value is always better correlation

**Audience:**

No

**Audience Explanation:**

While extensive raw evaluation data is presented, this paper is very light in interpretation and explanation for *why* these are the case.  With a few exceptions (correspondence of DSE and Self-Cluster is a good one, for example), there isn't enough analysis or insights into what was found, particularly for a paper of this length.  If the overall argument is that the results are negative and there are not many correlations, this could be made more cleanly with more summarization statistics and succinct plots and arguments; however, even then I think the paper could also say more on *why* it failed to find such correlations --- what is different about this setting from the original papers that did find use in their methods?

Similarly, the summaries of the methods, while clear and relatively concise, are restating the method mechanics of how to compute the values, with little in intuition or explanation of why this is a good value to compute, and in what ways it might correlate with label classification.  Why is full rank good, for example?  And then if found not to be, what conclusions can be drawn about either the method or data that the metric failed to capture?

**Claims And Evidence:**

Yes

**Claims Explanation:**

There is a lot of data presented, with claims drawn directly from these.  However, they also rely mostly on CIFAR and ImageNet.  CIFAR is much lower res than models' inputs, and ImageNet was sometimes a training dataset.  Better would be different datasets at full model res.  Also there is a limitation of least-squares solver vs classifier training, though this is acknowledged in the limitations.

**Requested Changes:**

I think a major revision, possibly resubmit, is needed here.  The paper needs to examine more of the reasons behind what is found in the data, and how these relate to the methods and/or experimental protocols, in order to gain more insight into methods' strengths or limitations.  Even better would be testable hypotheses on what each method captures (or doesn't), as would be reasons or interpretations that can point towards potential ways to improve.

---

> ### Author Response · Authors · 2026-07-01
> **Response to Reviewer 8w65**
>
> We would like to thank the reviewer for their careful reading of our manuscript and their helpful comments.
>
> ### Insufficient interpretation; paper does not explain why metrics succeed or fail; methods "blend together"
>
> Although the original manuscript focused primarily on providing a unified and rigorous description of the metrics, we agree that additional discussion of the intuition and hypotheses underlying each metric strengthens the paper and makes the empirical findings easier to interpret. Sections 3.1–3.3 are substantially rewritten. Each metric description now contains (a) a formal statement of what geometric property is captured, (b) a principled argument for why that property should correlate with downstream linear classification accuracy. For example, the RankMe section now argues via Cover's theorem (Cover, 1965) that higher effective rank gives a linear classifier greater separating capacity, but notes that "effective rank counts every high-variance direction, including those that carry no class-discriminative signal," and therefore "reliably flags rank collapse but does not sufficiently capture representation quality." For Self-Cluster, the revision draws on Wang & Isola (2020) showing that uniform hypersphere distributions maximise entropy and strongly predict downstream accuracy, grounding the motivation in established theory. For ID, the revision explains the TwoNN estimator and connects lower ID to the manifold hypothesis: fewer intrinsic dimensions reduce the effective complexity of the linear problem. We believe these additions directly address the request for "testable hypotheses on what each method captures."
>
> ### CIFAR is much lower resolution than model inputs; ImageNet is often a training dataset; better datasets needed
>
> Addressed. The evaluation is expanded from three to six datasets. The four added datasets: Food-101, Flowers-102, Places-365, and EuroSAT, are all used at full model input resolution and span fine-grained object classification, scene recognition, and remote sensing, covering task types not represented in the original three. A footnote in Table 2 explicitly warns that "ImageNet-1k results should be interpreted with caution as it appears in the training pipeline of most backbones." The expanded scope also allows more nuanced conclusions: ID is strongest on generic object classification ($\rho \in [-0.63, -0.60]$) and fine-grained object classification ($\rho \in [-0.76, -0.37]$), and weaker on scene recognition (Places-365, $\rho = -0.18$) and remote sensing (EuroSAT, $\rho = -0.21$).
>
>
> ### Sections 3.1–3.3 lack intuition connecting metric to downstream performance goal
>
> Addressed as discussed above
>
> ### ID description not detailed enough; no formula for how it is computed
>
> Addressed. Section 3.3 now includes the full TwoNN estimator derivation
>
> ### Training datasets for models; some models fine-tuned on ImageNet used as eval dataset
>
> The ImageNet caveat footnote is now in Table 2. Additionally, Appendix C (model list) records the training dataset for each backbone. We confirm that the additional four datasets (Food-101, Flowers-102, Places-365, EuroSAT) are not part of the training pipeline for any model in our zoo, providing clean held-out evaluation.
>
> ### Other entirely held-out datasets
>
> We added Food-101, Flowers- 102, Places-365 and EuroSAT to the main evaluation.
>
> ### What do circle sizes in scatter plots mean?
>
> All figure captions now read: "Marker size is proportional to each model's representation dimensionality, scaled linearly relative to the highest-dimensional model."
>
> ### Higher/lower-is-better arrows in correlation table are confusing since the table shows correlations, not metric values
>
> We agree that the arrows were potentially confusing in the context of correlation coefficients and have therefore removed them. In addition, we removed metrics that were subsequently shown to be unstable ($\alpha$-ReQ) or solver-dependent ($\kappa$), simplifying the presentation and making the key conclusions easier to interpret.

---

### Review · Reviewer_kVhE · 2026-06-24

**Summary Of Contributions:**

This paper is a systematic and comparative study of label-free representation quality metrics, i.e., metrics that estimate how good a representation is for downstream tasks using only the geometry of the features without labels. The main contributions are:
- A taxonomy that organises seven existing metrics into three families: (i) spectral metrics ($\alpha$-ReQ, RankMe, NE Sum, condition number $\kappa$), (ii) relational metrics (Self-Cluster, DSE), and (iii) manifold metrics (ID = Intrinsic Dimension).
- The paper reformulates metrics into a common form and establishes two clean equivalences/relationships: NE Sum equals Stable Rank under centering, and DSE and Self-Cluster are both functions of the normalised Gram matrix $\tilde X \tilde X^\hat$ under $\ell 2$-normalisation and large kernel bandwidth, which is then confirmed empirically by a near-perfect $\rho \approx -0.999$ correlation between them.
- The authors present a synthetic sensitivity analysis to decouple spectral shape (power-law exponent $\alpha$) from representation dimension ($d$) and characterise how RankMe, NE Sum and $\kappa$ respond, showing the three spectral metrics have complementary, and non-overlapping sensitivity regimes.
- They presenta a large empirical study of 260 vision backbones on CIFAR-10, CIFAR-100 and ImageNet-1k, reporting Spearman correlations between each metric and test accuracy, stratified by architecture class, training objective, and representation dimension. The headline finding is that ID is the most consistent predictor of accuracy, while the reliability of the other metrics is strongly moderated by architecture and training objective. Also, the popular $\alpha \approx 1$ hypothesis of Agrawal et al. (2022) does not generalise.

Strengths:
- the breadth (260 models × 3 datasets × multiple stratifications) is substantial
- the analytical reformulations (NE Sum = Stable Rank; DSE ≈ Self-Cluster) are clarifying in that showing two "different" metrics are essentially one is a useful service to the field
- the paper is honest about several of its own limitations (RankMe computed off-label on the backbone, the OLS-accuracy simplification, the too-small SSL+Sup group, the known bias of the ID estimator) and Appendix E re-runs the core analysis with a properly SGD-trained linear probe and reaches the same ID conclusion

Weaknesses:
- The statistical inference underlying the many significance-based claims is fragile: significance is computed on a highly non-independent model pool (dozens of near-duplicate variants from the same family) with no correction for multiple comparisons across hundreds of tests.
- The "ground-truth" test accuracy is itself a closed-form least-squares proxy whose regularisation is unspecified.
- The abstract overstates the headline ("most reliable predictor in all settings") relative to the body's own data, and there is an arrow-direction inconsistency for kappa between the tables and the figures.
These are largely fixable in revision and mostly affect the finer-grained claims rather than the central conclusions.

**Additional Comments:**

LeCun et al., Gradient-based learning applied to document recognition is dated 2002 but it should be 1998.

**Audience:**

Yes

**Audience Explanation:**

Label-free and training-free representation quality metrics are an active topic for the self-supervised-learning, transfer-learning, and model-selection communities (RankMe, α-ReQ, NE Sum, DSE, etc. are all recent and in use). A careful, apples-to-apples comparison across a large model zoo, together with the practical takeaway that ID is the most dependable single metric while spectral and relational metrics are strongly conditioned on architecture and training objective, is directly useful to practitioners choosing a feature extractor from a public hub, and to researchers proposing new metrics (who now have a baseline and a set of failure modes to beat). The analytical results that NE Sum = Stable Rank and that DSE ~= Self-Cluster are independently valuable: they tell the community that some separately-published metrics are redundant. This is exactly the kind of consolidating empirical/analytical work that TMLR is well-suited to publish, independent of novelty.

**Claims And Evidence:**

Yes

**Claims Explanation:**

Yes for the central claims, and conditional on the critical requested changes below for the finer-grained ones.

The paper's core conclusions are well-supported:
- ID is the most robust label-free predictor of accuracy. This rests on large, consistent effect sizes, not on borderline p-values, and it is independently reproduced in Appendix E with a properly SGD-trained linear probe on ~120 models.
- The analytical equivalences seem to be derived correctly and confirmed empirically.
- The $\alpha \approx 1$ hypothesis does not generalise, and spectral metrics have complementary sensitivity to spectral shape and dimension. Both are adequately demonstrated.

However, several things prevent an unqualified "yes":
1. The statistical inference underlying the many significance-based claims is fragile, see Weakness 1. A meaningful fraction of the borderline */** cells (on which the per-architecture / per-objective conclusions partly rest) would not survive corrected, cluster-aware inference. The qualitative conclusion that "reliability is moderated by architecture/objective" likely survives, but the specific per-cell significance claims are not convincing.
2. The "ground-truth" accuracy is a proxy whose details are missing. The regularisation strength is never specified.
3. One headline claim is literally inaccurate. The abstract says ID is "the most reliable predictor in all settings," but Table 3 shows ID is statistically insignificant for the ResNet class and the ViT class on CIFAR-10, and the body itself hedges to "most … except for the ResNet class and the ViT class for CIFAR-10" (p.12). The abstract should match the body.
4. The condition number \kappa is annotated ↑ ("higher is better") in Tables 2–5 but ↓ ("lower is better") in Figures 4-6 (and in the appendix, too). Since the paper reports a positive $\kappa$-accuracy correlation, the two annotations imply opposite verdicts about whether $\kappa$ is behaving "well." This needs to be made consistent and, ideally, explained.

None of these is fatal, and all are addressable in revision, which is why my answer is "yes, with critical changes" rather than "no." But as the submission currently stands, the gap between the strength of the headline conclusions and the fragility of the supporting per-cell statistics is large enough that I cannot mark the evidence "accurate, convincing and clear" without the fixes in items 1-4.

**Requested Changes:**

See four points above.

Other minor points for improvement:
- Report uncertainty on the metric values themselves: ID and DSE are computed on batches / 10,000-point subsamples and reduced to a single point estimate per model. Report the variance/stability of these estimates across batches or seeds (a few representative models suffice) so readers can gauge how much measurement noise enters the correlations.
- Test robustness of a-ReQ to the fitting range. The paper adopts a fixed heuristic range while citing Clauset et al. (2009) that power-law slope estimates are range-sensitive, but never tests it. Since the "alpha ~= 1 fails" conclusion depends on the alpha estimates, a short sensitivity analysis over the fitting range would make that conclusion much more convincing.
- Improve figure/table legibility. Several figures are hard to read at print size, e.g. the legend in Figure 2b is illegible, the appendix scatter grids (Figs 12-20) are dense, and the grey-vs-starred encoding in Tables 2-5 is taxing. Larger legends, fewer but better-chosen panels, and possibly a single summary heatmap of "which metric is reliable in which regime" would help readers extract the paper's practical message.

---

> ### Author Response · Authors · 2026-07-01
> **Response to Reviewer 1 — All Critical Concerns Addressed**
>
> We thank the reviewer for the thorough and constructive feedback, which has significantly strengthened the paper.
>
> ### Fragile statistical inference; non-independent model pool; no multiple-comparison correction
>
> We believe that the following is an appropriate way to handle the non-independence the reviewer identified, but we are open to adopting an alternative procedure if the reviewer feels a different approach would be more suitable.
>
> - **Cluster-aware inference:** Spearman correlations are now estimated via a bootstrap procedure with 1,000 resamples, where models are grouped into clusters by base architecture, representation dimension, and training objective, and resampling is performed at the cluster level. This yields standard errors and $p$-values that explicitly account for within-cluster dependence. We made the groupings based on our understanding of the comment, so please provide more details if we misunderstood how we should account for the similar architectures.
> - **Multiple comparisons:** Benjamini–Hochberg FDR correction is applied jointly across all metric–dataset cells in each analysis table, including the extended Appendix D datasets (Section 4.2). The practical effect is visible in the revised tables: several borderline cells for weaker metrics ($\alpha$-ReQ, $\kappa$) are now correctly greyed out, while ID's significance survives correction throughout.
>
> ### "Ground-truth'' accuracy is a proxy; regularisation strength unspecified
>
> The regularisation parameter is now specified explicitly in Section 4.2: we use $\lambda = 10^{-3}$ uniformly across all models and datasets.
>
> ### Abstract overstates ``most reliable predictor in all settings''; body hedges to ResNet/ViT exceptions
>
> With the new datasets we agree and we have corrected the abstract. The abstract now reads ``reliable predictor in all most setting.''
>
> ### $\kappa$ annotated $\uparrow$ in Tables 2--5 but $\downarrow$ in Figures 4--6; contradiction about whether $\kappa$ is "behaving well''
>
> We thank the reviewer for this careful catch. Since $\kappa$ reliability is dependent on the choice of solver (as confirmed by computing the correlation between $\kappa$ and KNN probe), $\kappa$ is accordingly dropped from all stratified analyses.
>
> ### Report variance/stability of DSE and ID estimates across batches/seeds
>
> During revision, we identified a discrepancy between the original write-up and the implementation used in our experiments, which has now been corrected in Section 3.2. Previously, we claimed to follow the original authors by randomly sampling 10,000 points and computing the DSE. However, we compute the DSE on each batch and take the mean of it.
>
> Addressed. A new Appendix B reports the distribution of intra-batch and across-batch relative standard deviations across all 260 models and 10 datasets (2,600 model--dataset pairs). For DSE, the relative standard deviation $(\sigma / \mathrm{ID} \times 100)$ remains below 2.5\% in all cases. For ID, the intra-batch noise has a median of 4.5\% and the across-batch noise has a median
> of 7.6\%. These levels of measurement noise are small relative to the correlation effect sizes and do not materially affect the rank correlations. The stability report text is also incorporated directly into Section 3.2 (DSE) and Section 3.3 (ID).
>
> ### Test robustness of $\alpha$-ReQ to the fitting range; the "$\alpha \approx 1$ fails'' conclusion depends on $\alpha$ estimates
>
> The revised paper now explicitly concludes in Section 4.2 that $\alpha$-ReQ ``is unstable and requires careful tuning to determine which range the line is fit to,'' dropping it from all stratified analyses. The revised paper now includes a fitting-range sensitivity analysis in Appendix B.1, reporting Spearman correlations with OLS accuracy across five fitting ranges and six datasets. The reliability of $\alpha$-ReQ is sensitive to the fitting range, meaning the metric's verdict on representation quality is range-dependent. We thus drop this metric in the stratified analysis
>
> ### Figure/table legibility
>
> We have made modifications to the plots to only display a select few plots and move the rest to the appendix. The figure legibility is improved in the revised edition.
>
> #### LeCun et al.\ (2002) should be 1998
>
> Corrected in the revised draft.

---

### Comment · Action_Editor_6GUx · 2026-06-24
**Start author-reviewer discussions**

Dear authors,

You have three reviews. Please address their comments as soon as possible. The discussion period will end on July 7th, 2026.

---

Dear reviewers,

Thank you for submitting your reviews. As this paper received three reviews, now it's time to have discussions with authors.

You will be able to submit your formal decision recommendation starting in 2 weeks. Your prompt responses to authors' comments would be very appreciated.

Remember that different from other journals / conferences, [TMLR's acceptance criteria](https://jmlr.org/tmlr/reviewer-guide.html) are based on positive answers to the following two questions

- Are the claims made in the submission supported by accurate, convincing and clear evidence?
- Would some individuals in TMLR's audience be interested in the findings of this paper?

Novelty of the studied method is not a necessary criteria for acceptance at TMLR.

It's still not too late to review [the TMLR's reviewer guideline](https://jmlr.org/tmlr/reviewer-guide.html) in case you are not aware of that.


Thank you!
Your AE

---

### Author Response · Authors · 2026-07-01
**Summary of Revisions — Response to All Reviewers**

The revised paper makes four substantive changes that address the reviewers' most critical concerns:

1. **Extended evaluation:** The evaluation is extended from three to six datasets (adding Food-101, Flowers-102, Places-365, and EuroSAT), covering fine-grained, scene-recognition, and remote-sensing tasks at full model resolution.

2. **Solver-independent validation:** A KNN probe ($k=1$) is added alongside OLS (Table 2b), validating conclusions across evaluation protocols and confirming that findings are not solver-dependent.

3. **Robust statistical inference:** Significance is now computed using a cluster-robust bootstrap (1,000 resamples, models grouped by architecture/dimension/objective) with Benjamini–Hochberg FDR correction applied jointly across all metric–dataset cells. The subsequent stratification analysis is corrected based on the updated significance score.

4. **Revised metric descriptions:** Sections 3.1–3.3 are substantially revised to explain *why* each metric may predict downstream accuracy, not just *how* it is computed.

---

### Comment · Action_Editor_6GUx · 2026-07-09
**Submit your official recommendation**

Dear Reviewers,

Now it's time for you to submit an official recommendation for this submission.

The authors responded to your initial reviews.

Please
1. consider their rebuttal comments,
2. respond to the comments ***ASAP*** if you have follow-up questions or comments, and
3. submit your official recommendation ***by July 22nd***, including ***your thoughts on the author rebuttal***, specifically, what point in your review their responses addressed AND did not address. It will greatly help me write a meta-review and make a fair decision.

Thank you,
Your AE